# Training Dynamics of Transformers to Recognize Word Co-occurrence via Gradient Flow Analysis

**Hongru Yang**[*]
The University of Texas at Austin
& Princeton University
hy6385@utexas.edu

**Bhavya Kailkhura**
Lawrence Livermore National Laboratory
kailkhura1@llnl.gov

**Zhangyang Wang**
The University of Texas at Austin
atlaswang@utexas.edu

**Yingbin Liang**
The Ohio State University
liang.889@osu.edu

## Abstract

Understanding the training dynamics of transformers is important to explain the impressive capabilities behind large language models. In this work, we study the dynamics of training a shallow transformer on a task of recognizing co-occurrence of two designated words. In the literature of studying training dynamics of transformers, several simplifications are commonly adopted such as weight reparameterization, attention linearization, special initialization, and lazy regime. In contrast, we analyze the gradient flow dynamics of simultaneously training three attention matrices and a linear MLP layer from random initialization, and provide a framework of analyzing such dynamics via a coupled dynamical system. We establish near minimum loss and characterize the attention model after training. We discover that gradient flow serves as an inherent mechanism that naturally divide the training process into two phases. In Phase 1, the linear MLP quickly aligns with the two target signals for correct classification, whereas the softmax attention remains almost unchanged. In Phase 2, the attention matrices and the MLP evolve jointly to enlarge the classification margin and reduce the loss to a near minimum value. Technically, we prove a novel property of the gradient flow, termed *automatic balancing of gradients*, which enables the loss values of different samples to decrease almost at the same rate and further facilitates the proof of near minimum training loss. We also conduct experiments to verify our theoretical results.

## 1 Introduction

Ever since the invention of self-attention [VSP+17], transformers have become a dominating backbone architecture in many machine learning applications such as computer vision [DBK+20, LLC+21] and natural language processing [DCLT18]. Nowadays, ChatGPT and GPT-4 [Ope23] have demonstrated astonishing abilities in many areas such as language understanding, mathematics and coding, which have sparked artificial general intelligence [BCE+23]. In the meantime, there has been a burgeoning development of large language models (LLMs) [TLI+23, MH23, ADF+23] as well as multi-modal models [Tea23].

Despite the huge empirical success, theoretical understanding of why a pre-trained language model can possess such impressive performance has been significantly lagging behind. Some previous

---

[*]Work done while doing a internship at Lawrence Livermore National Laboratory and visiting Princeton University.

38th Conference on Neural Information Processing Systems (NeurIPS 2024).

efforts have been made in understanding the capacity and representational power of transformers [EGKZ22, LAG$^+$22, ZPGA23, SHT23, BCW$^+$23]. However, most results of this type of works are existential and rely on manual construction of the weights. It is unclear whether the constructed weights are the actual solutions after training transformers. In order to understand the mechanism behind those pre-trained language models, a line of studies have aimed to open the black box of optimization via studying training dynamics of transformers and explaining why transformers can be trained to perform well [JSL22, LWLC22, TWCD23, TWZ$^+$23, LLR23, TLZO23, TLTO23, ZFB24, HCL23]. However, those previous works often relied on various simplifications in their analysis such as weight reparameterization, attention linearization, special initialization, lazy regime, etc. One goal of this paper is to take a further step to demystify the training dynamics of transformers and consider more practical training setup, thus better capturing the actual training process.

Our study of transformers' training dynamics will focus on a basic problem of recognizing co-occurrence of words under a binary classification setup, which is an important ability of LLMs to perform many tasks correctly in natural language processing (NLP). For example, the classical n-gram model [MS99, Dam18] predicts the next word based on co-occurrence of multiple words. Consider the following scenario: if the task for the language model is to read a paragraph describing a children and then answer some questions, say, "Is Bob eating a banana?". In order to answer the question correctly, the model must be able to detect the co-occurrence of the two words "Bob" and "banana" in the paragraph. Motivated by this, we study the problem of detecting co-occurrence of two target words via the model of a one-layer transformer with a self-attention module followed by a linear multi-layer perceptrons (MLP) layer. Our goal is to characterize the dynamics of the training process via the gradient flow analysis, thus providing a theory to explain how transformers can be trained to perform well.

Our contribution is summarized below:

- We study the gradient flow dynamics of detecting word co-occurrence. The training starts with random initialization and then simultaneously updates *four* weight matrices (including key, query, and value matrices and a linear MLP) in the transformer architecture via gradient flow. We show that gradient flow can achieve small loss although the loss function is highly nonconvex. We further characterize the explicit form of attention matrices after training, which captures the strong positive correlation between the two target signals and strong negative correlation between one target signal and the common token, both leading to large classification margin.

- We characterize the training process into two phases. In Phase 1 (alignment of MLP for correct classification), we show that the linear MLP of the transformer quickly aligns with the two target word tokens whereas all other variables in the dynamical system stay almost unchanged from their initialization values. All training samples are correctly classified at the end of Phase 1, but the loss value is still large due to small classification margin. In Phase 2 (evolution of attention and MLP for large classification margin), along with the continual evolution of MLP, correct classification by MLP also encourages the gradients of attention matrices to learn. Specifically, the softmax probability increases if the key and query tokens correspond to the two target words, and the value transform of the two words becomes more positively correlated, both leading to enlarge the classification margin. Thus, the training and test loss values both are driven down to nearly zero.

- Technically, our proof techniques do not rely on several commonly used assumptions in the literature such as weight reparameterization, attention linearization, special initialization, lazy regime, etc. Our main idea is to treat the problem as a *coupled* dynamical system with *six* different types of dynamic variables, for which we provide an articulated analysis on the gradient flow dynamics. In particular, we prove a novel property of the gradient flow, termed *automatic balancing of gradients*, which shows that the ratio of several important gradients will evolve closely within the same range during training. This enables us to show that the losses of all training samples can decrease almost at the same rate, and is also a key component in proving the near minimum training loss as well as analyzing the changes of softmax.

## 1.1 Related Work

**Transformer representational power.** Several previous works have studied the expressiveness of transformers. One line of work was from a universal approximation perspective and thus provided the existential results [YBR$^+$19, WCM22, PBM21, ZPGA23, LAG$^+$22, BCW$^+$23]. As a separate view, [EGKZ22] showed that a single attention head can represent a sparse function over the input

sequence with sample complexity much smaller than the context length. [ZZYW23] studied the approximation and generalization performance of transformers in in-context learning. [SHT23] proved that transformers can represent certain functions more efficiently than MLPs.

**Training transformers.** Various settings of training transformers have been studied recently. [WXM21] studied the impact of head and prompt tuning of transformer on the downstream learning tasks. [JSL22] proved that transformers can learn spatial structures. [LWLC22] studied how a shallow transformer learns a dataset with both label-relevant and label-irrelevant tokens. [TWCD23] studied a next-token prediction problem and showed that self-attention behaves like a discriminating scanning algorithm. [LLR23] analyzed a layer-wise optimization scheme on how transformers learn topic structures. [TLZO23, TLTO23, VDT24] studied a setting where transformers can learn a SVM solution. [LWM$^+$23] provided analysis of training graph transformers for node classification tasks. [Thr24] studied the implicit bias in the next-token prediction problem. For in-context linear regression, [VONR$^+$23] constructed transformer weights to solve this task and showed empirically that this is similar to what the transformer learned by gradient descent, [ACDS24] proved that the critical points of the training objective of linear transformers implement a pre-conditioned gradient descent, [ZFB24] provided the training dynamics of linear attention models, [HCL23] characterized the training dynamics of softmax transformers, and [CSWY24] studied a multi-task linear regression problem with a multi-headed softmax transformer. Further, [LWL$^+$24] focused on nonlinear self-attention and nonlinear MLP over classification tasks in in-context learning. [WLCC23] proved the convergence of transformers via neural tangent kernel. [NDL24] showed that two-layer transformers can learn causal structure via gradient descent. [CL24] developed algorithms for provably learning a multi-head attention layer. [HWCL24] studied how transformers learn feature-position correlation.

This paper studies a different problem of detecting co-occurrence of words via transformers. Such a setting has not been considered in the literature. More importantly, the previous studies of training dynamics of transformers have adopted various assumptions/simplifications such as weight reparameterization, special initialization, attention linearization, lazy regime, etc. In contrast, our analysis here based on gradient flow does not rely on those simplifications, which can be of independent interest for studying transformers in other settings.

## 2 Problem Setting

**Notations.** For a vector $v \in \mathbb{R}^d$, we use $\mathrm{diag}(v)$ to denote a diagonal matrix with $v$ being the diagonal entries. When we subtract the vector $v$ by a scalar $a$, we subtract each entry of $v$ by $a$, i.e., $v - a \in \mathbb{R}^d$ and $(v - a)_i = v_i - a$. We use $\widetilde{\Omega}, \widetilde{\Theta}, \widetilde{O}$ to hide polylogarithmic factors.

### 2.1 Data Model

**Definition 2.1** (Data distribution). *Given a set of orthonormal vectors $\{\mu_i\}_{i=1}^d$ as word embedding, let $\mu_1, \mu_2 \in \mathbb{R}^d$ be two target signals whose co-occurrence needs to be detected by the model, and let $\mu_3 \in \mathbb{R}^d$ be a common token vector. A data entry $(X, y) \in \mathbb{R}^{d \times L} \times \{\pm 1\}$, where $X = [x_1, x_2, \ldots, x_L]$ consists of $L$ tokens, is generated by the distribution $\mathcal{D}$ as follows:*

1. *Uniformly randomly select an index $i_3 \in [L]$ and set $x_{i_3} = \mu_3$.*

2. *Then, one of the following cases occurs:*

   - *With probability $1/2$, set $y = 1$ and uniformly randomly select two indices $i_1 \neq i_2 \in [L] \setminus \{i_3\}$ and set $x_{i_1} = \mu_1$, $x_{i_2} = \mu_2$. For $i \in [L] \setminus \{i_1, i_2, i_3\}$, set $x_i = \mathtt{Uniform}(\{\mu_i\}_{i=4}^d)$.*
   - *With probability $1/6$, set $y = -1$ and uniformly randomly select one index $i_1 \in [L] \setminus \{i_3\}$ and set $x_{i_1} = \mu_1$. For $i \in [L] \setminus \{i_3, i_1\}$, we set $x_i = \mathtt{Uniform}(\{\mu_i\}_{i=4}^d)$.*
   - *With probability $1/6$, set $y = -1$ and uniformly randomly select one index $i_2 \in [L] \setminus \{i_3\}$ and set $x_{i_2} = \mu_2$. For $i \in [L] \setminus \{i_3, i_2\}$, we set $x_i = \mathtt{Uniform}(\{\mu_i\}_{i=4}^d)$.*
   - *With probability $1/6$, set $y = -1$. For all $i \in [L] \setminus \{i_3\}$, we set $x_i = \mathtt{Uniform}(\{\mu_i\}_{i=4}^d)$.*

In summary, there are 4 types of data: (1) both $\mu_1, \mu_2$ appear, (2) only $\mu_1$ appears, (3) only $\mu_2$ appears, and (4) neither $\mu_1$ nor $\mu_2$ appears. We denote the set of indices of the above 4 different types of data by $I_1, I_2, I_3, I_4 \subseteq [n]$. We further define $\mathcal{R} = \{\mu_i\}_{i=4}^d$. For simplicity, our data distribution assumes $\mu_1, \mu_2, \mu_3$ appear only once in a data entry. The occurrence probability of each type of

data is chosen in the above way to make the distribution label-balanced. We assume there is a fixed set of orthonormal vectors as word embedding, which is analogous to the one-hot embedding of a set of vocabularies. Furthermore, in our daily language, there are some words appearing in almost every sentence such as "a" and "the". Thus, to model those words, we include a common token in every data entry. Finally, notice that if we ignore the common token and random tokens, the data distribution simplifies to a logical AND problem.

**Remark 2.2.** *Recognizing co-occurrence of words is an important ability for language models to perform many NLP tasks correctly. Consider the example of a language model first reading a paragraph describing a children and then answering the question "Is Bob eating a banana?" If the description is "Bob is watching a television while eating a banana", then the model should answer "Yes". If the description is "Bob is playing computer games", then the model should answer "No". Thus, the model needs to recognize the co-occurrence of "Bob" and "banana".*

For simplicity of our analysis, we make the following assumption on our training data set.

**Assumption 2.3.** *The training set satisfies: (i) $\frac{|I_1|}{n} = \frac{1}{2}$ and $\frac{|I_2|}{n} = \frac{|I_3|}{n} = \frac{|I_4|}{n} = \frac{1}{6}$; and (ii) for all $i_1, i_2 \in [n]$, $l_1, l_2 \in [L]$, if $X_{l_1}^{(i_1)}, X_{l_2}^{(i_2)} \notin \{\mu_1, \mu_2, \mu_3\}$, then $X_{l_1}^{(i_1)} \neq X_{l_2}^{(i_2)}$, i.e., all irrelevant words are different.*

The first assumption can be approximately satisfied with high probability given the total number $n$ of samples is large enough. Such an assumption can be removed by applying the standard concentration theorems. The second assumption implicitly assumes $nL \leq d$. If the irrelevant words are uniformly sampled from a large entire vocabulary, then each irrelevant word appears only very few times in the training set. Thus, letting irrelevant words appear only once in the entire training set is a reasonable way to simplify our analysis.

## 2.2 Transformer Architecture and Training

Consider a training set $\{(X^{(i)}, y_i)\}_{i=1}^n$ with $n$ training samples. Each data point $X^{(i)} \in \mathbb{R}^{d \times L}$ contains $L$ tokens, i.e., $X^{(i)} = [x_1^{(i)}, x_2^{(i)}, \ldots, x_L^{(i)}]$. We consider the transformer model with a self-attention module followed by a linear MLP:

$$F(X; W, W_V, W_K, W_Q) = \sum_{l=1}^{L} \sum_{j=1}^{m_1} a_j \left( w_j^\top W_V X \cdot \text{Softmax}\left( \frac{X^\top W_K^\top W_Q x_l}{\sqrt{m}} \right) \right) \qquad (1)$$

where the query matrix $W_Q \in \mathbb{R}^{m \times d}$, the key matrix $W_K \in \mathbb{R}^{m \times d}$, the value matrix $W_V \in \mathbb{R}^{m \times d}$, the hidden-layer MLP weights $W \in \mathbb{R}^{m_1 \times m}$ (with $w_j^\top$ being the $j$-th row of $W$), and the output-layer weights of the MLP $a \in \mathbb{R}^{m_1}$. We define the linear MLP function of the transformer to be $G(\mu) = \sum_{j=1}^{m_1} a_j w_j^\top W_V \mu$. We now introduce some shorthand notations $K = W_K X$, $Q = W_Q X$, $V = W_V X$ and let $k_l = W_K x_l$. Notice that $K = [k_1, k_2, \ldots, k_L]$. We further extend this shorthand to $q_l$ and $v_l$. We also define functions $k(\mu) = W_K \mu$, $q(\mu) = W_Q \mu$, $v(\mu) = W_V \mu$. We introduce the shorthand for the **score vector** $s_l := \frac{X^\top W_K^\top W_Q x_l}{\sqrt{m}}$ and the attention vector $p_l := \text{Softmax}(s_l)$. For the attention vector, if $\mu, \nu \in X^{(i)}$, let $l(i, \mu), l(i, \nu)$ be the indices such that $X_{l(i,\mu)}^{(i)} = \mu$, $X_{l(i,\nu)}^{(i)} = \nu$, and we define $p_{q \leftarrow \mu, k \leftarrow \nu}^{(i)} := p_{q \leftarrow l(i,\mu), k \leftarrow l(i,\nu)}^{(i)} := \text{Softmax}\left( \frac{X^{(i)\top} W_K^\top W_Q \mu}{\sqrt{m}} \right)_{l(i,\nu)}$.

**Initialization.** We initialize $a_j \overset{i.i.d.}{\sim} \text{Uniform}(\pm 1)$ and the value of $a$ is fixed during training. The trainable parameters are $[W, W_V, W_K, W_Q]$. We initialize $[W, W_V, W_K, W_Q]$ by $W_{i,j} \overset{i.i.d.}{\sim} \mathcal{N}(0, \sigma_1^2)$ and $(W_V)_{i,j}, (W_K)_{i,j}, (W_Q)_{i,j} \overset{i.i.d.}{\sim} \mathcal{N}(0, \sigma_0^2)$.

**Training.** We adopt the **cross-entropy loss** $l(x) = \log(1 + \exp(-x))$. The gradient of the cross-entropy loss is given by $l'(x) = -\frac{1}{1+\exp(x)}$ and we define $g(x) = \frac{1}{1+\exp(x)}$. The model is trained by gradient flow to minimize the following empirical loss:

$$\widehat{L}(W, W_V, W_K, W_Q) = \frac{1}{n} \sum_{i=1}^{n} l(y_i F(X^{(i)}; W, W_V, W_K, W_Q)). \qquad (2)$$

Similarly, we define the generalization loss $L := \mathbb{E}_{(X,y)\sim\mathcal{D}}\,\ell(yF(X))$. We introduce the parameter condition that we take throughout the entire analysis and proofs.

**Condition 1.** *We make the following parameter choices in our analysis:*

- *The embedding dimension and network width satisfy $m \geq \widetilde{\Omega}(\max\{m_1, L^2\})$ and $m_1 \geq \widetilde{\Omega}(1)$.*

- *The network weight initialization variance satisfies $\sigma_0 = \frac{1}{\widetilde{\Theta}(\sqrt{Lm})}$; and $\sigma_1 = \frac{1}{\widetilde{\Theta}(\sqrt{m_1})}$.*

- *The number of training samples and tokens satisfy $n \geq \widetilde{\Omega}(L^2)$ and $L \geq \widetilde{\Omega}(1)$.*

- *The failure probability satisfies $1/\delta \leq \mathrm{poly}(m)$.*

## 3   Main Results

**Challenges.** The essential goal is to derive the gradient flow update for each weight matrix (see the gradient expressions for all weight matrices in Appendix B). However, directly analyzing the dynamics of those weight matrices is extremely challenging, because: (i) keeping track of how the column and row spaces of each weight matrix change during training is difficult; and (ii) all attention and MLP weight matrices are affecting each other, leading to highly coupled dynamics.

**Our General Idea.** To overcome the above challenges, we first note that rather than tracking $W, W_V, W_K, W_Q$ directly, it is sufficient to analyzing their impact on inputs, i.e., $X^\top W_Q^\top W_K X$ and $a^\top W W_V X$, which are sufficient to compute $F(X; W, W_V, W_K, W_Q)$. Based on this observation, we formulate two *differential equations* to keep track of $w_j^{(t)\top} W_V^{(t)} \mu$ and $\nu^\top W_K^{(t)\top} W_Q^{(t)} \mu$ (with respect to $t$) for all $\mu, \nu \in \{\mu_i\}_{i=1}^d$ (See Equation (6) and Equation (7) in Appendix C). We further include additional equations to keep track of $\left\langle w_{j_1}^{(t)}, w_{j_2}^{(t)} \right\rangle$, $\nu^\top W_V^{(t)\top} W_V^{(t)} \mu$, $\nu^\top W_K^{(t)\top} W_K^{(t)} \mu$, and $\nu^\top W_Q^{(t)\top} W_Q^{(t)} \mu$ to complete the system. Intuitively, the additional equations keep track of the shape of the neurons and the word embedding after $W_V, W_K, W_Q$ transform. Although the dynamical system does not directly track the softmax, the softmax probability can be calculated via the scores of $\nu^\top W_K^{(t)\top} W_Q^{(t)} \mu$. The full dynamical system is presented in Appendix C. Then the training dynamics can be characterized by analyzing these differential equations (see a proof outline in Section 4).

In the next two theorems, we present our characterization of the training process into two phases.

**Theorem 3.1** (Phase 1). *With probability at least $1 - \delta$ over the randomness of weight initialization, there exists a time $T_1 = \widetilde{O}(1/m)$ such that*

- *The linear MLP functions satisfy: $G^{(T_1)}(\mu_1) \geq \Omega(1)$, $G^{(T_1)}(\mu_2) \geq \Omega(1)$, $G^{(T_1)}(\mu_3) \leq -\Omega(1)$.*

- *All training samples are correctly classified: $y_i F_i^{(T_1)} > 0$ for all $i \in [n]$.*

- *For $t \in [0, T_1]$, all dynamical variables $\left\langle w_{j_1}^{(t)}, w_{j_2}^{(t)} \right\rangle$, $\nu^\top W_K^{(t)\top} W_Q^{(t)} \mu$, $\nu^\top W_V^{(t)\top} W_V^{(t)} \mu$, $\nu^\top W_K^{(t)\top} W_K^{(t)} \mu$, and $\nu^\top W_Q^{(t)\top} W_Q^{(t)} \mu$ are close to their initialization values.*

- *Training loss is still large: $\widehat{L}^{(T_1)} = \Theta(1)$.*

In Theorem 3.1, item 1 implies that in a short time, the linear MLP function $G^{(T_1)}(\cdot)$ *positively* aligns with the two target signals $\mu_1$ and $\mu_2$, but *negatively* aligns with the common token $\mu_3$. This further guarantees item 2 of Theorem 3.1 that all training samples are classified correctly. Further, item 3 of Theorem 3.1 indicates that the attention matrices are still close to their initialization values, and hence have not started to learn any knowledge yet. This results in item 4 of Theorem 3.1, which shows that the training loss is still large.

**Theorem 3.2** (Phase 2). *With probability at least $1 - \delta$ over the randomness of weight initialization, there exists a time range $(T_1, T_2)$ with $T_2 = \mathrm{poly}(m)$ such that for all $t \in (T_1, T_2)$*

- $\mu_2^\top W_K^{(t)\top} W_Q^{(t)} \mu_1$ *and* $\mu_1^\top W_K^{(t)\top} W_Q^{(t)} \mu_2$ *increase, whereas* $\mu_3^\top W_K^{(t)\top} W_Q^{(t)} \mu_1$ *and* $\mu_3^\top W_K^{(t)\top} W_Q^{(t)} \mu_2$ *decrease.*

- $\mu_1^\top W_V^{(t)\top} W_V^{(t)} \mu_2$ increases, whereas $\mu_1^\top W_V^{(t)\top} W_V^{(t)} \mu_3$ and $\mu_2^\top W_V^{(t)\top} W_V^{(t)} \mu_3$ decrease.

- Linear MLP functions satisfy: $G^{(t)}(\mu_1) \geq \Omega(1)$, $G^{(t)}(\mu_2) \geq \Omega(1)$, $-G^{(t)}(\mu_3) \leq \Omega(1)$.

- $G^{(t)}(\mu_1) + G^{(t)}(\mu_2) + G^{(t)}(\mu_3) \geq \Omega(1)$, $G^{(t)}(\mu_1) + G^{(t)}(\mu_3) \leq -\Omega(1)$ and $G^{(t)}(\mu_2) + G^{(t)}(\mu_3) \leq -\Omega(1)$.

In Theorem 3.2, item 1 indicates that, during Phase 2, gradient flow drives the self-attention module to weigh more between the two target signals $\mu_1$ and $\mu_2$, and to weigh less between one of these signals and the common token $\mu_3$. Item 2 indicates that gradient flow drives the value matrix $W_V$ to positively align the two target signals $\mu_1$ and $\mu_2$, but negatively align one target signal ($\mu_1$ or $\mu_2$) with the common token $\mu_3$. Further, the last two items indicate that the MLP continue to classify correctly and further enlarge the classification margin. Hence, all items in Theorem 3.2 collectively indicate that attention and MLP evolve jointly to enlarge the classification margin and hence drive the loss value to decrease in Phase 2.

**Theorem 3.3** (Near Minimum Training Loss and Attention). *With probability at least $1 - \delta$, there exists a time $T^\star = \Theta(\text{poly}(m))$ such that*

- *The training and generalization losses satisfy $\widehat{L}^{(T^\star)} \leq 1/\text{poly}(m)$ and $L^{(T^\star)} \leq 1/\text{poly}(m)$.*

- *The attention matrices satisfies:*

$$W_K^{(T^\star)\top} W_Q^{(T^\star)} = W_K^{(0)\top} W_Q^{(0)} + \sum_{i_1,i_2 \in [d]} C_{i_1,i_2}^{(T^\star)} \mu_{i_1} \mu_{i_2}^\top, \tag{3}$$

*where* $C_{1,2}^{(T^\star)}, C_{2,1}^{(T^\star)}, -C_{3,1}^{(T^\star)}, -C_{3,2}^{(T^\star)} = \Theta\left(\frac{\sigma_0^2 m}{\sqrt{m}L\sigma_1^2 mm_1} + \frac{\sigma_0^2 \sqrt{m}}{\sqrt{m}\sigma_1^2 mm_1}\right)$ *and* $C_{i_1,i_2}^{(T^\star)} \leq \widetilde{O}\left(\frac{\sigma_0^2 m}{n\sqrt{m}L\sigma_1^2 mm_1} + \frac{\sigma_0^2 \sqrt{m}}{\sqrt{m}\sigma_1^2 mm_1}\right)$ *if one of $i_1, i_2 \in [d] \setminus [3]$.*

Theorem 3.3 indicates that both training and test losses converge nearly to zero as long as the embedding dimension $m$ is sufficiently large, because both the attention and MLP matrices are trained towards enlarging the classification margin in Phase 2. Theorem 3.3 also provides the explicit form of the attention matrix in Equation (3), in which the second term captures the learned information of the self-attention module. It can be seen that the large coefficients $C_{1,2}^{(T^\star)}$ and $C_{2,1}^{(T^\star)}$ capture strong coupling of the two target signals $\mu_1$ and $\mu_2$, and the large negative coefficients $C_{3,1}^{(T^\star)}$ and $C_{3,2}^{(T^\star)}$ encourages strong negative coupling of one target signal $\mu_1$ or $\mu_2$ and the common token $\mu_3$. All these attention terms contribute to enlarge correct classification margin. Further, the coefficients between all other random tokens are order-level smaller and hence do not corrupt the correct classification.

**Synthetic Experiment:** We next verify our theory and the two-phase characterization of the training process via synthetic experiments (see the experiment setup in Appendix A).

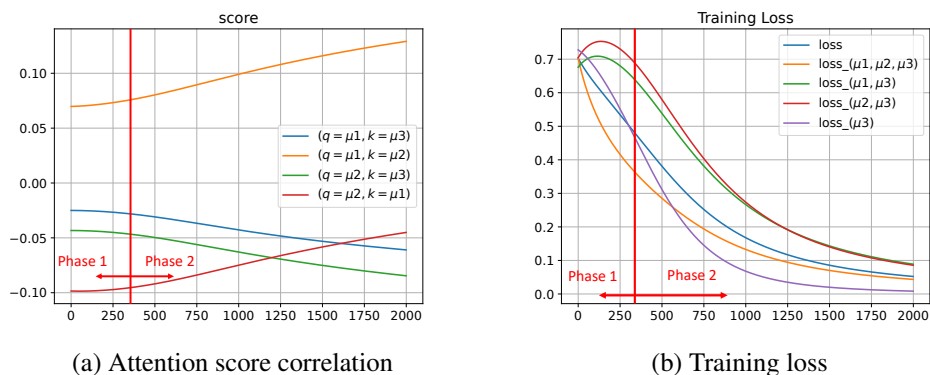

(a) Attention score correlation        (b) Training loss

Figure 1: Synthetic experiments with illustration of two training phases. The detailed experiment setup can be found in Appendix A.

Figure 1 (a) shows how the attention score correlation $\mu_i^\top W_K^{(t)\top} W_Q^{(t)} \mu_j$ evolves during the training. It is clear that these scores do not change significantly in Phase 1, verifying Theorem 3.1. In Phase 2, the score correlation between two target signals $\mu_1$ and $\mu_2$ increases, and the score between one target signal and the common token decreases, verifying Theorem 3.2.

Figure 1 (b) plots how the training loss changes during the two phases of training. The blue curve (indexed by 'loss') represents the overall training loss of all samples. The other curves correspond to the training loss of four types of samples as indicated in the legend. In Phase 1, the training loss for samples with both target signals (i.e., orange curve) decreases because the linear MLP layer aligns with the target signals (verifying Lemma 4.1 in Section 4.1). The training loss for samples with one target signal and the common token (i.e., green or red curves) first increases because the linear MLP layer initially has not aligned negatively enough with the common token yet (as captured by Lemma 4.2 in Section 4.1), and then decreases in the later stage of Phase 1 when the MLP layer aligns negatively with the common token (as captured by Lemma 4.3 in Section 4.1). All loss functions decrease in Phase 2 because all attention matrices and linear MLP jointly enlarge the classification margin, verifying Theorem 3.2.

## 4 Proof Outline: Two-phase Gradient Flow Analysis

### 4.1 Phase 1: Alignment of Linear MLP for Correct Classification

In Phase 1, the linear MLP quickly aligns with the two target word tokens while all attention matrices stay roughly unchanged from their initialization values. We show that the linear MLP functions $|G^{(t)}(\mu_1)|, |G^{(t)}(\mu_2)|, |G^{(t)}(\mu_3)|$ become sufficiently large (larger than some constant threshold) so that all training samples are correctly classified at the end of Phase 1.

We first analyze the dynamical system at the initialization. In particular, the following lemma shows that at the initialization, the linear MLP layer receives a sufficiently large gradient from the two target signals, and hence samples with the two target signals will be classified correctly as co-occurrence soon afte the training starts.

**Lemma 4.1** (Same as Lemma E.4). *With probability at least* $1 - \delta$ *over the weight initialization,*

$$\forall \mu \in \{\mu_1, \mu_2\}: \quad \frac{\partial a_j w_j^{(0)} W_V^{(0)} \mu}{\partial t} = \Theta((\sigma_0^2 + \sigma_1^2)m).$$

Further, by the definition of Phase 1 (see Definition E.2 for a formal definition), the gradients of the attention matrices in the dynamical system are much smaller than that of linear MLP given in Lemma 4.1. This implies that during Phase 1, mainly the linear MLP is performing learning, whereas all the attention matrices are changing slowly from their initialization. Based on this, we have $\frac{\partial}{\partial t} G^{(t)}(\mu_1) = \Theta((\sigma_0^2 + \sigma_1^2)mm_1)$ which implies that it takes only $O(1/(\sigma_0^2 + \sigma_1^2)mm_1)$ iterations for $G^{(t)}(\mu_1)$ to reach a certain constant magnitude.

Lemma 4.1 indicates that the samples with co-occurrence of the two target signals are classified correctly. The following lemma shows that the initial gradient from the common token, i.e., the gradient of $G^{(t)}(\mu_3)$, is much smaller than the gradient from the two target signals, which implies that the samples with only one target signal may be classified *incorrectly* as co-occurrence (since the network in this case will output a positive value). This is verified empirically by our experiments in Figure 1 (b), where the loss function corresponding to only one target signal and the common token first increases in Phase 1.

**Lemma 4.2** (Same as Lemma E.16). *Let* $F = \max_i |F_i^{(0)}|$. *With probability at least* $1 - \delta$ *over the weight initialization,*

$$\left| \frac{\partial a_j w_j^{(0)} W_V^{(0)} \mu_3}{\partial t} \right| = \widetilde{O}\left( \sigma_1^2 \sqrt{mm_1} + \sigma_0^2 \sqrt{mL} + \sigma_1^2 mF \right).$$

Notice that the model output $F$ depends on the weight initialization scale and can be made small.

We next show in the following lemma that the gradient $\frac{\partial G^{(t)}(\mu_3)}{\partial t}$ of the common token will quickly become negative soon after the training begins, which drives the transformer model to output a negative value when it sees those types of samples. This implies that negative samples (without co-occurrence of two target tokens) will be classified correctly towards the end of Phase 1. This is also verified empirically by our experiments in Figure 1 (b), where the loss function corresponding to only one target signal and the common token descreases towards the end of Phase 1.

**Lemma 4.3** (Abbreviated from Theorem E.19). *There exists a time* $T_{0.5} \leq T_1$ *and a constant* $C$ *such that for all* $t \in [T_{0.5}, T_1]$

$$(1 + C) \max\left( \frac{\partial G^{(t)}(\mu_1)}{\partial t}, \frac{\partial G^{(t)}(\mu_2)}{\partial t} \right) \leq -\frac{\partial G^{(t)}(\mu_3)}{\partial t} \leq (1 - C)\left( \frac{\partial G^{(t)}(\mu_1)}{\partial t} + \frac{\partial G^{(t)}(\mu_2)}{\partial t} \right).$$

*Proof Intuition of Lemma 4.3.* We first note that the term

$$\frac{1}{n} \sum_{i=1}^{n} g_i^{(0)} y_i \sum_{l_2=1}^{L} \sum_{j_2=1}^{m_1} \|w_{j_2}^{(0)}\|_2^2 (X^{(i)} p_{l_2}^{(0,i)})^\top \mu$$

makes the major contribution to the gradient $\frac{\partial G^{(t)}(\mu_3)}{\partial t}$. Such a term is small at the initialization due to the cancellation effect from positive and negative $y_i$'s. However, since the linear MLP $G$ will positively align the two target signals at the beginning, for the samples with positive labels, $g_i^{(t)}$ will decrease, whereas for samples with only one target signal, $g_i^{(t)}$ will increase. Hence, $\frac{\partial a_j w_j^{(t)} W_V^{(t)} \mu_3}{\partial t}$ will become negative. This trend will continue until the gradient from $\mu_3$ starts to match the gradients from $\mu_1, \mu_2$, which is what we establish in Lemma 4.3. $\qquad\square$

Using Lemma 4.3, we can show that all training samples are correctly classified at end of Phase 1.

## 4.2 Phase 2: Evolution of Attention and MLP for Large Classification Margin

In Phase 2, both attention and MLP matrices evolve towards enlarging the classification margin, thus driving the loss value small.

We now analyze what happens in Phase 2. Let $T_2$ denote the end of Phase 2. Recall that at the end of Phase 1, we have $G^{(t)}(\mu_1), G^{(t)}(\mu_2), -G^{(t)}(\mu_3) \geq \Omega(1)$. We will mainly need to show that such a condition continues to hold in Phase 2, so that attention matrices will evolve with MLP to learn better classifiers. To this end, we exam the following gradient flow in the dynamical system:

$$\frac{\partial G^{(t)}(\mu)}{\partial t} = \frac{1}{n} \sum_{i_2: \, \mu \in X^{(i_2)}} g_{i_2}^{(t)} y_{i_2} \sum_{l_2=1}^{L} p_{q \leftarrow l_2, k \leftarrow \mu}^{(t,i_2)} \cdot \sum_{j_1=1}^{m_1} \sum_{j_2=1}^{m_1} a_{j_1} a_{j_2} \left\langle w_{j_1}^{(t)}, w_{j_2}^{(t)} \right\rangle \qquad (4)$$
$$+ \frac{m_1}{n} \sum_{i_1=1}^{n} g_{i_1}^{(t)} y_{i_1} \sum_{l_1=1}^{L} a_{j_1} p_{l_1}^{(t,i_1)\top} V^{(t,i_1)\top} W_V^{(t)} \mu.$$

It has been proved that at the end of Phase 1, for $\mu \in \{\mu_1, \mu_2, \mu_3\}$, we have $\left| \sum_{j_1} \frac{\partial w_{j_1}^{(t)\top}}{\partial t} W_V^{(t)} \mu \right| \ll$ $\left| \sum_{j_1} w_{j_1}^{(t)\top} \frac{\partial W_V^{(t)}}{\partial t} \mu \right|$ since the magnitude of $\sum_{j_1=1}^{m_1} \sum_{j_2=1}^{m_1} \left\langle a_{j_1} w_{j_1}^{(t)}, a_{j_2} w_{j_2}^{(t)} \right\rangle$ is large. Assume this can hold for long enough (which we can indeed prove later). Then, we only need to focus on the first term in the sum on the right-hand side in Equation (4). On the other hand, from the dynamical system, we can calculate

$$\frac{\partial}{\partial t} \sum_{j_1=1}^{m_1} \sum_{j_2=1}^{m_1} \left\langle a_{j_1} w_{j_1}^{(t)}, a_{j_2} w_{j_2}^{(t)} \right\rangle = \frac{2m_1}{n} \sum_{i=1}^{n} g_i^{(t)} y_i F_i^{(t)}. \qquad (5)$$

Thus, if $y_i F_i^{(t)} > 0$ for all $i \in [n]$, then $\sum_{j_1=1}^{m_1} \sum_{j_2=1}^{m_1} \left\langle a_{j_1} w_{j_1}^{(t)}, a_{j_2} w_{j_2}^{(t)} \right\rangle$ is always increasing. Thus, $\frac{\partial G^{(t)}(\mu)}{\partial t}$ mainly depends on the behavior of $\frac{1}{n} \sum_{i_2: \, \mu \in X^{(i_2)}} g_{i_2}^{(t)} y_{i_2} \sum_{l_2=1}^{L} p_{q \leftarrow l_2, k \leftarrow \mu}^{(t,i_2)}$. Further, this is also a key quantity we need to analyze $\frac{\partial \nu^\top W_V^{(t)\top} W_V^{(t)} \mu}{\partial t}$ and $\frac{\partial \nu^\top W_K^{(t)\top} W_Q^{(t)} \mu}{\partial t}$. Note that $\frac{1}{n} \sum_{i_2: \, \mu \in X^{(i_2)}} g_{i_2}^{(t)} y_{i_2} \sum_{l_2=1}^{L} p_{q \leftarrow l_2, k \leftarrow \mu}^{(t,i_2)} \approx \frac{1}{n} \sum_{i_2: \, \mu \in X^{(i_2)}} g_{i_2}^{(t)} y_{i_2}$ if $p_{q \leftarrow l_2, k \leftarrow \mu}^{(t,i_2)} \approx 1/L$ which holds at the beginning of Phase 2. Later, we are going to prove convergence of the training loss via the following: (1) the training loss can decrease if the softmax probability is uniform; (2) even though the softmax probability will deviate from uniform distribution during training, we can bound such deviation and the loss value can still decrease.

**Automatic balancing of gradients.** As argued above, our main focus is on analyzing the behavior of $\frac{1}{n} \sum_{i_2: \, \mu \in X^{(i_2)}} g_{i_2}^{(t)} y_{i_2}$. This consists of two parts: (i) Lemma 4.4, which shows that the two groups of samples with only the presence of one target signal have gradients $\sum_{i \in I_2} g_i^{(t)}$ and $\sum_{i \in I_3} g_i^{(t)}$ close to each other during training; and (ii) Lemma 4.5, which shows that the gradient gaps $\sum_{i \in I_1} g_i^{(t)} - \sum_{i \in I_2} g_i^{(t)}$ and $\sum_{i \in I_2 \cup I_3 \cup I_4} g_i^{(t)} - \sum_{i \in I_1} g_i^{(t)}$ are not too small compared with $\sum_{i \in [n]} g_i^{(t)}$. Both

Lemmas 4.4 and 4.5 establish that the ratio of those important gradients are kept within certain ranges during training. We call such a key property as *automatic balancing of gradients*, which is further used for proving that the gradient flow can drive the training loss small.

**Lemma 4.4** (Same as Lemma F.5). *For $t \in [T_1, T_2]$, there exists a small constant $C \ll 1$ such that*

$$\frac{\left| \sum_{i \in I_2} g_i^{(t)} - \sum_{i \in I_3} g_i^{(t)} \right|}{\min(\sum_{i \in I_2} g_i^{(t)}, \sum_{i \in I_3} g_i^{(t)})} \leq C.$$

*Proof Intuition of Lemma 4.4.* The intuition behind the result is as follows. If $\sum_{i \in I_2} g_i^{(t)}$ becomes much bigger than $\sum_{i \in I_3} g_i^{(t)}$ during the training, then $\sum_{i \in I_1} g_i^{(t)} - \sum_{i \in I_2} g_i^{(t)}$ is much smaller than $\sum_{i \in I_1} g_i^{(t)} - \sum_{i \in I_3} g_i^{(t)}$ which makes $\frac{\partial G^{(t)}(\mu_1)}{\partial t} < \frac{\partial G^{(t)}(\mu_2)}{\partial t}$. It is not hard to show that random tokens make negligible contributions to the gradient. Thus, for $i \in I_2$, we have $\frac{\partial F_i^{(t)}}{\partial t} \approx \frac{\partial G^{(t)}(\mu_1)}{\partial t} + \frac{\partial G^{(t)}(\mu_3)}{\partial t}$. By the chain rule, we have $\frac{\partial g_i^{(t)}}{\partial t} = g'(y_i F_i^{(t)}) y_i \frac{\partial F_i^{(t)}}{\partial t}$. Since $\frac{\partial G^{(t)}(\mu_3)}{\partial t} < 0$, if $\frac{\partial G^{(t)}(\mu_1)}{\partial t} < \frac{\partial G^{(t)}(\mu_2)}{\partial t}$, then $\sum_{i \in I_2} g_i^{(t)}$ will drop faster than $\sum_{i \in I_3} g_i^{(t)}$, i.e., $-\frac{\partial}{\partial t} \sum_{i \in I_2} g_i^{(t)} > -\frac{\partial}{\partial t} \sum_{i \in I_3} g_i^{(t)}$. In Appendix, we formally prove Lemma 4.4 by analyzing the ratio $\sum_{i \in I_2} g_i^{(t)} / \sum_{i \in I_3} g_i^{(t)}$, and show that this ratio hangs over around 1 during training. □

**Lemma 4.5** (Abbreviated from Lemma F.6). *For $t \in [T_1, T_2]$, the gradient satisfies that*

$$\frac{\sum_{i \in [n]} g_i^{(t)}}{\sum_{i \in I_2 \cup I_3 \cup I_4} g_i^{(t)} - \sum_{i \in I_1} g_i^{(t)}} = O(1), \qquad \frac{\sum_{i \in [n]} g_i^{(t)}}{\sum_{i \in I_1} g_i^{(t)} - \sum_{i \in I_2} g_i^{(t)}} = O(1).$$

*Further, for some constant $C$, we have*

$$(1 + C) \max \left( \frac{\partial G^{(t)}(\mu_1)}{\partial t}, \frac{\partial G^{(t)}(\mu_2)}{\partial t} \right) \leq -\frac{\partial G^{(t)}(\mu_3)}{\partial t} \leq (1 - C) \left( \frac{\partial G^{(t)}(\mu_1)}{\partial t} + \frac{\partial G^{(t)}(\mu_2)}{\partial t} \right).$$

*Proof Sketch of Lemma 4.5.* The proof of Lemma 4.5 relies on analyzing how the ratio between $\frac{\partial G^{(t)}(\mu_1)}{\partial t}$ and $-\frac{\partial G^{(t)}(\mu_3)}{\partial t}$ changes. We show that this ratio will hang over around some range. Recall the relationship that $\frac{\partial G^{(t)}(\mu)}{\partial t} \approx \frac{1}{n} \sum_{i_2: \mu \in X^{(i_2)}} g_{i_2}^{(t)} y_{i_2} \cdot \sum_{j_1=1}^{m_1} \sum_{j_2=1}^{m_1} a_{j_1} a_{j_2} \left\langle w_{j_1}^{(t)}, w_{j_2}^{(t)} \right\rangle$. It is not hard to show that

$$\frac{-\frac{\partial G^{(t)}(\mu_3)}{\partial t}}{\frac{\partial G^{(t)}(\mu_1)}{\partial t}} \approx \frac{-\sum_{i \in I_1} g_i^{(t)} + \sum_{i \in I_2 \cup I_3 \cup I_4} g_i^{(t)}}{\sum_{i \in I_1} g_i^{(t)} - \sum_{i \in I_2} g_i^{(t)}}.$$

Define $R(t) := \frac{-\sum_{i \in I_1} g_i^{(t)} + \sum_{i \in I_2 \cup I_3 \cup I_4} g_i^{(t)}}{\sum_{i \in I_1} g_i^{(t)} - \sum_{i \in I_2} g_i^{(t)}}$. Solving when $\frac{\partial}{\partial t} R(t) \geq 0$ yields a quadratic inequality, and further analysis shows that the root is contractive and is within some specific range. □

Utilizing the *gradient automatic balancing* properties, the following corollary characterizes how the attention matrices in the dynamical system change in Phase 2. In particular, we can show that after $W_V$-transform, $\mu_1$ and $\mu_2$ become more positively correlated whereas $\mu_1$ and $\mu_3$ (also $\mu_2$ and $\mu_3$) become negatively correlated. This is a direct result following from updates of the dynamical system.

**Corollary 4.6** (Abbreviated from Corollary F.13). *For $t \in [T_1, T_2]$,*

$$\frac{\partial}{\partial t} \mu_2^\top W_V^{(t)\top} W_V^{(t)} \mu_1 > 0, \qquad \frac{\partial}{\partial t} \mu_1^\top W_V^{(t)\top} W_V^{(t)} \mu_3 < 0.$$

Since we have analyzed how $G^{(t)}(\cdot)$ will change in stage 2, we can utilize this information to analyze the change of softmax attention via the following relationship: by Appendix C, we can derive

$$\frac{\partial \mu_1^\top W_K^{(t)\top} W_Q^{(t)} \mu_2}{\partial t}$$

$$= \frac{1}{n\sqrt{m}} \sum_{i=1}^{n} g_i^{(t)} y_i \sum_{l=1}^{L} \mu_1^\top W_K^{(t)\top} K^{(t,i)} \cdot \mathrm{diag}\left(G^{(t)}(X^{(i)}) - (G^{(t)}(X^{(i)}))^\top p_l^{(t,i)}\right) p_l^{(t,i)} x_l^{(i)\top} \mu_2$$

$$+ \frac{1}{n\sqrt{m}} \sum_{i=1}^{n} g_i^{(t)} y_i \sum_{l=1}^{L} \mu_2^\top W_Q^{(t)\top} q_l^{(t,i)} p_l^{(t,i)\top} \cdot \mathrm{diag}\left(G^{(t)}(X^{(i)}) - (G^{(t)}(X^{(i)}))^\top p_l^{(t,i)}\right) X^{(i)\top} \mu_1.$$

The following lemma shows that the attention score between the two target signals $\mu_1$ and $\mu_2$ increases, whereas that between one target signal $\mu_1$ or $\mu_2$ and the common token $\mu_3$ decreases.

**Lemma 4.7** (Abbreviated from Lemma F.16). *For $\mu, \nu \in \{\mu_1, \mu_2\}$, $\mu \neq \nu$, and for $t \in [T_1, T_2]$,*

$$\frac{\partial}{\partial t} \nu^\top W_K^{(t)\top} W_Q^{(t)} \mu = \frac{1}{\sqrt{m}} \widetilde{\Theta}(\widehat{L}^{(t)} \sigma_0^2 m) \frac{1}{L}, \qquad \frac{\partial}{\partial t} \mu_3^\top W_K^{(t)\top} W_Q^{(t)} \mu = -\frac{1}{\sqrt{m}} \widetilde{\Theta}(\widehat{L}^{(t)} \sigma_0^2 m) \frac{1}{L}.$$

Lemma 4.7 is keeping track of the attention coefficients $C_{i_1,i_2}^{(t)}$ in Theorem 3.3 via gradient flow, which proves the second item of Theorem 3.3.

## 5 Discussion and Future Directions

In this work, we developed a novel gradient flow based framework for analyzing the training dynamics of a one-layer transformer to recognize co-occurring tokens. We provided a two-phase characterization of the training process. In Phase 1, the linear MLP layer is trained to classify samples correctly, with attention weights almost unchanged. In Phase 2, both attention matrices and the linear MLP jointly evolve to enlarge the classification margin, thus reducing the loss to near minimum.

As future work, it will be interesting to analyze more general transformer architectures such as multi-headed attention, multi-layer transformer, etc. Further, it is of interest to study the dynamics of more advanced gradient descent algorithms such as gradient descent with adaptive learning rate, with momentum, etc., and explore how the hyperparameters will affect the training dynamics. Another direction is to study more practical language sequences where tokens are generated in a correlated fashion. Then the next token prediction becomes an intriguing problem.

## Acknowledgement

H. Yang would like to thank Jason D. Lee and Yunwei Ren for insightful discussion and suggestions.

This work was performed under the auspices of the U.S. Department of Energy by the Lawrence Livermore National Laboratory under Contract No. DE-AC52-07NA27344 and supported by the LLNL-LDRD Program under Project No. 22-SI-004 and 24-ERD-010. The work of Y. Liang was supported in part by the U.S. National Science Foundation under the grants ECCS-2113860 and DMS-2134145. The work of Z. Wang was in part supported by an NSF Scale-MoDL grant (award number: 2133861) and the CAREER Award (award number: 2145346).

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

# Contents

## A  Setup of Synthetic Experiment

We conduct synthetic experiment to verify our theoretical results. We create a dataset following our data distribution in Definition 2.1 with 60 training samples: 30 samples have both $\mu_1$ and $\mu_2$ in it, 10 samples have only $\mu_1$, 10 samples have only $\mu_2$, and 10 samples have neither $\mu_1$ nor $\mu_2$. Each data consists of 5 patches and each patch has dimension 64. The embedding dimension $m$ is set to be 128 and the number of neurons is set to be 256. We use Kaiming initialization to initialize the transformer weights. The transformer is trained by gradient descent with learning rate 0.01 for 30000 epochs.

## B  Gradient Flow Update for Weight Matrices

We provide the gradient flow update for each weight matrix as follows.

$$\frac{\partial w_j^{(t)}}{\partial t} = \frac{1}{n} \sum_{i=1}^{n} g_i^{(t)} y_i \sum_{l=1}^{L} a_j V^{(t,i)} p_l^{(t,i)} = \frac{1}{n} \sum_{i=1}^{n} g_i^{(t)} y_i \sum_{l=1}^{L} a_j \sum_{h=1}^{L} v_h^{(t,i)} p_{l,h}^{(t,i)}$$

$$\frac{\partial W_V^{(t)}}{\partial t} = \frac{1}{n} \sum_{i=1}^{n} g_i^{(t)} y_i \sum_{l=1}^{L} \sum_{j=1}^{m_1} a_j w_j^{(t)} \left( X^{(i)} p_l^{(t,i)} \right)^{\top}$$

$$\frac{\partial W_K^{(t)}}{\partial t} = \frac{1}{n\sqrt{m}} \sum_{i=1}^{n} g_i^{(t)} y_i \sum_{l=1}^{L} \sum_{j=1}^{m_1} a_j \sum_{h=1}^{L} w_j^{(t)\top} v_h^{(t,i)} \sum_{h'=1}^{L} \frac{\partial p_{l,h}^{(t,i)}}{\partial s_{l,h'}} q_l^{(t,i)} x_{h'}^{(i)\top}$$

$$= \frac{1}{n\sqrt{m}} \sum_{i=1}^{n} g_i^{(t)} y_i \sum_{l=1}^{L} \sum_{j=1}^{m_1} a_j \sum_{h=1}^{L} w_j^{(t)\top} v_h^{(t,i)} \sum_{h'=1}^{L} p_{l,h}^{(t,i)} (\mathbb{I}(h = h') - p_{l,h'}^{(t,i)}) q_l^{(t,i)} x_{h'}^{(i)\top}$$

$$= \frac{1}{n\sqrt{m}} \sum_{i=1}^{n} g_i^{(t)} y_i \sum_{l=1}^{L} \sum_{j=1}^{m_1} a_j \left( -w_j^{(t)\top} V^{(t,i)} p_l^{(t,i)} q_l^{(t,i)} (X^{(i)} p_l^{(t,i)})^{\top} + q_l^{(t,i)} w_j^{(t)\top} V^{(t,i)} \mathrm{diag}(p_l^{(t,i)}) X^{(i)\top} \right)$$

$$= \frac{1}{n\sqrt{m}} \sum_{i=1}^{n} g_i^{(t)} y_i \sum_{l=1}^{L} \sum_{j=1}^{m_1} a_j q_l^{(t,i)} \left( -w_j^{(t)\top} V^{(t,i)} p_l^{(t,i)} p_l^{(t,i)\top} + w_j^{(t)\top} V^{(t,i)} \mathrm{diag}(p_l^{(t,i)}) \right) X^{(i)\top}$$

$$= \frac{1}{n\sqrt{m}} \sum_{i=1}^{n} g_i^{(t)} y_i \sum_{l=1}^{L} \sum_{j=1}^{m_1} a_j q_l^{(t,i)} p_l^{(t,i)\top} \mathrm{diag} \left( w_j^{(t)\top} V^{(t,i)} - w_j^{(t)\top} V^{(t,i)} p_l^{(t,i)} \right) X^{(i)\top}$$

$$\frac{\partial W_Q^{(t)}}{\partial t} = \frac{1}{n\sqrt{m}} \sum_{i=1}^{n} g_i^{(t)} y_i \sum_{l=1}^{L} \sum_{j=1}^{m_1} a_j \sum_{h=1}^{L} w_j^{(t)\top} v_h^{(t,i)} \sum_{h'=1}^{L} p_{l,h}^{(t,i)} (\mathbb{I}(h = h') - p_{l,h'}^{(t,i)}) k_{h'}^{(t,i)} x_l^{(i)\top}$$

$$= \frac{1}{n\sqrt{m}} \sum_{i=1}^{n} g_i^{(t)} y_i \sum_{l=1}^{L} \sum_{j=1}^{m_1} a_j \left( -w_j^{(t)\top} V^{(t,i)} p_l^{(t,i)} (K^{(t,i)} p_l^{(t,i)}) + K^{(t,i)} \mathrm{diag}(p_l^{(t,i)}) V^{(t,i)\top} w_j^{(t)} \right) x_l^{(i)\top}$$

$$= \frac{1}{n\sqrt{m}} \sum_{i=1}^{n} g_i^{(t)} y_i \sum_{l=1}^{L} \sum_{j=1}^{m_1} a_j K^{(t,i)} \left( -w_j^{(t)\top} V^{(t,i)} p_l^{(t,i)} p_l^{(t,i)} + \mathrm{diag}(p_l^{(t,i)}) V^{(t,i)\top} w_j^{(t)} \right) x_l^{(i)\top}$$

$$= \frac{1}{n\sqrt{m}} \sum_{i=1}^{n} g_i^{(t)} y_i \sum_{l=1}^{L} \sum_{j=1}^{m_1} a_j K^{(t,i)} \mathrm{diag} \left( V^{(t,i)\top} w_j^{(t)} - w_j^{(t)\top} V^{(t,i)} p_l^{(t,i)} \right) p_l^{(t,i)} x_l^{(i)\top}$$

## C  Gradient Flow Dynamical System

We first provide our complete dynamical system. The derivation of each equation is provided in Appendix C.1.

$$\frac{\partial w_{j_1}^{(t)\top} W_V^{(t)} \mu}{\partial t}$$

$$= \frac{1}{n} \sum_{i_2=1}^{n} g_{i_2}^{(t)} y_{i_2} \sum_{l_2=1}^{L} \sum_{j_2=1}^{m_1} a_{j_2} \left\langle w_{j_1}^{(t)}, w_{j_2}^{(t)} \right\rangle \left( X^{(i_2)} p_{l_2}^{(t,i_2)} \right)^{\top} \mu + \frac{1}{n} \sum_{i_1=1}^{n} g_{i_1}^{(t)} y_{i_1} \sum_{l_1=1}^{L} a_{j_1} p_{l_1}^{(t,i_1)\top} V^{(t,i_1)\top} W_V^{(t)} \mu$$

$$\tag{6}$$

$$\frac{\partial \nu^{\top} W_K^{(t)\top} W_Q^{(t)} \mu}{\partial t}$$

$$= \frac{1}{n\sqrt{m}} \sum_{i=1}^{n} g_i^{(t)} y_i \sum_{l=1}^{L} \sum_{j=1}^{m_1} a_j \nu^{\top} W_K^{(t)\top} K^{(t,i)} \cdot \text{diag} \left( V^{(t,i)\top} w_j^{(t)} - w_j^{(t)\top} V^{(t,i)} p_l^{(t,i)} \right) p_l^{(t,i)} x_l^{(i)\top} \mu$$

$$+ \frac{1}{n\sqrt{m}} \sum_{i=1}^{n} g_i^{(t)} y_i \sum_{l=1}^{L} \sum_{j=1}^{m_1} a_j \mu^{\top} W_Q^{(t)\top} q_l^{(t,i)} p_l^{(t,i)\top} \cdot \text{diag} \left( w_j^{(t)\top} V^{(t,i)} - w_j^{(t)\top} V^{(t,i)} p_l^{(t,i)} \right) X^{(i)\top} \nu$$

$$\tag{7}$$

$$\frac{\partial \left\langle w_{j_1}^{(t)}, w_{j_2}^{(t)} \right\rangle}{\partial t} = \frac{1}{n} \sum_{i=1}^{n} g_i^{(t)} y_i \sum_{l=1}^{L} a_{j_2} w_{j_1}^{(t)\top} V^{(t,i)} p_l^{(t,i)} + \frac{1}{n} \sum_{i=1}^{n} g_i^{(t)} y_i \sum_{l=1}^{L} a_{j_1} w_{j_2}^{(t)\top} V^{(t,i)} p_l^{(t,i)}$$

$$\frac{\partial \nu^{\top} W_V^{(t)\top} W_V^{(t)} \mu}{\partial t}$$

$$= \frac{1}{n} \sum_{i:\mu \in X^{(i)}} g_i^{(t)} y_i \sum_{l=1}^{L} \sum_{j=1}^{m_1} a_j \nu^{\top} W_V^{(t)\top} w_j^{(t)} p_{q \leftarrow l, k \leftarrow \mu}^{(t,i)} + \frac{1}{n} \sum_{i:\nu \in X^{(i)}} g_i^{(t)} y_i \sum_{l=1}^{L} \sum_{j=1}^{m_1} a_j \mu^{\top} W_V^{(t)\top} w_j^{(t)} p_{q \leftarrow l, k \leftarrow \nu}^{(t,i)}$$

$$\frac{\partial \nu^{\top} W_Q^{(t)\top} W_Q^{(t)} \mu}{\partial t}$$

$$= \frac{1}{n\sqrt{m}} \sum_{i:\mu \in X^{(i)}} g_i^{(t)} y_i \sum_{j=1}^{m_1} a_j \nu^{\top} W_Q^{(t)\top} K^{(t,i)} \cdot \text{diag} \left( V^{(t,i)\top} w_j^{(t)} - w_j^{(t)\top} V^{(t,i)} p_{l(i,\mu)}^{(t,i)} \right) p_{l(i,\mu)}^{(t,i)}$$

$$+ \frac{1}{n\sqrt{m}} \sum_{i:\nu \in X^{(i)}} g_i^{(t)} y_i \sum_{j=1}^{m_1} a_j \mu^{\top} W_Q^{(t)\top} K^{(t,i)} \cdot \text{diag} \left( V^{(t,i)\top} w_j^{(t)} - w_j^{(t)\top} V^{(t,i)} p_{l(i,\nu)}^{(t,i)} \right) p_{l(i,\nu)}^{(t,i)}$$

$$\frac{\partial \nu^{\top} W_K^{(t)\top} W_K^{(t)} \mu}{\partial t}$$

$$= \frac{1}{n\sqrt{m}} \sum_{i:\mu \in X^{(i)}} g_i^{(t)} y_i \sum_{l=1}^{L} \sum_{j=1}^{m_1} a_j \nu^{\top} W_K^{(t)\top} q_l^{(t,i)} p_{q \leftarrow l, k \leftarrow \mu}^{(t,i)} \cdot \left( w_j^{(t)\top} v^{(t,i)}(\mu) - w_j^{(t)\top} V^{(t,i)} p_l^{(t,i)} \right)$$

$$+ \frac{1}{n\sqrt{m}} \sum_{i:\nu \in X^{(i)}} g_i^{(t)} y_i \sum_{l=1}^{L} \sum_{j=1}^{m_1} a_j \mu^{\top} W_K^{(t)\top} q_l^{(t,i)} p_{q \leftarrow l, k \leftarrow \nu}^{(t,i)} \cdot \left( w_j^{(t)\top} v^{(t,i)}(\nu) - w_j^{(t)\top} V^{(t,i)} p_l^{(t,i)} \right)$$

## C.1 Derivation of the Dynamical System

**Lemma C.1.** *Let* $\mu \in \{\mu_i\}_{i=1}^{d}$. *For all* $j \in [m]$, *we have*

$$\frac{\partial w_{j_1}^{(t)\top} W_V^{(t)} \mu}{\partial t} = \frac{1}{n} \sum_{i_2=1}^{n} g_{i_2}^{(t)} y_{i_2} \sum_{l_2=1}^{L} \sum_{j_2=1}^{m_1} a_{j_2} \left\langle w_{j_1}^{(t)}, w_{j_2}^{(t)} \right\rangle \left( X^{(i_2)} p_{l_2}^{(t,i_2)} \right)^{\top} \mu$$

$$+ \frac{1}{n} \sum_{i_1=1}^{n} g_{i_1}^{(t)} y_{i_1} \sum_{l_1=1}^{L} a_{j_1} p_{l_1}^{(t,i_1)\top} V^{(t,i_1)\top} W_V^{(t)} \mu$$

$$= \frac{1}{n} \sum_{i:\mu \in X^{(i)}} g_i^{(t)} y_i \sum_{l=1}^{L} a_j \left( \|w_j^{(t)}\|_2^2 + \|v^{(t)}(\mu)\|_2^2 \right) p_{q \leftarrow l, k \leftarrow \mu}^{(t,i)} + \varepsilon,$$

*where*

$$\varepsilon = \frac{1}{n} \sum_{i:\mu\in X^{(i)}} g_i^{(t)} y_i \sum_{l=1}^{L} \sum_{j_2\neq j_1} a_{j_2} \left\langle w_{j_1}^{(t)}, w_{j_2}^{(t)} \right\rangle p_{q\leftarrow l, k\leftarrow\mu}^{(t,i)}$$

$$+ \frac{1}{n} \sum_{i=1}^{n} g_i^{(t)} y_i \sum_{l_1=1}^{L} a_{j_1} \sum_{l_2=1}^{L} \left\langle p_{l_1,l_2}^{(t,i)} v_{l_2}^{(t,i)}, v^{(t)}(\mu) \right\rangle \mathbb{I}(v_{l_2}^{(t,i)} \neq v^{(t)}(\mu)).$$

*Proof.* Let $l(i,\mu)$ denote the index such that $X_{l(i,\mu)}^{(i)} = \mu$. By the gradient flow update, we have

$$\frac{\partial w_{j_1}^{(t)\top} W_V^{(t)} \mu}{\partial t} = \frac{1}{n} \sum_{i_2=1}^{n} g_{i_2}^{(t)} y_{i_2} \sum_{l_2=1}^{L} \sum_{j_2=1}^{m_1} a_{j_2} \left\langle w_{j_1}^{(t)}, w_{j_2}^{(t)} \right\rangle \left( X^{(i_2)} p_{l_2}^{(t,i_2)} \right)^{\top} \mu$$

$$+ \frac{1}{n} \sum_{i_1=1}^{n} g_{i_1}^{(t)} y_{i_1} \sum_{l_1=1}^{L} a_{j_1} p_{l_1}^{(t,i_1)\top} V^{(t,i_1)\top} W_V^{(t)} \mu$$

$$= \frac{1}{n} \sum_{i_2:\, \mu\in X^{(i_2)}} g_{i_2}^{(t)} y_{i_2} \sum_{l_2=1}^{L} \sum_{j_2=1}^{m_1} a_{j_2} \left\langle w_{j_1}^{(t)}, w_{j_2}^{(t)} \right\rangle p_{q\leftarrow l_2, k\leftarrow\mu}^{(t,i_2)}$$

$$+ \frac{1}{n} \sum_{i_1=1}^{n} g_{i_1}^{(t)} y_{i_1} \sum_{l_1=1}^{L} a_{j_1} \sum_{l_2=1}^{L} \left\langle p_{l_1,l_2}^{(t,i_1)} v_{l_2}^{(t,i_1)}, v^{(t)}(\mu) \right\rangle$$

$$= \frac{1}{n} \sum_{i_2:\, \mu\in X^{(i_2)}} g_{i_2}^{(t)} y_{i_2} \sum_{l_2=1}^{L} \left( a_{j_1} \|w_{j_1}^{(t)}\|_2^2 + \sum_{j_2\neq j_1} a_{j_2} \left\langle w_{j_1}^{(t)}, w_{j_2}^{(t)} \right\rangle \right) p_{q\leftarrow l_2, k\leftarrow\mu}^{(t,i_2)}$$

$$+ \frac{1}{n} \sum_{i_1:\mu\notin X^{(i_1)}} g_{i_1}^{(t)} y_{i_1} \sum_{l_1=1}^{L} a_{j_1} \sum_{l_2=1}^{L} \left\langle p_{l_1,l_2}^{(t,i_1)} v_{l_2}^{(t,i_1)}, v^{(t)}(\mu) \right\rangle$$

$$+ \frac{1}{n} \sum_{i_1:\mu\in X^{(i_1)}} g_{i_1}^{(t)} y_{i_1} \sum_{l_1=1}^{L} a_{j_1} \left( \|v^{(t)}(\mu)\|_2^2 p_{q\leftarrow l_1, k\leftarrow\mu}^{(t,i_1)} + \sum_{l_2\neq l(i_1,\mu)} \left\langle p_{l_1,l_2}^{(t,i_1)} v_{l_2}^{(t,i_1)}, v^{(t)}(\mu) \right\rangle \right)$$

$$= \frac{1}{n} \sum_{i:\mu\in X^{(i)}} g_i^{(t)} y_i \sum_{l=1}^{L} a_{j_1} \left( \|w_{j_1}^{(t)}\|_2^2 + \|v^{(t)}(\mu)\|_2^2 \right) p_{q\leftarrow l, k\leftarrow\mu}^{(t,i)}$$

$$+ \frac{1}{n} \sum_{i:\mu\in X^{(i)}} g_i^{(t)} y_i \sum_{l=1}^{L} \sum_{j_2\neq j_1} a_{j_2} \left\langle w_{j_1}^{(t)}, w_{j_2}^{(t)} \right\rangle p_{q\leftarrow l, k\leftarrow\mu}^{(t,i)}$$

$$+ \frac{1}{n} \sum_{i=1}^{n} g_i^{(t)} y_i \sum_{l_1=1}^{L} a_{j_1} \sum_{l_2=1}^{L} \left\langle p_{l_1,l_2}^{(t,i)} v_{l_2}^{(t,i)}, v^{(t)}(\mu) \right\rangle \mathbb{I}(v_{l_2}^{(t,i)} \neq v^{(t)}(\mu)).$$

$\square$

**Lemma C.2.** *The following equation on the gradient flow holds:*

$$\frac{\partial \left\langle w_{j_1}^{(t)}, w_{j_2}^{(t)} \right\rangle}{\partial t} = \frac{1}{n} \sum_{i=1}^{n} g_i^{(t)} y_i \sum_{l=1}^{L} \left( \sum_{l':X_{l'}^{(i)}\in\mathcal{U}} a_{j_2} w_{j_1}^{(t)\top} V_{l'}^{(t,i)} p_{l,l'}^{(t,i)} + a_{j_1} w_{j_2}^{(t)\top} V_{l'}^{(t,i)} p_{l,l'}^{(t,i)} \right)$$

$$+ \frac{1}{n} \sum_{i=1}^{n} g_i^{(t)} y_i \sum_{l=1}^{L} \left( \sum_{l':X_{l'}^{(i)}\notin\mathcal{U}} a_{j_2} w_{j_1}^{(t)\top} V_{l'}^{(t,i)} p_{l,l'}^{(t,i)} + a_{j_1} w_{j_2}^{(t)\top} V_{l'}^{(t,i)} p_{l,l'}^{(t,i)} \right).$$

*Proof.* By gradient flow update, we have

$$\frac{\partial \left\langle w_{j_1}^{(t)}, w_{j_2}^{(t)} \right\rangle}{\partial t} = \frac{1}{n} \sum_{i=1}^{n} g_i^{(t)} y_i \sum_{l=1}^{L} a_{j_2} w_{j_1}^{(t)\top} V^{(t,i)} p_l^{(t,i)} + \frac{1}{n} \sum_{i=1}^{n} g_i^{(t)} y_i \sum_{l=1}^{L} a_{j_1} w_{j_2}^{(t)\top} V^{(t,i)} p_l^{(t,i)}$$

$$= \frac{1}{n} \sum_{i=1}^{n} g_i^{(t)} y_i \sum_{l=1}^{L} \left( \sum_{l':X_{l'}^{(i)} \in \mathcal{U}} a_{j_2} w_{j_1}^{(t)\top} V_{l'}^{(t,i)} p_{l,l'}^{(t,i)} + a_{j_1} w_{j_2}^{(t)\top} V_{l'}^{(t,i)} p_{l,l'}^{(t,i)} \right)$$

$$+ \frac{1}{n} \sum_{i=1}^{n} g_i^{(t)} y_i \sum_{l=1}^{L} \left( \sum_{l':X_{l'}^{(i)} \notin \mathcal{U}} a_{j_2} w_{j_1}^{(t)\top} V_{l'}^{(t,i)} p_{l,l'}^{(t,i)} + a_{j_1} w_{j_2}^{(t)\top} V_{l'}^{(t,i)} p_{l,l'}^{(t,i)} \right).$$

$\square$

**Lemma C.3.** *Let $\mu, \nu \in \{\mu_i\}_{i=1}^{d}$. Then the following equation on gradient flow holds:*

$$\frac{\partial \nu W_V^{(t)\top} W_V^{(t)} \mu}{\partial t} = \frac{1}{n} \sum_{i:\mu \in X^{(i)}} g_i^{(t)} y_i \sum_{l=1}^{L} \sum_{j=1}^{m_1} a_j \nu^\top W_V^{(t)\top} w_j^{(t)} p_{q \leftarrow l, k \leftarrow \mu}^{(t,i)}$$

$$+ \frac{1}{n} \sum_{i:\nu \in X^{(i)}} g_i^{(t)} y_i \sum_{l=1}^{L} \sum_{j=1}^{m_1} a_j \mu^\top W_V^{(t)\top} w_j^{(t)} p_{q \leftarrow l, k \leftarrow \nu}^{(t,i)}.$$

*Proof.* The gradient flow update can be derived as follows:

$$\frac{\partial \nu W_V^{(t)\top} W_V^{(t)} \mu}{\partial t} = \frac{1}{n} \sum_{i=1}^{n} g_i^{(t)} y_i \sum_{l=1}^{L} \sum_{j=1}^{m_1} a_j \nu^\top W_V^{(t)\top} w_j^{(t)} \left( X^{(i)} p_l^{(t,i)} \right)^\top \mu$$

$$+ \frac{1}{n} \sum_{i=1}^{n} g_i^{(t)} y_i \sum_{l=1}^{L} \sum_{j=1}^{m_1} a_j \mu^\top W_V^{(t)\top} w_j^{(t)} \left( X^{(i)} p_l^{(t,i)} \right)^\top \nu$$

$$= \frac{1}{n} \sum_{i:\mu \in X^{(i)}} g_i^{(t)} y_i \sum_{l=1}^{L} \sum_{j=1}^{m_1} a_j \nu^\top W_V^{(t)\top} w_j^{(t)} p_{q \leftarrow l, k \leftarrow \mu}^{(t,i)}$$

$$+ \frac{1}{n} \sum_{i:\nu \in X^{(i)}} g_i^{(t)} y_i \sum_{l=1}^{L} \sum_{j=1}^{m_1} a_j \mu^\top W_V^{(t)\top} w_j^{(t)} p_{q \leftarrow l, k \leftarrow \nu}^{(t,i)}.$$

$\square$

**Lemma C.4.** *Let $\mu, \nu \in \{\mu_i\}_{i=1}^{d}$. Then the following equations on gradient flow hold.*

$$\frac{\partial \nu^\top W_Q^{(t)\top} W_Q^{(t)} \mu}{\partial t}$$

$$= \frac{1}{n\sqrt{m}} \sum_{i:\mu \in X^{(i)}} g_i^{(t)} y_i \sum_{j=1}^{m_1} a_j \nu^\top W_Q^{(t)\top} K^{(t,i)} \text{diag}\left( V^{(t,i)\top} w_j^{(t)} - w_j^{(t)\top} V^{(t,i)} p_{l(i,\mu)}^{(t,i)} \right) p_{l(i,\mu)}^{(t,i)}$$

$$+ \frac{1}{n\sqrt{m}} \sum_{i:\nu \in X^{(i)}} g_i^{(t)} y_i \sum_{j=1}^{m_1} a_j \mu^\top W_Q^{(t)\top} K^{(t,i)} \text{diag}\left( V^{(t,i)\top} w_j^{(t)} - w_j^{(t)\top} V^{(t,i)} p_{l(i,\nu)}^{(t,i)} \right) p_{l(i,\nu)}^{(t,i)},$$

$$\frac{\partial \nu^\top W_K^{(t)\top} W_K^{(t)} \mu}{\partial t}$$

$$= \frac{1}{n\sqrt{m}} \sum_{i:\mu \in X^{(i)}} g_i^{(t)} y_i \sum_{l=1}^{L} \sum_{j=1}^{m_1} a_j \nu^\top W_K^{(t)\top} q_l^{(t,i)} p_{q \leftarrow l, k \leftarrow \mu}^{(t,i)} \left( w_j^{(t)\top} v^{(t,i)}(\mu) - w_j^{(t)\top} V^{(t,i)} p_l^{(t,i)} \right)$$

$$+ \frac{1}{n\sqrt{m}} \sum_{i:\nu \in X^{(i)}} g_i^{(t)} y_i \sum_{l=1}^{L} \sum_{j=1}^{m_1} a_j \mu^{\top} W_K^{(t)\top} q_l^{(t,i)} p_{q\leftarrow l, k\leftarrow \nu}^{(t,i)} \left( w_j^{(t)\top} v^{(t,i)}(\nu) - w_j^{(t)\top} V^{(t,i)} p_l^{(t,i)} \right),$$

$$\frac{\partial \nu^{\top} W_K^{(t)\top} W_Q^{(t)} \mu}{\partial t}$$

$$= \frac{1}{n\sqrt{m}} \sum_{i:\mu,\nu \in X^{(i)}} g_i^{(t)} y_i \sum_{j=1}^{m_1} a_j \|k^{(t)}(\nu)\|_2^2 \left( v^{(t)\top}(\nu) w_j^{(t)} - w_j^{(t)\top} V^{(t,i)} p_{l(i,\mu)}^{(t,i)} \right) p_{q\leftarrow \mu, k\leftarrow \nu}^{(t,i)}$$

$$+ \frac{1}{n\sqrt{m}} \sum_{i:\mu \in X^{(i)}} g_i^{(t)} y_i \sum_{j=1}^{m_1} a_j \sum_{l=1}^{L} \nu^{\top} W_K^{(t)\top} K_l^{(t,i)} \left( V_l^{(t,i)\top} w_j^{(t)} - w_j^{(t)\top} V^{(t,i)} p_{l(i,\mu)}^{(t,i)} \right) p_{q\leftarrow \mu, k\leftarrow l}^{(t,i)} \mathbb{I}(K_l^{(t,i)} \neq k^{(t)}(\nu))$$

$$+ \frac{1}{n\sqrt{m}} \sum_{i:\nu,\mu \in X^{(i)}} g_i^{(t)} y_i \sum_{j=1}^{m_1} a_j \|q^{(t)}(\mu)\|_2^2 p_{q\leftarrow \mu, k\leftarrow \nu}^{(t,i)} \left( w_j^{(t)\top} v^{(t,i)}(\nu) - w_j^{(t)\top} V^{(t,i)} p_{l(i,\mu)}^{(t,i)} \right)$$

$$+ \frac{1}{n\sqrt{m}} \sum_{i:\nu \in X^{(i)}} g_i^{(t)} y_i \sum_{l=1}^{L} \sum_{j=1}^{m_1} a_j \mu^{\top} W_Q^{(t)\top} q_l^{(t,i)} p_{q\leftarrow l, k\leftarrow \nu}^{(t,i)} \left( w_j^{(t)\top} v^{(t,i)}(\nu) - w_j^{(t)\top} V^{(t,i)} p_l^{(t,i)} \right) \mathbb{I}(q_l^{(t,i)} \neq q^{(t)}(\mu)).$$

*Proof.* To prove the first result, we have

$$\frac{\partial \nu^{\top} W_Q^{(t)\top} W_Q^{(t)} \mu}{\partial t}$$

$$= \frac{1}{n\sqrt{m}} \sum_{i=1}^{n} g_i^{(t)} y_i \sum_{l=1}^{L} \sum_{j=1}^{m_1} a_j \nu^{\top} W_Q^{(t)\top} K^{(t,i)} \text{diag} \left( V^{(t,i)\top} w_j^{(t)} - w_j^{(t)\top} V^{(t,i)} p_l^{(t,i)} \right) p_l^{(t,i)} x_l^{(i)\top} \mu$$

$$+ \frac{1}{n\sqrt{m}} \sum_{i=1}^{n} g_i^{(t)} y_i \sum_{l=1}^{L} \sum_{j=1}^{m_1} a_j \mu^{\top} W_Q^{(t)\top} K^{(t,i)} \text{diag} \left( V^{(t,i)\top} w_j^{(t)} - w_j^{(t)\top} V^{(t,i)} p_l^{(t,i)} \right) p_l^{(t,i)} x_l^{(i)\top} \nu$$

$$= \frac{1}{n\sqrt{m}} \sum_{i:\mu \in X^{(i)}} g_i^{(t)} y_i \sum_{j=1}^{m_1} a_j \nu^{\top} W_Q^{(t)\top} K^{(t,i)} \text{diag} \left( V^{(t,i)\top} w_j^{(t)} - w_j^{(t)\top} V^{(t,i)} p_{l(i,\mu)}^{(t,i)} \right) p_{l(i,\mu)}^{(t,i)}$$

$$+ \frac{1}{n\sqrt{m}} \sum_{i:\nu \in X^{(i)}} g_i^{(t)} y_i \sum_{j=1}^{m_1} a_j \mu^{\top} W_Q^{(t)\top} K^{(t,i)} \text{diag} \left( V^{(t,i)\top} w_j^{(t)} - w_j^{(t)\top} V^{(t,i)} p_{l(i,\nu)}^{(t,i)} \right) p_{l(i,\nu)}^{(t,i)}.$$

To prove the second result, we have

$$\frac{\partial \nu^{\top} W_K^{(t)\top} W_K^{(t)} \mu}{\partial t}$$

$$= \frac{1}{n\sqrt{m}} \sum_{i=1}^{n} g_i^{(t)} y_i \sum_{l=1}^{L} \sum_{j=1}^{m_1} a_j \nu^{\top} W_K^{(t)\top} q_l^{(t,i)} p_l^{(t,i)\top} \text{diag} \left( w_j^{(t)\top} V^{(t,i)} - w_j^{(t)\top} V^{(t,i)} p_l^{(t,i)} \right) X^{(i)\top} \mu$$

$$+ \frac{1}{n\sqrt{m}} \sum_{i=1}^{n} g_i^{(t)} y_i \sum_{l=1}^{L} \sum_{j=1}^{m_1} a_j \mu^{\top} W_K^{(t)\top} q_l^{(t,i)} p_l^{(t,i)\top} \text{diag} \left( w_j^{(t)\top} V^{(t,i)} - w_j^{(t)\top} V^{(t,i)} p_l^{(t,i)} \right) X^{(i)\top} \nu$$

$$= \frac{1}{n\sqrt{m}} \sum_{i:\mu \in X^{(i)}} g_i^{(t)} y_i \sum_{l=1}^{L} \sum_{j=1}^{m_1} a_j \nu^{\top} W_K^{(t)\top} q_l^{(t,i)} p_{q\leftarrow l, k\leftarrow \mu}^{(t,i)} \left( w_j^{(t)\top} v^{(t,i)}(\mu) - w_j^{(t)\top} V^{(t,i)} p_l^{(t,i)} \right)$$

$$+ \frac{1}{n\sqrt{m}} \sum_{i:\nu \in X^{(i)}} g_i^{(t)} y_i \sum_{l=1}^{L} \sum_{j=1}^{m_1} a_j \mu^{\top} W_K^{(t)\top} q_l^{(t,i)} p_{q\leftarrow l, k\leftarrow \nu}^{(t,i)} \left( w_j^{(t)\top} v^{(t,i)}(\nu) - w_j^{(t)\top} V^{(t,i)} p_l^{(t,i)} \right).$$

To prove the third result, we have

$$\frac{\partial \nu^{\top} W_K^{(t)\top} W_Q^{(t)} \mu}{\partial t}$$

$$= \frac{1}{n\sqrt{m}} \sum_{i=1}^{n} g_i^{(t)} y_i \sum_{l=1}^{L} \sum_{j=1}^{m_1} a_j \nu^\top W_K^{(t)\top} K^{(t,i)} \mathrm{diag}\left(V^{(t,i)\top} w_j^{(t)} - w_j^{(t)\top} V^{(t,i)} p_l^{(t,i)}\right) p_l^{(t,i)} x_l^{(i)\top} \mu$$

$$+ \frac{1}{n\sqrt{m}} \sum_{i=1}^{n} g_i^{(t)} y_i \sum_{l=1}^{L} \sum_{j=1}^{m_1} a_j \mu^\top W_Q^{(t)\top} q_l^{(t,i)} p_l^{(t,i)\top} \mathrm{diag}\left(w_j^{(t)\top} V^{(t,i)} - w_j^{(t)\top} V^{(t,i)} p_l^{(t,i)}\right) X^{(i)\top} \nu$$

$$= \frac{1}{n\sqrt{m}} \sum_{i:\mu \in X^{(i)}} g_i^{(t)} y_i \sum_{j=1}^{m_1} a_j \nu^\top W_K^{(t)\top} K^{(t,i)} \mathrm{diag}\left(V^{(t,i)\top} w_j^{(t)} - w_j^{(t)\top} V^{(t,i)} p_{l(i,\mu)}^{(t,i)}\right) p_{l(i,\mu)}^{(t,i)}$$

$$+ \frac{1}{n\sqrt{m}} \sum_{i:\nu \in X^{(i)}} g_i^{(t)} y_i \sum_{l=1}^{L} \sum_{j=1}^{m_1} a_j \mu^\top W_Q^{(t)\top} q_l^{(t,i)} p_{q\leftarrow l, k\leftarrow \nu}^{(t,i)} \left(w_j^{(t)\top} v^{(t,i)}(\nu) - w_j^{(t)\top} V^{(t,i)} p_l^{(t,i)}\right)$$

$$= \frac{1}{n\sqrt{m}} \sum_{i:\mu,\nu \in X^{(i)}} g_i^{(t)} y_i \sum_{j=1}^{m_1} a_j \|k^{(t)}(\nu)\|_2^2 \left(v^{(t)\top}(\nu) w_j^{(t)} - w_j^{(t)\top} V^{(t,i)} p_{l(i,\mu)}^{(t,i)}\right) p_{q\leftarrow \mu, k\leftarrow \nu}^{(t,i)}$$

$$+ \frac{1}{n\sqrt{m}} \sum_{i:\mu \in X^{(i)}} g_i^{(t)} y_i \sum_{j=1}^{m_1} a_j \sum_{l=1}^{L} \nu^\top W_K^{(t)\top} K_l^{(t,i)} \left(V_l^{(t,i)\top} w_j^{(t)} - w_j^{(t)\top} V^{(t,i)} p_{l(i,\mu)}^{(t,i)}\right) p_{q\leftarrow \mu, k\leftarrow l}^{(t,i)} \mathbb{I}(K_l^{(t,i)} \neq k^{(t)}(\nu))$$

$$+ \frac{1}{n\sqrt{m}} \sum_{i:\nu,\mu \in X^{(i)}} g_i^{(t)} y_i \sum_{j=1}^{m_1} a_j \|q^{(t)}(\mu)\|_2^2 p_{q\leftarrow \mu, k\leftarrow \nu}^{(t,i)} \left(w_j^{(t)\top} v^{(t,i)}(\nu) - w_j^{(t)\top} V^{(t,i)} p_{l(i,\mu)}^{(t,i)}\right)$$

$$+ \frac{1}{n\sqrt{m}} \sum_{i:\nu \in X^{(i)}} g_i^{(t)} y_i \sum_{l=1}^{L} \sum_{j=1}^{m_1} a_j \mu^\top W_Q^{(t)\top} q_l^{(t,i)} p_{q\leftarrow l, k\leftarrow \nu}^{(t,i)} \left(w_j^{(t)\top} v^{(t,i)}(\nu) - w_j^{(t)\top} V^{(t,i)} p_l^{(t,i)}\right) \mathbb{I}(q_l^{(t,i)} \neq q^{(t)}(\mu)).$$

$\square$

## D   Initialization

**Lemma D.1.** *With probability at least $1 - \delta$ over the randomness of the initialization of $W_K$ and $W_Q$, for any $l_1, l_2 \in [d]$, we have*

$$\left|\left\langle W_K^{(0)} \mu_{l_1}, W_Q^{(0)} \mu_{l_2}\right\rangle\right| \leq \sigma_0^2 m \left(\sqrt{\frac{4}{m} \log \frac{2d}{\delta}} + \frac{4}{m} \log \frac{2d}{\delta}\right),$$

$$\left|\left\langle W_K^{(0)} \mu_{l_1}, W_K^{(0)} \mu_{l_2}\right\rangle\right| \leq \sigma_0^2 m \left(\sqrt{\frac{4}{m} \log \frac{2d}{\delta}} + \frac{4}{m} \log \frac{2d}{\delta}\right), \quad l_1 \neq l_2$$

$$\left|\left\langle W_Q^{(0)} \mu_{l_1}, W_Q^{(0)} \mu_{l_2}\right\rangle\right| \leq \sigma_0^2 m \left(\sqrt{\frac{4}{m} \log \frac{2d}{\delta}} + \frac{4}{m} \log \frac{2d}{\delta}\right), \quad l_1 \neq l_2$$

*and for any $l \in [d]$,*

$$\|W_K^{(0)} \mu_l\|_2^2 = \sigma_0^2 m \left(1 \pm \left(\sqrt{\frac{4}{m} \log \frac{2d}{\delta}} + \frac{4}{m} \log \frac{2d}{\delta}\right)\right),$$

$$\|W_Q^{(0)} \mu_l\|_2^2 = \sigma_0^2 m \left(1 \pm \left(\sqrt{\frac{4}{m} \log \frac{2d}{\delta}} + \frac{4}{m} \log \frac{2d}{\delta}\right)\right).$$

*Proof.* Note that $W_K^{(0)} \mu_{l_1}, W_Q^{(0)} \mu_{l_2} \sim \mathcal{N}(0, \sigma_0^2 I)$. The rest of proof applies Lemma H.2. $\square$

**Corollary D.2.** *For all $i \in [n]$, $l, k \in [L]$, we have*

$$p_{l,k}^{(0,i)} = \frac{1}{L} \pm \widetilde{O}\left(\frac{1}{Lm}\right).$$

*Proof.* Following from Lemma G.2 and from the first-order Taylor approximation on the softmax function from 0, we have

$$p_{l,k}^{(0,i)} = \frac{1}{L} \pm O\left(\frac{\sigma_0^2 m}{L\sqrt{m}}\left(\sqrt{\frac{4}{m}\log\frac{2d}{\delta}} + \frac{4}{m}\log\frac{2d}{\delta}\right)\right).$$

The corollary then follows from Condition 1. $\qquad\square$

**Lemma D.3.** *With probability at least $1 - \delta$ over the randomness of the initialization of $W$ and $W_V$, then for $l_1 \neq l_2 \in [d]$, we have*

$$\left|\left\langle W_V^{(0)}\mu_{l_1}, W_V^{(0)}\mu_{l_2}\right\rangle\right| \leq \sigma_0^2 m\left(\sqrt{\frac{4}{m}\log\frac{2d}{\delta}} + \frac{4}{m}\log\frac{2d}{\delta}\right),$$

*for $j_1 \neq j_2 \in [m_1]$, we have*

$$\left|\left\langle w_{j_1}^{(0)}, w_{j_2}^{(0)}\right\rangle\right| \leq \sigma_1^2 m\left(\sqrt{\frac{4}{m}\log\frac{2m_1^2}{\delta}} + \frac{4}{m}\log\frac{2m_1^2}{\delta}\right),$$

*and for all $j \in [m_1]$, $l \in [d]$, we have*

$$\left|\left\langle w_j^{(0)}, W_V^{(0)}\mu_l\right\rangle\right| \leq \sigma_0\sigma_1 m\left(\sqrt{\frac{4}{m}\log\frac{2m_1 d}{\delta}} + \frac{4}{m}\log\frac{2m_1 d}{\delta}\right)$$

$$\|w_j\|_2^2 = \sigma_1^2 m\left(1 \pm \left(\sqrt{\frac{4}{m}\log\frac{2m_1}{\delta}} + \frac{4}{m}\log\frac{2m_1}{\delta}\right)\right)$$

$$\|W_V^{(0)}\mu_l\|_2^2 = \sigma_0^2 m\left(1 \pm \left(\sqrt{\frac{4}{m}\log\frac{2d}{\delta}} + \frac{4}{m}\log\frac{2d}{\delta}\right)\right).$$

*Proof.* The proof is similar to that for Lemma D.1 and is omitted. $\qquad\square$

**Lemma D.4.** *Conditioned on the success of the event in Corollary D.2, for all $i \in [n]$, $l \in [L]$, with probability at least $1 - \delta$ over the randomness in the initialization of $W_V$,*

$$\|W_V^{(0)}X^{(i)}p_l^{(0,i)}\|_2^2 = \frac{\sigma_0^2 m}{L}\left(1 \pm \sqrt{\frac{4}{m}\log\frac{2nL}{\delta}} \pm \frac{4}{m}\log\frac{2nL}{\delta}\right)$$

*Proof.* First of all, by Corollary D.2 and Assumption 2.3,

$$\mathbb{E}\left[\|W_V^{(0)}X^{(i)}p_l^{(0,i)}\|_2^2\right] = \mathbb{E}\left[\sum_{j=1}^{m}\left(\left\langle\left(W_V^{(0)}\right)_j, X^{(i)}p_l^{(0,i)}\right\rangle\right)^2\right] = \sigma_0^2 m\|X^{(i)}p_l^{(0,i)}\|_2^2 = \frac{\sigma_0^2 m}{L}.$$

Finally, applying Bernstein's inequality and taking a union bound over $[n]$ and $[L]$ we finish the proof. $\qquad\square$

**Corollary D.5.** *Conditioned on the success of Lemma D.4, with probability at least $1 - \delta$ over the randomness of $W$, for all $j \in [m_1]$, $i \in [n]$, $l \in [L]$, we have*

$$\left|w_j^{(0)\top}V^{(0,i)}p_l^{(0,i)}\right| \leq \sigma_1\sigma_0\sqrt{\frac{m}{L}\log\frac{m_1 nL}{\delta}}.$$

*Proof.* Conditioned on $V^{(0,i)}p_l^{(0,i)}$, we have $w_j^{(0)\top}V^{(0,i)}p_l^{(0,i)} \sim \mathcal{N}(0, \sigma_1^2\|V^{(0,i)}p_l^{(0,i)}\|_2^2)$. Thus, by Gaussian tail bound and a union bound over $j \in [m_1]$, $i \in [n]$, $l \in [L]$, with probability at least $1 - \delta$, we have

$$\left|w_j^{(0)\top}V^{(0,i)}p_l^{(0,i)}\right| \leq \sigma_1\sigma_0\sqrt{\frac{m}{L}\log\frac{m_1 nL}{\delta}}.$$

$\qquad\square$

**Lemma D.6.** *With probability at least $1 - \delta$ over the randomness in the initialization of $a$, we have*

$$|\mathcal{S}_{+1}| = m_1 \left( \frac{1}{2} \pm \sqrt{\frac{2\log(4/\delta)}{m_1}} \right),$$

$$|\mathcal{S}_{-1}| = m_1 \left( \frac{1}{2} \pm \sqrt{\frac{2\log(4/\delta)}{m_1}} \right).$$

*Proof.* The proof follows by applying Hoeffding's inequality. $\square$

**Lemma D.7** (Initial sub-network output). *Assume that the success of the events in Lemma D.3 holds. For all $i \in [d]$, with probability at least $1 - \delta$ over the randomness in the weight initialization, we have*

$$|G^{(0)}(\mu_i)| \leq \widetilde{O}(\sigma_1 \sigma_0 \sqrt{mm_1}).$$

*Proof.* Consider a fixed $i \in [d]$. By Lemma D.3, we have $\|W_V^{(0)} \mu_i\|_2^2 = \Theta(\sigma_0^2 m)$. Thus, conditioned on $W_V^{(0)} \mu_i$, we have $\sum_{j=1}^{m_1} w_j^{(0)} W_V^{(0)} \mu_i \sim \mathcal{N}(0, \sigma_1^2 \sigma_0^2 mm_1)$. Thus, by Gaussian concentration bound, we have $|G^{(0)}(\mu_i)| \leq \widetilde{O}(\sigma_1 \sigma_0 \sqrt{mm_1})$. $\square$

**Lemma D.8** (Initial network output). *Assume that the success of the events in Lemma D.3 and Lemma D.4 holds. For all $i \in [n]$, with probability at least $1 - \delta$ over the randomness in the weight initialization, we have*

$$|F^{(0)}(X^{(i)})| \leq 2\sigma_0 \sigma_1 \sqrt{2Lm_1 m \log(2Ln/\delta)} \leq 0.01.$$

*Proof.* For fixed $l \in L$, $j \in [m]$, $i \in [n]$, by Corollary D.5, we have

$$\left| w_j^{(0)\top} V^{(0,i)} p_l^{(0,i)} \right| \leq \sigma_1 \sigma_0 \sqrt{\frac{m}{L} \log \frac{m_1 nL}{\delta}}.$$

Thus, this implies that $a_j w_j^{(0)\top} V^{(0,i)} p_l^{(0,i)}$ is a sub-Gaussian random variable with variance proxy $\sigma_0^2 \sigma_1^2 \frac{m}{L} \log \frac{m_1 nL}{\delta}$. Therefore, the following inequality holds.

$$\mathbb{P}\left[ \left| \sum_{j=1}^{m_1} a_j w_j^{(0)\top} V^{(0,i)} p_l^{(0,i)} \right| \geq 2\sigma_0 \sigma_1 \sqrt{\frac{2m_1 m (\log \frac{m_1 nL}{\delta}) \log(2/\delta)}{L}} \right] \leq \delta.$$

Taking a union bound over $i \in [n]$, $l \in [L]$, with probability at least $1 - \delta$, for all $i \in [n]$, we have

$$|F^{(0)}(X^{(i)})| \leq 2\sigma_0 \sigma_1 \sqrt{2Lm_1 m (\log \frac{m_1 nL}{\delta}) \log(2Ln/\delta)}.$$

Finally, by Condition 1, we can make $|F^{(0)}(X^{(i)})| \leq 0.01$. $\square$

## E   Training Dynamics: Phase 1

During Phase 1 of training, the linear layer quickly aligns with the target signals and all the remaining quantities stay roughly the same. The analysis need to keep track of the evolution of the above quantities with respect to the two signals $\mu_1, \mu_2$, the common token $\mu_3$ and the random tokens.

**Definition E.1** (Radius of keys and queries). *Define the radius of keys and queries $R_K, R_Q$ respectively to be*

$$R_K := \max_{i,j \in [d]} \left| \mu_i^\top W_K^{(t)\top} W_K^{(t)} \mu_j - \mu_i^\top W_K^{(0)\top} W_K^{(0)} \mu_j \right|,$$

$$R_Q := \max_{i,j \in [d]} \left| \mu_i^\top W_Q^{(t)\top} W_Q^{(t)} \mu_j - \mu_i^\top W_Q^{(0)\top} W_Q^{(0)} \mu_j \right|.$$

**Definition E.2** (Phase 1)**.** *Define the range of Phase 1 to be $[0, T_1]$, where $T_1 = \min\{t', C_{T_1}/(\sigma_1^2 m m_1)\}$ for some sufficiently large constant $C_{T_1}$ and $t'$ is defined to be the maximum time such that for all $t \leq t'$, all of the following hold:*

1. $\max_{j \in [m], \mu \in \{\mu_i\}_{i=1}^3} \left| w_j^{(t)} W_V^{(t)} \mu - w_j^{(0)} W_V^{(0)} \mu \right| \leq R$ where $R < O(1/m_1)$;

2. $\max_{j \in [m], \mu \notin \{\mu_i\}_{i=1}^3} \left| w_j^{(t)} W_V^{(t)} \mu - w_j^{(0)} W_V^{(0)} \mu \right| \leq O(R/n + R/\sqrt{m})$;

3. $\max_{\mu,\nu \in \{\mu\}_{i=1}^d} \left| \mu^\top W_Q^{(t)\top} W_K^{(t)} \nu - \mu^\top W_Q^{(0)\top} W_K^{(0)} \nu \right| \leq R_S$ where $R_S \leq O(1/(m\sqrt{m}))$;

4. $R_K, R_Q \leq \widetilde{O}(\sigma_0^2 \sqrt{m})$.

Based on this definition, we can further obtain the maximum softmax probability change as follows.

**Proposition E.3.** *Define*

$$R_P := \max_{i \in [n],\ \mu,\nu \in X^{(i)}} \left| p_{q \leftarrow \mu, k \leftarrow \nu}^{(t,i)} - p_{q \leftarrow \mu, k \leftarrow \nu}^{(0,i)} \right|.$$

*Then*

$$R_P = O\left( \frac{1}{\sqrt{m}L} + \frac{L}{m} \right).$$

*Proof.* By Lemma G.2, we have $R_P \leq O(R_S/L + R_S^2 L) = O\left( \frac{1}{\sqrt{m}L} + \frac{L}{m} \right)$. $\qquad\square$

Initially, the loss for the samples with one signal will increase.

## E.1   Initial Gradients

**Lemma E.4** (Signal updates, same as Lemma 4.1)**.** *At $t = 0$, for $\mu \in \{\mu_1, \mu_2\}$, we have*

$$\frac{\partial a_j w_j^{(0)} W_V^{(0)} \mu}{\partial t} = \Theta((\sigma_0^2 + \sigma_1^2)m).$$

*Proof.* Take $\mu = \mu_1$. First of all, by the gradient flow update in Lemma C.1, we have

$$
\begin{aligned}
\frac{\partial w_{j_1}^{(t)\top} W_V^{(t)} \mu}{\partial t} &= \frac{1}{n} \sum_{i_2=1}^n g_{i_2}^{(t)} y_{i_2} \sum_{l_2=1}^L \sum_{j_2=1}^{m_1} a_{j_2} \left\langle w_{j_1}^{(t)}, w_{j_2}^{(t)} \right\rangle \left( X^{(i_2)} p_{l_2}^{(t,i_2)} \right)^\top \mu \\
&\quad + \frac{1}{n} \sum_{i_1=1}^n g_{i_1}^{(t)} y_{i_1} \sum_{l_1=1}^L a_{j_1} p_{l_1}^{(t,i_1)\top} V^{(t,i_1)\top} W_V^{(t)} \mu \\
&= \frac{1}{n} \sum_{i_2:\ \mu \in X^{(i_2)}} g_{i_2}^{(t)} y_{i_2} \sum_{l_2=1}^L \sum_{j_2=1}^{m_1} a_{j_2} \left\langle w_{j_1}^{(t)}, w_{j_2}^{(t)} \right\rangle p_{q \leftarrow l_2, k \leftarrow \mu}^{(t,i_2)} \\
&\quad + \frac{1}{n} \sum_{i_1=1}^n g_{i_1}^{(t)} y_{i_1} \sum_{l_1=1}^L a_{j_1} \sum_{l_2=1}^L \left\langle p_{l_1,l_2}^{(t,i_1)} v_{l_2}^{(t,i_1)}, v^{(t)}(\mu) \right\rangle \\
&= \frac{1}{n} \sum_{i_2:\ \mu \in X^{(i_2)}} g_{i_2}^{(t)} y_{i_2} \sum_{l_2=1}^L \left( a_{j_1} \|w_{j_1}^{(t)}\|_2^2 + \sum_{j_2 \neq j_1} a_{j_2} \left\langle w_{j_1}^{(t)}, w_{j_2}^{(t)} \right\rangle \right) p_{q \leftarrow l_2, k \leftarrow \mu}^{(t,i_2)} \\
&\quad + \frac{1}{n} \sum_{i_1:\mu \notin X^{(i_1)}} g_{i_1}^{(t)} y_{i_1} \sum_{l_1=1}^L a_{j_1} \sum_{l_2=1}^L \left\langle p_{l_1,l_2}^{(t,i_1)} v_{l_2}^{(t,i_1)}, v^{(t)}(\mu) \right\rangle \\
&\quad + \frac{1}{n} \sum_{i_1:\mu \in X^{(i_1)}} g_{i_1}^{(t)} y_{i_1} \sum_{l_1=1}^L a_{j_1} \left( \|v^{(t)}(\mu)\|_2^2 p_{q \leftarrow l_1, k \leftarrow \mu}^{(t,i_1)} + \sum_{l_2 \neq l(i_1,\mu)} \left\langle p_{l_1,l_2}^{(t,i_1)} v_{l_2}^{(t,i_1)}, v^{(t)}(\mu) \right\rangle \right)
\end{aligned}
$$

$$= \frac{1}{n} \sum_{i:\mu \in X^{(i)}} g_i^{(t)} y_i \sum_{l=1}^{L} a_{j_1} \left( \|w_{j_1}^{(t)}\|_2^2 + \|v^{(t)}(\mu)\|_2^2 \right) p_{q \leftarrow l, k \leftarrow \mu}^{(t,i)}$$

$$+ \underbrace{\frac{1}{n} \sum_{i:\mu \in X^{(i)}} g_i^{(t)} y_i \sum_{l=1}^{L} \sum_{j_2 \neq j_1} a_{j_2} \left\langle w_{j_1}^{(t)}, w_{j_2}^{(t)} \right\rangle p_{q \leftarrow l, k \leftarrow \mu}^{(t,i)}}_{\varepsilon_1}$$

$$+ \underbrace{\frac{1}{n} \sum_{i=1}^{n} g_i^{(t)} y_i \sum_{l_1=1}^{L} a_{j_1} \sum_{l_2=1}^{L} \left\langle p_{l_1,l_2}^{(t,i)} v_{l_2}^{(t,i)}, v^{(t)}(\mu) \right\rangle \mathbb{I}(v_{l_2}^{(t,i)} \neq v^{(t)}(\mu))}_{\varepsilon_2}.$$

Now, by Lemma D.3, Corollary D.2 and Lemma D.8, we have

$$a_{j_1} \frac{1}{n} \sum_{i:\mu_1 \in X^{(i)}} g_i^{(0)} y_i \sum_{l=1}^{L} a_{j_1} \left( \|w_{j_1}^{(0)}\|_2^2 + \|v^{(0)}(\mu)\|_2^2 \right) p_{q \leftarrow l, k \leftarrow \mu_1}^{(0,i)}$$

$$= \frac{1}{n} \sum_{i \in I_1} g_i^{(0)} \sum_{l=1}^{L} \left( \|w_j^{(0)}\|_2^2 + \|v^{(0)}(\mu_1)\|_2^2 \right) p_{q \leftarrow l, k \leftarrow \mu_1}^{(0,i)} - \frac{1}{n} \sum_{i \in I_2} g_i^{(0)} \sum_{l=1}^{L} \left( \|w_j^{(0)}\|_2^2 + \|v^{(0)}(\mu_1)\|_2^2 \right) p_{q \leftarrow l, k \leftarrow \mu_1}^{(0,i)}$$

$$= \left( \frac{1}{3} \pm 0.01 \right) L \cdot (\sigma_1^2 m + \sigma_0^2 m) \left( 1 \pm \left( \sqrt{\frac{4}{m} \log \frac{2d}{\delta}} + \frac{4}{m} \log \frac{2d}{\delta} \right) \right) \frac{1}{L}(1 + o(1)).$$

On the other hand, by Proposition E.5, we have

$$|\varepsilon_1| = \left| \frac{1}{n} \sum_{i:\mu_1 \in X^{(i)}} g_i^{(0)} y_i \sum_{l=1}^{L} \sum_{j_2 \neq j_1} a_{j_2} \left\langle w_{j_1}^{(0)}, w_{j_2}^{(0)} \right\rangle p_{q \leftarrow l, k \leftarrow \mu_1}^{(0,i)} \right|$$

$$\leq \sigma_1^2 \sqrt{m_1} m \left( \sqrt{\frac{4}{m} \log \frac{2m_1^2}{\delta}} + \frac{4}{m} \log \frac{2m_1^2}{\delta} \right) \sqrt{\log \frac{m_1}{\delta}};$$

$$|\varepsilon_2| = \left| \frac{1}{n} \sum_{i=1}^{n} g_i^{(0)} y_i \sum_{l_1=1}^{L} a_{j_1} \sum_{l_2=1}^{L} \left\langle p_{l_1,l_2}^{(0,i)} v_{l_2}^{(0,i)}, v^{(0)}(\mu_1) \right\rangle \mathbb{I}(v_{l_2}^{(0,i)} \neq v^{(0)}(\mu_1)) \right|$$

$$\leq \sigma_0^2 m \sqrt{L} \left( \sqrt{\frac{4}{m} \log \frac{2nL}{\delta}} + \frac{4}{m} \log \frac{2nL}{\delta} \right).$$

If $m \geq C m_1 \log^2 \frac{m_1}{\delta}$ in Condition 1 for some sufficiently large $C$, then $|\varepsilon_1| \leq 0.01 \sigma_1^2 m$; and if $m \geq C' L \log \frac{2nL}{\delta}$ for some sufficiently large $C'$, then $|\varepsilon_2| \leq 0.01 \sigma_0^2 m$. $\qquad \square$

**Proposition E.5.** *Assume the events in Lemma D.3 and Corollary D.2 succeed. With probability at least $1 - \delta$ over the randomness in the weight initialization, for all $j_1 \in [m_1]$, we have*

$$\left| \sum_{j_2:j_2 \neq j_1} a_{j_2} \left\langle w_{j_1}^{(0)}, w_{j_2}^{(0)} \right\rangle \right| \leq \sigma_1^2 \sqrt{m_1} m \left( \sqrt{\frac{4}{m} \log \frac{2m_1^2}{\delta}} + \frac{4}{m} \log \frac{2m_1^2}{\delta} \right) \sqrt{\log \frac{4m_1}{\delta}}.$$

*Further, for all $\mu \in \{\mu_i\}_{i=1}^{d}$, we have*

$$\left| \frac{1}{n} \sum_{i:\mu \in X^{(i)}} g_i^{(0)} y_i \sum_{l=1}^{L} \sum_{j_2 \neq j_1} a_{j_2} \left\langle w_{j_1}^{(0)}, w_{j_2}^{(0)} \right\rangle p_{q \leftarrow l, k \leftarrow \mu}^{(0,i)} \right|$$

$$\leq \frac{|i : \mu \in X^{(i)}|}{n} \sigma_1^2 \sqrt{m_1} m \left( \sqrt{\frac{4}{m} \log \frac{2m_1^2}{\delta}} + \frac{4}{m} \log \frac{2m_1^2}{\delta} \right) \sqrt{\log \frac{m_1}{\delta}},$$

*and*

$$\left| \frac{1}{n} \sum_{i=1}^{n} g_i^{(0)} y_i \sum_{l_1=1}^{L} a_{j_1} \sum_{l_2=1}^{L} \left\langle p_{l_1,l_2}^{(0,i)} v_{l_2}^{(0,i)}, v^{(0)}(\mu) \right\rangle \mathbb{I}(v_{l_2}^{(0,i)} \neq v^{(0)}(\mu)) \right|$$

$$\leq \sigma_0^2 m\sqrt{L}\left(\sqrt{\frac{4}{m}\log\frac{2nLd}{\delta}}+\frac{4}{m}\log\frac{2nLd}{\delta}\right).$$

*Proof.* First, fix $i,l$, and consider the randomness of $a$. By Lemma D.3, $a_{j_2}\left\langle w_{j_1}^{(0)},w_{j_2}^{(0)}\right\rangle$ is a sub-Gaussian random variable with variance proxy $\sigma_1^4 m^2\left(\sqrt{\frac{4}{m}\log\frac{2m_1^2}{\delta}}+\frac{4}{m}\log\frac{2m_1^2}{\delta}\right)^2$. This implies that with probability at least $1-\delta/2$, for all $j_1\in[m_1]$, we have

$$\left|\sum_{j_2:j_2\neq j_1}a_{j_2}\left\langle w_{j_1}^{(0)},w_{j_2}^{(0)}\right\rangle\right|\leq\sigma_1^2\sqrt{m_1}m\left(\sqrt{\frac{4}{m}\log\frac{2m_1^2}{\delta}}+\frac{4}{m}\log\frac{2m_1^2}{\delta}\right)\sqrt{\log\frac{4m_1}{\delta}}.$$

Thus, by Corollary D.2, for all $\mu\in\{\mu_i\}_{i=1}^d$, we have

$$\left|\frac{1}{n}\sum_{i:\mu_1\in X^{(i)}}g_i^{(0)}y_i\sum_{l=1}^L\sum_{j_2:j_2\neq j_1}a_{j_2}\left\langle w_{j_1}^{(0)},w_{j_2}^{(0)}\right\rangle p_{q\leftarrow l,k\leftarrow\mu}^{(0,i)}\right|$$

$$\leq\frac{|i:\mu\in X^{(i)}|}{n}\sigma_1^2\sqrt{m_1}m\left(\sqrt{\frac{4}{m}\log\frac{2m_1^2}{\delta}}+\frac{4}{m}\log\frac{2m_1^2}{\delta}\right)\sqrt{\log\frac{m_1}{\delta}}.$$

We next derive the second inequality. Consider the randomness in $W_V^{(0)}$. Note that

$$\sum_{l_2=1}^L p_{l_1,l_2}^{(0,i)}v_{l_2}^{(0,i)}\mathbb{I}(v_{l_2}^{(0,i)}\neq v^{(0)}(\mu))\sim\mathcal{N}\left(0,\sigma_0^2\sum_{l_2=1}^L(p_{l_1,l_2}^{(0,i)})^2\mathbb{I}(v_{l_2}^{(0,i)}\neq v^{(0)}(\mu))I\right).$$

Thus, by Lemma H.2 and Corollary D.2 and taking a union bound over $i\in[n]$, $l_1\in[L]$, $\mu\in\{\mu_i\}_{i=1}^d$, we have with probability at least $1-\delta/2$,

$$\left|\left\langle\sum_{l_2=1}^L p_{l_1,l_2}^{(0,i)}v_{l_2}^{(0,i)}\mathbb{I}(v_{l_2}^{(0,i)}\neq v^{(0)}(\mu)),v^{(0)}(\mu)\right\rangle\right|\leq\sigma_0^2 m\frac{1}{\sqrt{L}}\left(\sqrt{\frac{4}{m}\log\frac{2nLd}{\delta}}+\frac{4}{m}\log\frac{2nLd}{\delta}\right).$$

Therefore,

$$\left|\frac{1}{n}\sum_{i=1}^n g_i^{(0)}y_i\sum_{l_1=1}^L a_{j_1}\sum_{l_2=1}^L\left\langle p_{l_1,l_2}^{(0,i)}v_{l_2}^{(0,i)},v^{(0)}(\mu)\right\rangle\mathbb{I}(v_{l_2}^{(0,i)}\neq v^{(0)}(\mu))\right|$$

$$\leq\sigma_0^2 m\sqrt{L}\left(\sqrt{\frac{4}{m}\log\frac{2nLd}{\delta}}+\frac{4}{m}\log\frac{2nLd}{\delta}\right).$$

$\square$

**Lemma E.6** (Random token updates). *For $\mu\in\{\mu_i\}_{i=4}^d$, we have*

$$\frac{\partial a_j w_j^{(0)}W_V^{(0)}\mu}{\partial t}=O\left(\frac{1}{n}(\sigma_0^2+\sigma_1^2)m+\sigma_0^2\sqrt{mL}\right).$$

*Proof.* Following the proof of Lemma E.4, we have

$$\frac{\partial a_j w_j^{(0)}W_V^{(0)}\mu}{\partial t}=\frac{1}{n}\sum_{i_2=1}^n g_{i_2}^{(0)}y_{i_2}\sum_{l_2=1}^L\sum_{j_2=1}^{m_1}a_{j_2}\left\langle w_{j_1}^{(0)},w_{j_2}^{(0)}\right\rangle\left(X^{(i_2)}p_{l_2}^{(0,i_2)}\right)^\top\mu$$

$$+\frac{1}{n}\sum_{i_1=1}^n g_{i_1}^{(0)}y_{i_1}\sum_{l_1=1}^L a_{j_1}p_{l_1}^{(0,i_1)\top}V^{(0,i_1)\top}W_V^{(0)}\mu$$

$$=\frac{1}{n}\sum_{i:\mu\in X^{(i)}}g_i^{(0)}y_i\sum_{l=1}^L a_{j_1}\left(\|w_{j_1}^{(0)}\|_2^2+\|v^{(0)}(\mu)\|_2^2\right)p_{q\leftarrow l,k\leftarrow\mu}^{(0,i)}$$

$$+ \frac{1}{n} \underbrace{\sum_{i:\mu \in X^{(i)}} g_i^{(0)} y_i \sum_{l=1}^{L} \sum_{j_2:j_2 \neq j_1} a_{j_2} \left\langle w_{j_1}^{(0)}, w_{j_2}^{(0)} \right\rangle p_{q \leftarrow l, k \leftarrow \mu}^{(0,i)}}_{\varepsilon_1}$$

$$+ \frac{1}{n} \underbrace{\sum_{i=1}^{n} g_i^{(0)} y_i \sum_{l_1=1}^{L} a_{j_1} \sum_{l_2=1}^{L} \left\langle p_{l_1,l_2}^{(0,i)} v_{l_2}^{(0,i)}, v^{(0)}(\mu) \right\rangle \mathbb{I}(v_{l_2}^{(0,i)} \neq v^{(0)}(\mu))}_{\varepsilon_2}$$

By Proposition E.5 and the fact that only one $X^{(i)}$ satisfies $\mu \in X^{(i)}$, we have

$$\frac{1}{n} \sum_{i:\mu \in X^{(i)}} g_i^{(0)} y_i \sum_{l=1}^{L} a_{j_1} \left( \|w_{j_1}^{(0)}\|_2^2 + \|v^{(0)}(\mu)\|_2^2 \right) p_{q \leftarrow l, k \leftarrow \mu}^{(0,i)} = \Theta\left(\frac{1}{n}(\sigma_0^2 + \sigma_1^2)m\right),$$

and

$$|\varepsilon_1| \leq \frac{1}{n} \sigma_1^2 \sqrt{m_1} m \left( \sqrt{\frac{4}{m} \log \frac{2m_1^2}{\delta}} + \frac{4}{m} \log \frac{2m_1^2}{\delta} \right) \sqrt{\log \frac{4m_1}{\delta}},$$

$$|\varepsilon_2| \leq \sigma_0^2 m \sqrt{L} \left( \sqrt{\frac{4}{m} \log \frac{2nLd}{\delta}} + \frac{4}{m} \log \frac{2nLd}{\delta} \right).$$

$\square$

## E.2 Maximum Perturbation of Neuron Outputs

**Lemma E.7.** *For all $t \leq T_1$, for all $i \in [n]$, $l \in [L]$, we have*

$$\left| w_j^{(t)} W_V^{(t)} X^{(i)} p_l^{(t,i)} - w_j^{(0)} W_V^{(0)} X^{(i)} p_l^{(0,i)} \right| \leq \widetilde{O}(LR_P \sigma_0 \sigma_1 \sqrt{m}) + 4R/L.$$

*Proof.* By Definition E.2, we have

$$\left| w_j^{(t)} V^{(t,i)} p_l^{(t,i)} - w_j^{(0)} V^{(0,i)} p_l^{(0,i)} \right|$$
$$\leq \sum_{l'=1}^{L} \left| w_j^{(t)} V_{l'}^{(t,i)} p_{l,l'}^{(t,i)} - w_j^{(0)} V_{l'}^{(0,i)} p_{l,l'}^{(0,i)} \right|$$
$$\leq \sum_{l':V_{l'} \in v(\mu_1,\mu_2,\mu_3)} (R_P \widetilde{O}(\sigma_0 \sigma_1 \sqrt{m}) + R/L) + \sum_{l':V_{l'} \notin v(\mu_1,\mu_2,\mu_3)} (R_P \widetilde{O}(\sigma_0 \sigma_1 \sqrt{m}) + R/(nL))$$
$$\leq \widetilde{O}(R_P \sigma_0 \sigma_1 \sqrt{m} + R/L) + \widetilde{O}(LR_P \sigma_0 \sigma_1 \sqrt{m} + R/n)$$
$$= \widetilde{O}(LR_P \sigma_0 \sigma_1 \sqrt{m}) + 4R/L.$$

By our choice of parameters in Condition 1 and Definition E.2, we have $\widetilde{O}(LR_P \sigma_0 \sigma_1 \sqrt{m}) + 4R/L < \sigma_0 \sigma_1 \sqrt{m/L}$. $\square$

## E.3 Perturbation Term Involving Correlation of Value-transformed Data

**Proposition E.8.** *With probability at least $1 - \delta$, for all $\mu \in \{\mu_i\}_{i=1}^{d}$, we have*

$$\left| \sum_{j=1}^{m_1} a_j \mu^\top W_V^{(0)\top} w_j^{(0)} \right| \leq \widetilde{O}(\sigma_0 \sigma_1 \sqrt{mm_1}).$$

*Proof.* The proof is similar to Proposition E.22, and is omitted. $\square$

**Lemma E.9** (Value correlation change). *For all $\mu, \nu \in \{\mu_i\}_{i=1}^d$, we have*

$$\left|\frac{\partial \langle v^{(t)}(\mu), v^{(t)}(\nu)\rangle}{\partial t}\right| \leq \frac{\left|\{i : \mu \in X^{(i)}\}\right| + \left|\{i : \nu \in X^{(i)}\}\right|}{n} O(1),$$

*and thus,*

$$\left|\left\langle v^{(t)}(\mu), v^{(t)}(\nu)\right\rangle - \left\langle v^{(0)}(\mu), v^{(0)}(\nu)\right\rangle\right| \leq t\frac{\left|\{i : \mu \in X^{(i)}\}\right| + \left|\{i : \nu \in X^{(i)}\}\right|}{n} O(1)$$

*for all $t \leq T_1$. Thus, for $\mu \neq \nu$, we have $\left|\langle v^{(t)}(\mu), v^{(t)}(\nu)\rangle\right| \leq \widetilde{O}(\sigma_0^2 \sqrt{m})$ and $\|v^{(t)}(\mu)\|_2^2 = \Theta(\sigma_0^2 m)$ for $t \leq T_1$.*

*Proof.* By Lemma C.3, we have

$$\frac{\partial \nu W_V^{(t)\top} W_V^{(t)} \mu}{\partial t} = \frac{1}{n} \sum_{i:\mu \in X^{(i)}} g_i^{(t)} y_i \sum_{l=1}^{L} \sum_{j=1}^{m_1} a_j \nu^\top W_V^{(t)\top} w_j^{(t)} p_{q\leftarrow l, k\leftarrow \mu}^{(t,i)}$$

$$+ \frac{1}{n} \sum_{i:\nu \in X^{(i)}} g_i^{(t)} y_i \sum_{l=1}^{L} \sum_{j=1}^{m_1} a_j \mu^\top W_V^{(t)\top} w_j^{(t)} p_{q\leftarrow l, k\leftarrow \nu}^{(t,i)}.$$

Further,

$$\left|\sum_{j=1}^{m_1} a_j \nu^\top W_V^{(t)\top} w_j^{(t)} - \sum_{j=1}^{m_1} a_j \nu^\top W_V^{(0)\top} w_j^{(0)}\right| \leq m_1 R.$$

Thus, by Proposition E.8, we have

$$\left|\sum_{j=1}^{m_1} a_j w_j^{(t)\top} W_V^{(t)} \nu\right| \leq \left|\sum_{j=1}^{m_1} a_j \dot{\sigma}_{i,l,j}^{(0)} w_j^{(0)\top} W_V^{(0)} \nu\right| + \left|\sum_{j=1}^{m_1} a_j \nu^\top W_V^{(t)\top} w_j^{(t)} - \sum_{j=1}^{m_1} a_j \nu^\top W_V^{(0)\top} w_j^{(0)}\right|$$

$$\leq \widetilde{O}(\sigma_0 \sigma_1 \sqrt{mm_1}) + m_1 R,$$

which implies

$$\left|\frac{1}{n} \sum_{i:\mu \in X^{(i)}} g_i^{(t)} y_i \sum_{l=1}^{L} \sum_{j=1}^{m_1} a_j \nu^\top W_V^{(t)\top} w_j^{(t)} p_{q\leftarrow l, k\leftarrow \mu}^{(t,i)}\right.$$

$$\left. + \frac{1}{n} \sum_{i:\nu \in X^{(i)}} g_i^{(t)} y_i \sum_{l=1}^{L} \sum_{j=1}^{m_1} a_j \mu^\top W_V^{(t)\top} w_j^{(t)} p_{q\leftarrow l, k\leftarrow \nu}^{(t,i)}\right|$$

$$\leq \frac{1}{n}\left(\left|\{i : \mu \in X^{(i)}\}\right| + \left|\{i : \nu \in X^{(i)}\}\right|\right)\left(\widetilde{O}(\sigma_0 \sigma_1 \sqrt{mm_1}) + m_1 R\right)$$

$$\leq \frac{\left|\{i : \mu \in X^{(i)}\}\right| + \left|\{i : \nu \in X^{(i)}\}\right|}{n} O(1)$$

where the last inequality applies Lemma E.7. $\square$

**Corollary E.10.** *For all $t \leq T_1$, we have*

$$\left|\frac{1}{n} \sum_{i=1}^{n} g_i^{(t)} y_i \sum_{l_1=1}^{L} a_{j_1} \sum_{l_2=1}^{L} \left\langle p_{l_1,l_2}^{(t,i)} v_{l_2}^{(t,i)}, v^{(t)}(\mu_1)\right\rangle \mathbb{I}(v_{l_2}^{(t,i)} \neq v^{(t)}(\mu_1))\right| \leq L \cdot \widetilde{O}\left(\sigma_0^2 \sqrt{m}\right).$$

*Proof.* We derive the following bound:

$$\left|\frac{1}{n} \sum_{i=1}^{n} g_i^{(t)} y_i \sum_{l_1=1}^{L} a_{j_1} \sum_{l_2=1}^{L} \left\langle p_{l_1,l_2}^{(t,i)} v_{l_2}^{(t,i)}, v^{(t)}(\mu)\right\rangle \mathbb{I}(v_{l_2}^{(t,i)} \neq v^{(t)}(\mu))\right|$$

$$\leq \sum_{l_1=1}^{L} \sum_{l_2=1}^{L} p_{l_1,l_2}^{(t,i)} \left|\left\langle v_{l_2}^{(t,i)}, v^{(t)}(\mu)\right\rangle\right| \mathbb{I}(v_{l_2}^{(t,i)} \neq v^{(t)}(\mu))$$

$$\leq L \cdot \widetilde{O}\left(\sigma_0^2 \sqrt{m}\right),$$

where the last inequality follows from Lemma E.9 and Lemma D.3. $\square$

### E.4 Perturbation Term Involving Correlation of Neurons

**Lemma E.11.** *For all $t \leq T_1$, we have*

$$\left| \frac{1}{n} \sum_{i:\mu_1 \in X^{(i)}} g_i^{(t)} y_i \sum_{l=1}^{L} \sum_{j_2 \neq j_1} a_{j_2} \left\langle w_{j_1}^{(t)}, w_{j_2}^{(t)} \right\rangle p_{q \leftarrow l, k \leftarrow \mu_1}^{(t,i)} \right| \leq \frac{|\{i : \mu_1 \in X^{(i)}\}|}{n} \widetilde{O}(\sigma_1^2 \sqrt{mm_1}).$$

*Proof.* We first derive:

$$\left| \frac{1}{n} \sum_{i:\mu_1 \in X^{(i)}} g_i^{(t)} y_i \sum_{l=1}^{L} \sum_{j_2 \neq j_1} a_{j_2} \left\langle w_{j_1}^{(t)}, w_{j_2}^{(t)} \right\rangle p_{q \leftarrow l, k \leftarrow \mu_1}^{(t,i)} \right|$$

$$\leq \left| \frac{1}{n} \sum_{i:\mu_1 \in X^{(i)}} g_i^{(t)} y_i \sum_{l=1}^{L} \sum_{j_2 \neq j_1} a_{j_2} \left\langle w_{j_1}^{(t)}, w_{j_2}^{(t)} \right\rangle \right| \cdot \max_{i,l} p_{q \leftarrow l, k \leftarrow \mu_1}^{(t,i)}$$

$$\leq \underbrace{\left| \frac{1}{n} \sum_{i:\mu_1 \in X^{(i)}} g_i^{(t)} y_i \sum_{l=1}^{L} \sum_{j_2 \neq j_1} a_{j_2} \left\langle w_{j_1}^{(0)}, w_{j_2}^{(0)} \right\rangle \right| \cdot \max_{i,l} p_{q \leftarrow l, k \leftarrow \mu_1}^{(t,i)}}_{(1)}$$

$$+ \underbrace{\left| \frac{1}{n} \sum_{i:\mu_1 \in X^{(i)}} g_i^{(t)} y_i \sum_{l=1}^{L} \sum_{j_2 \neq j_1} a_{j_2} \left( \left\langle w_{j_1}^{(t)}, w_{j_2}^{(t)} \right\rangle - \left\langle w_{j_1}^{(0)}, w_{j_2}^{(0)} \right\rangle \right) \right| \cdot \max_{i,l} p_{q \leftarrow l, k \leftarrow \mu_1}^{(t,i)}}_{(2)}.$$

For term $(1)$, by Proposition E.5, we have

$$\left| \frac{1}{n} \sum_{i:\mu_1 \in X^{(i)}} g_i^{(t)} y_i \sum_{l=1}^{L} \sum_{j_2 \neq j_1} a_{j_2} \left\langle w_{j_1}^{(0)}, w_{j_2}^{(0)} \right\rangle \right| \leq \frac{|\{i : \mu_1 \in X^{(i)}\}|}{n} \widetilde{O}\left( \sigma_1^2 \sqrt{mm_1} L \right).$$

For term $(2)$, note that

$$\left| \frac{1}{n} \sum_{i:\mu_1 \in X^{(i)}} g_i^{(t)} y_i \sum_{l=1}^{L} \sum_{j_2 \neq j_1} a_{j_2} \left( \left\langle w_{j_1}^{(t)}, w_{j_2}^{(t)} \right\rangle - \left\langle w_{j_1}^{(0)}, w_{j_2}^{(0)} \right\rangle \right) \right|$$

$$\leq \frac{|\{i : \mu_1 \in X^{(i)}\}|}{n} L m_1 \cdot O\left( \frac{1}{m} \right),$$

where the inequality is by Lemma D.3, Proposition E.12.

Finally, since $p_{q \leftarrow l, k \leftarrow \mu_1}^{(t,i)} \leq \frac{1}{L} + \frac{1}{L^2} + R_P$, combining the upper bound for both terms $(1)$ and $(2)$, we have

$$\left| \frac{\alpha}{n} \sum_{i:\mu_1 \in X^{(i)}} g_i^{(t)} y_i \sum_{l=1}^{L} \sum_{j_2 \neq j_1} a_{j_2} \dot{\sigma}_{i,l,j_2}^{(t)} \left\langle w_{j_1}^{(t)}, w_{j_2}^{(t)} \right\rangle p_{q \leftarrow l, k \leftarrow \mu_1}^{(t,i)} \right|$$

$$\leq \alpha \frac{|\{i : \mu_1 \in X^{(i)}\}|}{n} \cdot \widetilde{O}\left( \sigma_1^2 \sqrt{mm_1} L \right).$$

$\square$

**Proposition E.12** (Neuron correlation change, Phase 1)**.** *For $t \leq T_1$, for all $j_1, j_2 \in [m_1]$, we have*

$$\left| \frac{\partial \left\langle w_{j_1}^{(t)}, w_{j_2}^{(t)} \right\rangle}{\partial t} \right| \leq 2L \left( \sigma_1 \sigma_0 \sqrt{\frac{m}{L} \log \frac{m_1 nL}{\delta}} + \widetilde{O}(LR_P \sigma_0 \sigma_1 \sqrt{m}) + 4R/L \right),$$

*and thus,*

$$\left| \left\langle w_{j_1}^{(t)}, w_{j_2}^{(t)} \right\rangle - \left\langle w_{j_1}^{(0)}, w_{j_2}^{(0)} \right\rangle \right| \leq t2L \left( \sigma_1 \sigma_0 \sqrt{\frac{m}{L} \log \frac{m_1 nL}{\delta}} + \widetilde{O}(LR_P \sigma_0 \sigma_1 \sqrt{m}) + 4R/L \right).$$

*Proof.* By the gradient flow update in Lemma C.2, we have

$$\frac{\partial \left\langle w_{j_1}^{(t)}, w_{j_2}^{(t)} \right\rangle}{\partial t} = \frac{1}{n} \sum_{i'=1}^{n} g_{i'}^{(t)} y_{i'} \sum_{l'=1}^{L} a_{j_2} w_{j_1}^{(t)\top} V^{(t,i')} p_{l'}^{(t,i')} + \frac{1}{n} \sum_{i'=1}^{n} g_{i'}^{(t)} y_{i'} \sum_{l'=1}^{L} a_{j_1} w_{j_2}^{(t)\top} V^{(t,i')} p_{l'}^{(t,i')}$$

$$= \frac{1}{n} \sum_{i'=1}^{n} g_{i'}^{(t)} y_{i'} \sum_{l'=1}^{L} a_{j_2} \left( w_{j_1}^{(0)\top} V^{(0,i')} p_{l'}^{(0,i')} + (w_{j_1}^{(t)\top} V^{(t,i')} p_{l'}^{(t,i')} - w_{j_1}^{(0)\top} V^{(0,i')} p_{l'}^{(0,i')}) \right)$$

$$+ \frac{1}{n} \sum_{i'=1}^{n} g_{i'}^{(t)} y_{i'} \sum_{l'=1}^{L} a_{j_1} \left( w_{j_2}^{(0)\top} V^{(0,i')} p_{l'}^{(0,i')} + (w_{j_2}^{(t)\top} V^{(t,i')} p_{l'}^{(t,i')} - w_{j_2}^{(0)\top} V^{(0,i')} p_{l'}^{(0,i')}) \right).$$

Now, since by Lemma E.7 we have

$$\left| w_j^{(t)} W_V^{(t)} X^{(i)} p_l^{(t,i)} - w_j^{(0)} W_V^{(0)} X^{(i)} p_l^{(0,i)} \right| \leq \widetilde{O}(LR_P \sigma_0 \sigma_1 \sqrt{m}) + 4R/L,$$

by Definition E.2 and Corollary D.5, we have

$$\left| \frac{1}{n} \sum_{i'=1}^{n} g_{i'}^{(t)} y_{i'} \sum_{l'=1}^{L} a_{j_2} \left( w_{j_1}^{(0)\top} V^{(0,i')} p_{l'}^{(0,i')} + (w_{j_1}^{(t)\top} V^{(t,i')} p_{l'}^{(t,i')} - w_{j_1}^{(0)\top} V^{(0,i')} p_{l'}^{(0,i')}) \right) \right|$$

$$\leq L \left( \sigma_1 \sigma_0 \sqrt{\frac{m}{L} \log \frac{m_1 nL}{\delta}} + \widetilde{O}(LR_P \sigma_0 \sigma_1 \sqrt{m}) + 4R/L \right)$$

$$\left| \frac{1}{n} \sum_{i'=1}^{n} g_{i'}^{(t)} y_{i'} \sum_{l'=1}^{L} a_{j_1} \left( w_{j_2}^{(0)\top} V^{(0,i')} p_{l'}^{(0,i')} + (w_{j_2}^{(t)\top} V^{(t,i')} p_{l'}^{(t,i')} - w_{j_2}^{(0)\top} V^{(0,i')} p_{l'}^{(0,i')}) \right) \right|$$

$$\leq L \left( \sigma_1 \sigma_0 \sqrt{\frac{m}{L} \log \frac{m_1 nL}{\delta}} + \widetilde{O}(LR_P \sigma_0 \sigma_1 \sqrt{m}) + 4R/L \right).$$

Thus,

$$\left| \left\langle w_{j_1}^{(t)}, w_{j_2}^{(t)} \right\rangle - \left\langle w_{j_1}^{(0)}, w_{j_2}^{(0)} \right\rangle \right|$$

$$\leq \int_{\tau=0}^{t} \left| \frac{\partial \left\langle w_{j_1}^{(\tau)}, w_{j_2}^{(\tau)} \right\rangle}{\partial \tau} \right| \leq t2L \left( \sigma_1 \sigma_0 \sqrt{\frac{m}{L} \log \frac{m_1 nL}{\delta}} + \widetilde{O}(LR_P \sigma_0 \sigma_1 \sqrt{m}) + 4R/L \right).$$

$\square$

**Corollary E.13** (Neuron Norm Change, Phase 1). *For all $t \leq T_1$ and all $j \in [m]$, we have*

$$\left| \|w_j^{(t)}\|_2^2 - \|w_j^{(0)}\|_2^2 \right| \leq t2L(\sigma_1 \sigma_0 \sqrt{\frac{m}{L} \log \frac{m_1 nL}{\delta}} + \widetilde{O}(LR_P \sigma_0 \sigma_1 \sqrt{m}) + 4R/L).$$

### E.5  Neuron Weights Align with Signal Value

**Theorem E.14** (Signal correlation growth, phase 1). *For $t \leq T_1$, for $\mu \in \{\mu_1, \mu_2\}$,*

$$\frac{\partial a_j w_j^{(t)} W_V^{(t)} \mu}{\partial t} = \frac{1}{n} \left( \sum_{\substack{i:\mu \in X^{(i)}, \\ y_i=1}} g_i^{(t)} \sum_{l=1}^{L} p_{q \leftarrow l, k \leftarrow \mu}^{(t,i)} - \sum_{\substack{i:\mu \in X^{(i)}, \\ y_i=-1}} g_i^{(t)} \sum_{l=1}^{L} p_{q \leftarrow l, k \leftarrow \mu}^{(t,i)} \right) \Theta((\sigma_0^2 + \sigma_1^2)m) + \varepsilon$$

*where*

$$|\varepsilon| \leq \widetilde{O}(L\sigma_0^2 \sqrt{m} + \sigma_1^2 \sqrt{mm_1}).$$

*Proof.* We take $\mu = \mu_1$ and the proof is similar for $\mu = \mu_2$. By Lemma C.1, we have

$$\frac{\partial a_j w_j^{(t)} W_V^{(t)} \mu}{\partial t}$$

$$= \frac{1}{n} \sum_{i:\mu_1 \in X^{(i)}, y_i=1} g_i^{(t)} \sum_{l=1}^{L} \left( \|w_j^{(t)}\|_2^2 + \|v^{(t)}(\mu_1)\|_2^2 \right) p_{q \leftarrow l, k \leftarrow \mu_1}^{(t,i)}$$

$$- \frac{1}{n} \sum_{i:\mu_1 \in X^{(i)}, y_i=-1} g_i^{(t)} \sum_{l=1}^{L} \left( \|w_j^{(t)}\|_2^2 + \|v^{(t)}(\mu_1)\|_2^2 \right) p_{q \leftarrow l, k \leftarrow \mu_1}^{(t,i)} + \varepsilon$$

$$= \left( \|w_j^{(t)}\|_2^2 + \|v^{(t)}(\mu_1)\|_2^2 \right) \frac{1}{n} \left( \sum_{\substack{i:\mu_1 \in X^{(i)}, \\ y_i=1}} g_i^{(t)} \sum_{l=1}^{L} p_{q \leftarrow l, k \leftarrow \mu_1}^{(t,i)} - \sum_{\substack{i:\mu_1 \in X^{(i)}, \\ y_i=-1}} g_i^{(t)} \sum_{l=1}^{L} p_{q \leftarrow l, k \leftarrow \mu_1}^{(t,i)} \right) + \varepsilon$$

where

$$\varepsilon = \frac{1}{n} \sum_{i:\mu_1 \in X^{(i)}} g_i^{(t)} y_i \sum_{l=1}^{L} \sum_{j_2 \neq j_1} a_{j_2} \left\langle w_{j_1}^{(t)}, w_{j_2}^{(t)} \right\rangle p_{q \leftarrow l, k \leftarrow \mu_1}^{(t,i)}$$

$$+ \frac{1}{n} \sum_{i=1}^{n} g_i^{(t)} y_i \sum_{l_1=1}^{L} a_{j_1} \sum_{l_2=1}^{L} \left\langle p_{l_1, l_2}^{(t,i)} v_{l_2}^{(t,i)}, v^{(t)}(\mu_1) \right\rangle \mathbb{I}(v_{l_2}^{(t,i)} \neq v^{(t)}(\mu_1)).$$

We can now bound the magnitude of $\varepsilon$ by Corollary E.10, Lemma E.11 and Proposition E.5 and obtain:

$$|\varepsilon| \leq L \cdot \widetilde{O} \left( \sigma_0^2 \sqrt{m} \right) + \widetilde{O}(\sigma_1^2 \sqrt{m m_1}).$$

Finally, by Lemma D.3 and Corollary E.13, we have $\|w_j^{(t)}\|_2^2 = \Theta(\sigma_1^2 m)$ and by Lemma E.9, we have $\|v^{(t)}(\mu_1)\|_2^2 = \Theta(\sigma_0^2 m)$. The proof is completed. $\square$

**Theorem E.15** (Random token growth, Phase 1). *For $t \leq T_1$, for $\mu \in \mathcal{R}$, we have*

$$\frac{\partial a_j w_j^{(t)} W_V^{(t)} \mu}{\partial t} = O\left( \frac{1}{n}(\sigma_0^2 + \sigma_1^2)m \right) + \widetilde{O}(\sigma_0^2 L \sqrt{m}).$$

*Proof.* Fix a $\mu \in \mathcal{R}$. By our Assumption 2.3, $\mu$ appears at most once in the training data set. Now, assume $\mu$ is in the training set and let $i^\star$ be the index of the sample containing $\mu$. Applying Lemma C.1 on the random token $\mu$, we have

$$\frac{\partial a_j w_j^{(t)} W_V^{(t)} \mu}{\partial t} = \frac{1}{n} g_{i^\star}^{(t)} y_{i^\star} \sum_{l=1}^{L} a_j \left( \|w_j^{(t)}\|_2^2 + \|v^{(t)}(\mu)\|_2^2 \right) p_{q \leftarrow l, k \leftarrow \mu}^{(t,i^\star)} + \varepsilon,$$

where

$$\varepsilon = \frac{1}{n} g_{i^\star}^{(t)} y_{i^\star} \sum_{l=1}^{L} \sum_{j_2 \neq j_1} a_{j_2} \left\langle w_{j_1}^{(t)}, w_{j_2}^{(t)} \right\rangle p_{q \leftarrow l, k \leftarrow \mu}^{(t,i^\star)}$$

$$+ \frac{1}{n} \sum_{i=1}^{n} g^{(t)} y_i \sum_{l_1=1}^{L} a_{j_1} \sum_{l_2=1}^{L} \left\langle p_{l_1, l_2}^{(t,i)} v_{l_2}^{(t,i)}, v^{(t)}(\mu) \right\rangle \mathbb{I}(v_{l_2}^{(t,i)} \neq v^{(t)}(\mu)).$$

We can now bound the magnitude of $\varepsilon$ by Corollary E.10 and Lemma E.11 as follows:

$$|\varepsilon| \leq L \cdot \widetilde{O} \left( \sigma_0^2 \sqrt{m} \right) + \frac{1}{n} \widetilde{O}(\sigma_1^2 \sqrt{m m_1}).$$

Finally, by Lemma D.3 and Corollary E.13, we have

$$\frac{\partial a_j w_j^{(t)} W_V^{(t)} \mu}{\partial t} = O\left( \frac{1}{n}(\sigma_0^2 + \sigma_1^2)m \right) + \widetilde{O}(\sigma_0^2 L \sqrt{m}).$$

$\square$

### E.6 Alignment of Common Token

**Lemma E.16** (Initial Per Neuron Gradient of Common Token, same as Lemma 4.2). *Let $F = \max_i F_i^{(0)}$. We have*

$$\left| \frac{\partial a_j w_j^{(0)} W_V^{(0)} \mu_3}{\partial t} \right| = \widetilde{O}\left( \sigma_1^2 \sqrt{mm_1} + \sigma_0^2 \sqrt{mL} + (\sigma_0^2 + \sigma_1^2)mF \right).$$

*Proof.* Following from the proof of Lemma E.4, we have

$$\frac{\partial a_{j_1} w_{j_1}^{(0)} W_V^{(0)} \mu_3}{\partial t} = \frac{1}{n} \sum_{i:\mu_3 \in X^{(i)}} g_i^{(0)} y_i \sum_{l=1}^{L} \left( \|w_{j_1}^{(0)}\|_2^2 + \|v^{(0)}(\mu_1)\|_2^2 \right) p_{q \leftarrow l, k \leftarrow \mu_3}^{(0,i)}$$

$$+ a_{j_1} \underbrace{\frac{1}{n} \sum_{i:\mu_3 \in X^{(i)}} g_i^{(0)} y_i \sum_{l=1}^{L} \sum_{j_2 \neq j_1} a_{j_2} \left\langle w_{j_1}^{(0)}, w_{j_2}^{(0)} \right\rangle p_{q \leftarrow l, k \leftarrow \mu_3}^{(0,i)}}_{\varepsilon_1}$$

$$+ a_{j_1} \underbrace{\frac{1}{n} \sum_{i=1}^{n} g_i^{(0)} y_i \sum_{l_1=1}^{L} a_{j_1} \sum_{l_2=1}^{L} \left\langle p_{l_1,l_2}^{(0,i)} v_{l_2}^{(0,i)}, v^{(0)}(\mu_3) \right\rangle \mathbb{I}(v_{l_2}^{(0,i)} \neq v^{(0)}(\mu_3))}_{\varepsilon_2}.$$

For the first term, we have

$$\left| \frac{1}{n} \sum_{i:\mu_3 \in X^{(i)}} g_i^{(0)} y_i \sum_{l=1}^{L} \left( \|w_{j_1}^{(0)}\|_2^2 + \|v^{(0)}(\mu_3)\|_2^2 \right) p_{q \leftarrow l, k \leftarrow \mu_3}^{(0,i)} \right|$$

$$= \left| \frac{1}{n} \sum_{y_i=1} g_i^{(0)} \sum_{l=1}^{L} \left( \|w_{j_1}^{(0)}\|_2^2 + \|v^{(0)}(\mu_3)\|_2^2 \right) p_{q \leftarrow l, k \leftarrow \mu_3}^{(0,i)} \right.$$

$$\left. - \frac{1}{n} \sum_{y_i=-1} g_i^{(0)} \sum_{l=1}^{L} \left( \|w_{j_1}^{(0)}\|_2^2 + \|v^{(0)}(\mu_3)\|_2^2 \right) p_{q \leftarrow l, k \leftarrow \mu_3}^{(0,i)} \right|$$

$$= \frac{1}{n} \left( \|w_{j_1}^{(0)}\|_2^2 + \|v^{(0)}(\mu_3)\|_2^2 \right) \left| \sum_{y_i=1} g_i^{(0)} \sum_{l=1}^{L} p_{q \leftarrow l, k \leftarrow \mu_3}^{(0,i)} - \sum_{y_i=-1} g_i^{(0)} \sum_{l=1}^{L} p_{q \leftarrow l, k \leftarrow \mu_3}^{(0,i)} \right|.$$

By Lemma D.8, and Corollary D.2, we have

$$\frac{1}{n} \left| \sum_{y_i=1} g_i^{(0)} \sum_{l=1}^{L} p_{q \leftarrow l, k \leftarrow \mu_3}^{(0,i)} - \sum_{y_i=-1} g_i^{(0)} \sum_{l=1}^{L} p_{q \leftarrow l, k \leftarrow \mu_3}^{(0,i)} \right|$$

$$\leq \frac{1}{n} \left( \sum_{y_i=1} \left( \frac{1}{2} + F \right) \sum_{l=1}^{L} \left( \frac{1}{L} + \frac{1}{L^2} \right) - \sum_{y_i=-1} \left( \frac{1}{2} - F \right) \sum_{l=1}^{L} \left( \frac{1}{L} - \frac{1}{L^2} \right) \right)$$

$$\leq \left( \frac{1}{2} + F \right) \left( \frac{1}{L} + \frac{1}{Lm} \right) \frac{L}{2} - \left( \frac{1}{2} - F \right) \left( \frac{1}{L} - \frac{1}{Lm} \right) \frac{L}{2}$$

$$\leq 2F + O(1/m).$$

By Lemma D.3, we have $\|w_{j_1}^{(0)}\|_2^2 = \Theta(\sigma_1^2 m)$ and $\|v^{(0)}(\mu_3)\|_2^2 = \Theta(\sigma_0^2 m)$, and thus,

$$\left| \frac{1}{n} \sum_{i:\mu_3 \in X^{(i)}} g_i^{(0)} y_i \sum_{l=1}^{L} a_{j_1} \left( \|w_{j_1}^{(0)}\|_2^2 + \|v^{(0)}(\mu_3)\|_2^2 \right) p_{q \leftarrow l, k \leftarrow \mu_3}^{(0,i)} \right| \leq \widetilde{O}(\alpha(\sigma_0^2 + \sigma_1^2)mF).$$

Further, by Proposition E.5, we have

$$|\varepsilon_1| = \left| \frac{1}{n} \sum_{i:\mu_3 \in X^{(i)}} g_i^{(0)} y_i \sum_{l=1}^{L} \sum_{j_2 \neq j_1} a_{j_2} \left\langle w_{j_1}^{(0)}, w_{j_2}^{(0)} \right\rangle p_{q \leftarrow l, k \leftarrow \mu_3}^{(0,i)} \right|$$

$$\leq \sigma_1^2 \sqrt{m_1} m \left( \sqrt{\frac{4}{m} \log \frac{2m_1^2}{\delta}} + \frac{4}{m} \log \frac{2m_1^2}{\delta} \right) \sqrt{\log \frac{m_1}{\delta}}.$$

$$|\varepsilon_2| = \left| \frac{1}{n} \sum_{i=1}^{n} g_i^{(0)} y_i \sum_{l_1=1}^{L} a_{j_1} \sum_{l_2=1}^{L} \left\langle p_{l_1,l_2}^{(0,i)} v_{l_2}^{(0,i)}, v^{(0)}(\mu_3) \right\rangle \mathbb{I}(v_{l_2}^{(0,i)} \neq v^{(0)}(\mu_3)) \right|$$

$$\leq \sigma_0^2 m \sqrt{L} \left( \sqrt{\frac{4}{m} \log \frac{2nL}{\delta}} + \frac{4}{m} \log \frac{2nL}{\delta} \right).$$

Finally, we have

$$\left| \frac{\partial a_{j_1} w_{j_1}^{(0)} W_V^{(0)} \mu_3}{\partial t} \right| \leq \widetilde{O} \left( \sigma_1^2 \sqrt{mm_1} + \sigma_0^2 \sqrt{mL} + (\sigma_0^2 + \sigma_1^2) mF \right).$$

$\square$

**Lemma E.17** (Per neuron gradient of common token). *For $t \leq T_1$, we have*

$$\left| \frac{\partial a_j w_j^{(0)} W_V^{(0)} \mu_3}{\partial t} \right| \leq \widetilde{O}(\sigma_1^2 m)$$

*Proof.* The proof is similar to that for Theorem E.14 and we omit it here. $\square$

**Definition E.18** (sub-network). *Define the sub-network structure as*

$$G(\mu) = \sum_{j=1}^{m_1} a_j w_j W_V \mu.$$

We can compute the gradient of the sub-network as

$$\frac{\partial G(\mu)}{\partial t} = \sum_{j=1}^{m_1} \frac{\partial a_j w_j^{(t)} W_V^{(t)} \mu}{\partial t} = \sum_{j=1}^{m_1} a_j w_j^{(t)} \frac{\partial W_V^{(t)} \mu}{\partial t} + a_j \frac{\partial w_j^{(t)}}{\partial t} W_V^{(t)} \mu$$

$$= \frac{1}{n} \sum_{i_2: \mu \in X^{(i_2)}} g_{i_2}^{(t)} y_{i_2} \sum_{l_2=1}^{L} \sum_{j_1=1}^{m_1} \sum_{j_2=1}^{m_1} a_{j_1} a_{j_2} \left\langle w_{j_1}^{(t)}, w_{j_2}^{(t)} \right\rangle p_{q \leftarrow l_2, k \leftarrow \mu}^{(t,i_2)}$$

$$+ \frac{1}{n} \sum_{i_1=1}^{n} g_{i_1}^{(t)} y_{i_1} m_1 \sum_{l_1=1}^{L} \sum_{l_2=1}^{L} \left\langle p_{l_1,l_2}^{(t,i_1)} v_{l_2}^{(t,i_1)}, v^{(t)}(\mu) \right\rangle.$$

**Theorem E.19** (Complete version of Lemma 4.3). *There exists a $T_{0.5} \leq T_1$ and a constant $0 < C < 1$ such that for all $T_{0.5} \leq t \leq T_1$,*

$$(1 + C) \max \left( \frac{\partial G^{(t)}(\mu_1)}{\partial t}, \frac{\partial G^{(t)}(\mu_2)}{\partial t} \right) \leq -\frac{\partial G^{(t)}(\mu_3)}{\partial t} \leq (1 - C) \left( \frac{\partial G^{(t)}(\mu_1)}{\partial t} + \frac{\partial G^{(t)}(\mu_2)}{\partial t} \right).$$

*Further,*

$$\frac{\partial G^{(t)}(\mu_1)}{\partial t}, \frac{\partial G^{(t)}(\mu_2)}{\partial t} \geq \Omega((\sigma_0^2 + \sigma_1^2) mm_1).$$

*Proof.* First of all, we have

$$\frac{\partial F_i^{(t)}}{\partial t} = \sum_{l_1=1}^{L} \sum_{j=1}^{m_1} \sum_{l_2=1}^{L} \left( \frac{\partial a_j w_j^{(t)} W_V^{(t)} X_{l_2}}{\partial t} p_{l_1,l_2}^{(t,i)} + a_j w_j^{(t)} W_V^{(t)} X_{l_2} \frac{\partial p_{l_1,l_2}^{(t,i)}}{\partial t} \right).$$

By Lemma E.4 and Lemma E.16, we have

$$\frac{\partial a_j w_j^{(0)} W_V^{(0)} \mu_1}{\partial t} - \frac{\partial a_j w_j^{(0)} W_V^{(0)} \mu_3}{\partial t} = \Theta((\sigma_0^2 + \sigma_1^2) m),$$

$$\frac{\partial a_j w_j^{(0)} W_V^{(0)} \mu_2}{\partial t} - \frac{\partial a_j w_j^{(0)} W_V^{(0)} \mu_3}{\partial t} = \Theta((\sigma_0^2 + \sigma_1^2)m).$$

By Theorem E.15 and Corollary E.24, we have

$$\sum_{l_1=1}^{L} \sum_{j=1}^{m_1} \sum_{l_2=1}^{L} \left( \frac{\partial a_j w_j^{(t)} W_V^{(t)} X_{l_2}}{\partial t} \mathbb{I}(X_{l_2} \notin \{\mu_i\}_{i=1}^3) p_{l_1,l_2}^{(t,i)} + a_j w_j^{(t)} W_V^{(t)} X_{l_2} \frac{\partial p_{l_1,l_2}^{(t,i)}}{\partial t} \right)$$

$$= O\left( \frac{L}{n}(\sigma_0^2 + \sigma_1^2)mm_1 \right).$$

Thus, for all $i \in I_1 \cup I_2 \cup I_3$, $\frac{\partial F_i^{(0)}}{\partial t} > 0$, which implies $\frac{\partial g_i^{(0)}}{\partial t} < 0$ for $i \in I_1$ and $\frac{\partial g_i^{(0)}}{\partial t} > 0$ for $i \in I_2 \cup I_3$.

Next, we show that there must exist a time such that $-\frac{\partial G^{(t)}(\mu_3)}{\partial t} = \max(\frac{\partial G^{(t)}(\mu_1)}{\partial t}, \frac{\partial G^{(t)}(\mu_2)}{\partial t})$. Without loss of generality, assume $\frac{\partial G^{(t)}(\mu_1)}{\partial t} > \frac{\partial G^{(t)}(\mu_2)}{\partial t}$. We now analyze the condition when the magnitude of the update of the common token is less than the magnitude of the update of the signals. By Lemma C.1, we have

$$\sum_{j=1}^{m_1} \frac{\partial a_j w_j^{(t)} W_V^{(t)} \mu_1}{\partial t} \geq -\sum_{j=1}^{m_1} \frac{\partial a_j w_j^{(t)} W_V^{(t)} \mu_3}{\partial t}$$

$$\Leftrightarrow \frac{1}{n} \left( \|W^{(t)}\|_F^2 + m_1 \|v^{(t)}(\mu_1)\|_2^2 \right) \left( \sum_{i \in I_1} g_i^{(t)} \sum_{l=1}^{L} p_{q \leftarrow l, k \leftarrow \mu_1}^{(t,i)} - \sum_{i \in I_2} g_i^{(t)} \sum_{l=1}^{L} p_{q \leftarrow l, k \leftarrow \mu_1}^{(t,i)} \right) + \sum_{j=1}^{m_1} a_j \varepsilon^{(t)}(\mu_1)$$

$$\geq \frac{1}{n} \left( \|W^{(t)}\|_F^2 + m_1 \|v^{(t)}(\mu_3)\|_2^2 \right) \left( \sum_{i \in I_2 \cup I_3 \cup I_4} g_i^{(t)} \sum_{l=1}^{L} p_{q \leftarrow l, k \leftarrow \mu_3}^{(t,i)} - \sum_{i \in I_1} g_i^{(t)} \sum_{l=1}^{L} p_{q \leftarrow l, k \leftarrow \mu_3}^{(t,i)} \right) + \sum_{j=1}^{m_1} a_j \varepsilon^{(t)}(\mu_3)$$

$$\Leftrightarrow \frac{1}{n} \sum_{i \in I_1} g_i^{(t)} \sum_{l=1}^{L} p_{q \leftarrow l, k \leftarrow \mu_1}^{(t,i)} + \frac{1}{n} \frac{\|W^{(t)}\|_F^2 + m_1 \|v^{(t)}(\mu_3)\|_2^2}{\|W^{(t)}\|_F^2 + m_1 \|v^{(t)}(\mu_1)\|_2^2} \sum_{i \in I_1} g_i^{(t)} \sum_{l=1}^{L} p_{q \leftarrow l, k \leftarrow \mu_3}^{(t,i)}$$

$$\geq \frac{1}{n} \frac{\|W^{(t)}\|_F^2 + m_1 \|v^{(t)}(\mu_3)\|_2^2}{\|W^{(t)}\|_F^2 + m_1 \|v^{(t)}(\mu_1)\|_2^2} \sum_{i \in I_2 \cup I_3 \cup I_4} g_i^{(t)} \sum_{l=1}^{L} p_{q \leftarrow l, k \leftarrow \mu_3}^{(t,i)} + \frac{1}{n} \sum_{i \in I_2} g_i^{(t)} \sum_{l=1}^{L} p_{q \leftarrow l, k \leftarrow \mu_1}^{(t,i)}$$

$$+ \frac{\sum_{j=1}^{m_1} a_j \varepsilon^{(t)}(\mu_3)}{\|W^{(t)}\|_F^2 + m_1 \|v^{(t)}(\mu_1)\|_2^2} - \frac{\sum_{j=1}^{m_1} a_j \varepsilon^{(t)}(\mu_1)}{\|W^{(t)}\|_F^2 + m_1 \|v^{(t)}(\mu_1)\|_2^2}$$

$$\Leftrightarrow \frac{1}{n}(1 + o(1)) \sum_{i \in I_1} g_i^{(t)} + \frac{1}{n}(1 + o(1)) \sum_{i \in I_1} g_i^{(t)}$$

$$\geq \frac{1}{n}(1 + o(1)) \sum_{i \in I_2 \cup I_3 \cup I_4} g_i^{(t)} + \frac{1}{n}(1 + o(1)) \sum_{i \in I_2} g_i^{(t)} \pm O\left( \sqrt{\frac{m_1}{m}} \right)$$

where the last equality applies Corollary E.24. If $m$ is sufficiently larger than $m_1$ (by some large constant factor $C$), then the last term is negligible. Note that the above implies that if $\frac{\partial G^{(t)}(\mu_1)}{\partial t} \geq \frac{\partial G^{(t)}(\mu_3)}{\partial t}$ then $\sum_{i \in I_1} g_i^{(t)} - \sum_{i \in I_2} g_i^{(t)} = \Omega(1)$ since $\sum_{i \in I_3 \cup I_4} g_i^{(t)} = \Omega(1)$ for all $t \leq T_1$. Now, by Proposition E.21, if the initialization level is small enough, then $\sum_{i \in I_1} g_i^{(t)} - \sum_{i \in I_3} g_i^{(t)} \geq \Omega(1)$. Thus, if $\frac{\partial G^{(t)}(\mu_1)}{\partial t} \geq -\frac{\partial G^{(t)}(\mu_3)}{\partial t}$, then $\frac{1}{n} \sum_{i \in I_1} \frac{\partial g_i^{(t)}}{\partial t} = -\Theta((\sigma_0^2 + \sigma_1^2)mm_1)$. Therefore, there must exist a time $t' = \Theta(1/m)$ such that $\frac{\partial G^{(t')}(\mu_1)}{\partial t} = -\frac{\partial G^{(t')}(\mu_3)}{\partial t}$. Further, for $t \geq \Theta(F/m)$, we have $y_i F_i^{(t)} > 0$ for all $i \in I_4$ and $\sum_{i \in I_4} g_i^{(t)} < \min(\sum_{i \in I_2} g_i^{(t)}, \sum_{i \in I_3} g_i^{(t)})$, where $F = \max_{i \in [n]} |F_i^{(0)}|$.

Now, consider a time point $t'$ when $\frac{\partial G^{(t')}(\mu_1)}{\partial t} = -\frac{\partial G^{(t')}(\mu_3)}{\partial t}$. At $t'$, we must have $\min(\sum_{i \in I_2} g_i^{(t')}, \sum_{i \in I_3} g_i^{(t')}) - \sum_{i \in I_4} g_i^{(t')} = \Omega(1)$. Thus,

$$2 \sum_{i \in I_1} g_i^{(t')} - 2 \sum_{i \in I_4} g_i^{(t')} = \Omega(1),$$

which implies

$$\frac{\partial}{\partial F} 2 \sum_{i \in I_1} g_i^{(t')} - \frac{\partial}{\partial F} \sum_{i \in I_4} g_i^{(t')} = \Omega(1).$$

For $i \in I_1$, we have

$$\frac{\partial F_i^{(t')}}{\partial t} = \frac{\partial G^{(t')}(\mu_1)}{\partial t} + \frac{\partial G^{(t')}(\mu_2)}{\partial t} + \frac{\partial G^{(t')}(\mu_3)}{\partial t} + O\left(\frac{L}{n}\sigma_1^2 m m_1\right)$$

$$= \frac{\partial G^{(t')}(\mu_2)}{\partial t} + O\left(\frac{L}{n}\sigma_1^2 m m_1\right).$$

For $i \in I_2$,

$$\frac{\partial F_i^{(t)}}{\partial t} = \frac{\partial G^{(t')}(\mu_1)}{\partial t} + \frac{\partial G^{(t')}(\mu_3)}{\partial t} + O\left(\frac{L}{n}\sigma_1^2 m m_1\right) = O\left(\frac{L}{n}\sigma_1^2 m m_1\right).$$

For $i \in I_3$, by Proposition E.21,

$$\frac{\partial F_i^{(t)}}{\partial t} = \frac{\partial G^{(t')}(\mu_2)}{\partial t} + \frac{\partial G^{(t')}(\mu_3)}{\partial t} + O\left(\frac{L}{n}\sigma_1^2 m m_1\right) = O\left(F\sigma_1^2 m m_1\right),$$

and for $i \in I_4$,

$$\frac{\partial F_i^{(t)}}{\partial t} = \frac{\partial G^{(t')}(\mu_3)}{\partial t} + O\left(\frac{L}{n}\sigma_1^2 m m_1\right).$$

Thus, by chain rule $\frac{\partial g_i^{(t)}}{\partial t} = \frac{\partial g_i}{\partial F_i} \frac{\partial F_i^{(t)}}{\partial t}$, we have

$$\frac{\partial}{\partial t}\left(2 \sum_{i \in I_1} g_i^{(t')} - 2 \sum_{i \in I_2} g_i^{(t')} - \sum_{i \in I_3 \cup I_4} g_i^{(t')}\right) = -\Theta(\sigma_1^2 m m_1). \tag{8}$$

This implies that there exists a constant $L$ such that

$$\frac{\partial}{\partial t}\left(\sum_{i \in I_1} g_i^{(t)} - \sum_{i \in I_2} g_i^{(t)} + (1 + L)\left(\sum_{i \in I_1} g_i^{(t)} - \sum_{i \in I_2 \cup I_3 \cup I_4} g_i^{(t)}\right)\right) = -\Theta(\sigma_1^2 m m_1),$$

which implies that there must exist a time $T_{0.5} \leq O(1/m)$ such that for all $t \geq T_{0.5}$,

$$-\frac{\partial G^{(t)}(\mu_3)}{\partial t} \geq (1 + L)\frac{\partial G^{(t)}(\mu_1)}{\partial t}.$$

Moreover, by Theorem E.26, if $C_{T_1}$ is sufficiently large, we have $T_{0.5} < T_1$.

Finally, consider the time when $\frac{\partial G^{(t)}(\mu_1)}{\partial t} + \frac{\partial G^{(t)}(\mu_2)}{\partial t} > -\frac{\partial G^{(t)}(\mu_3)}{\partial t}$. During this time, note that we have

$$\frac{\partial G^{(t)}(\mu_1)}{\partial t}, \frac{\partial G^{(t)}(\mu_2)}{\partial t} \geq \Theta(\sigma_1^2 m m_1).$$

Further, if $\frac{\partial G^{(t)}(\mu_1)}{\partial t} + \frac{\partial G^{(t)}(\mu_2)}{\partial t} = -\frac{\partial G^{(t)}(\mu_3)}{\partial t}$, then $\frac{\partial}{\partial t} g_i^{(t)} = -\Theta(\sigma_1^2 m m_1)$ for all $i \in I_2 \cup I_3 \cup I_4$. Thus, there must exist a constant $U$ such that

$$-\frac{\partial G^{(t)}(\mu_3)}{\partial t} \leq (1 - U)\left(\frac{\partial G^{(t)}(\mu_1)}{\partial t} + \frac{\partial G^{(t)}(\mu_2)}{\partial t}\right)$$

for all $t \leq T_1$. $\qquad\square$

**Corollary E.20.** *There exists a small positive constant $C$ such that if we define $T' := \min\{t : \min_{i \in I_2 \cup I_3} F_i^{(t)} \geq C\}$, then $T' \leq O(1/m)$.*

**Proposition E.21.** *Let $F = \max_{i \in [n]} |F_i^{(0)}|$. For $t \leq T_1$, we have*

$$\left| \frac{1}{n} \sum_{i \in I_2} g_i^{(t)} - \frac{1}{n} \sum_{i \in I_3} g_i^{(t)} \right| \leq O\left(F\right).$$

*Proof.* First of all, by Lipschitzness of $g$, we have

$$\left| \frac{1}{n} \sum_{i \in I_2} g_i^{(t)} - \frac{1}{n} \sum_{i \in I_3} g_i^{(t)} \right| \leq |G^{(t)}(\mu_1) - G^{(0)}(\mu_1) - (G^{(t)}(\mu_2) - G^{(0)}(\mu_2))| + 2 \max_{i \in [n]} \left| F_i^{(0)} \right| + o(1).$$

Without loss of generality, assume $G^{(t)}(\mu_1) > G^{(t)}(\mu_2)$ for all $t \leq T_1$; otherwise, we can break the interval $[0, T_1]$ into sub-intervals by the time points when $G^{(t)}(\mu_1) - G^{(t)}(\mu_2)$ changes its sign and then apply the analysis below to each sub-interval. We first derive

$$\frac{\partial G^{(t)}(\mu_1)}{\partial t} - \frac{\partial G^{(t)}(\mu_2)}{\partial t}$$

$$= \sum_{j_1=1}^{m_1} \left( \frac{1}{n} \sum_{i_2 \in I_1 \cup I_2} g_{i_2}^{(t)} y_{i_2} \sum_{l_2=1}^{L} \sum_{j_2=1}^{m_1} a_{j_1} a_{j_2} \left\langle w_{j_1}^{(t)}, w_{j_2}^{(t)} \right\rangle p_{q \leftarrow l_2, k \leftarrow \mu_1}^{(t,i_2)} + \frac{1}{n} \sum_{i_1=1}^{n} g_{i_1}^{(t)} y_{i_1} \sum_{l_1=1}^{L} p_{l_1}^{(t,i_1)\top} V^{(t,i_1)\top} W_V^{(t)} \mu_1 \right.$$

$$\left. - \frac{1}{n} \sum_{i_2 \in I_1 \cup I_3} g_{i_2}^{(t)} y_{i_2} \sum_{l_2=1}^{L} \sum_{j_2=1}^{m_1} a_{j_1} a_{j_2} \left\langle w_{j_1}^{(t)}, w_{j_2}^{(t)} \right\rangle p_{q \leftarrow l_2, k \leftarrow \mu_2}^{(t,i_2)} - \frac{1}{n} \sum_{i_1=1}^{n} g_{i_1}^{(t)} y_{i_1} \sum_{l_1=1}^{L} p_{l_1}^{(t,i_1)\top} V^{(t,i_1)\top} W_V^{(t)} \mu_2 \right)$$

$$= \sum_{j_1=1}^{m_1} \sum_{j_2=1}^{m_1} a_{j_1} a_{j_2} \left\langle w_{j_1}^{(t)}, w_{j_2}^{(t)} \right\rangle \left( \frac{1}{n} \sum_{i \in I_1} g_i^{(t)} y_i \sum_{l=1}^{L} p_{q \leftarrow l, k \leftarrow \mu_1}^{(t,i)} - \frac{1}{n} \sum_{i \in I_1} g_i^{(t)} y_i \sum_{l=1}^{L} p_{q \leftarrow l, k \leftarrow \mu_2}^{(t,i)} \right)$$

$$+ \sum_{j_1=1}^{m_1} \sum_{j_2=1}^{m_1} a_{j_1} a_{j_2} \left\langle w_{j_1}^{(t)}, w_{j_2}^{(t)} \right\rangle \left( \frac{1}{n} \sum_{i \in I_2} g_i^{(t)} y_i \sum_{l=1}^{L} p_{q \leftarrow l, k \leftarrow \mu_1}^{(t,i)} - \frac{1}{n} \sum_{i \in I_3} g_i^{(t)} y_i \sum_{l=1}^{L} p_{q \leftarrow l, k \leftarrow \mu_2}^{(t,i)} \right)$$

$$+ m_1 \left( \frac{1}{n} \sum_{i_1=1}^{n} g_{i_1}^{(t)} y_{i_1} \sum_{l_1=1}^{L} p_{l_1}^{(t,i_1)\top} V^{(t,i_1)\top} W_V^{(t)} \mu_1 - \frac{1}{n} \sum_{i_1=1}^{n} g_{i_1}^{(t)} y_{i_1} \sum_{l_1=1}^{L} p_{l_1}^{(t,i_1)\top} V^{(t,i_1)\top} W_V^{(t)} \mu_2 \right)$$

$$= \sum_{j_1=1}^{m_1} \sum_{j_2=1}^{m_1} a_{j_1} a_{j_2} \left\langle w_{j_1}^{(t)}, w_{j_2}^{(t)} \right\rangle \left( \frac{1}{n} \sum_{i \in I_3} g_i^{(t)} - \frac{1}{n} \sum_{i \in I_2} g_i^{(t)} + o(1) \right)$$

$$+ m_1 \left( \frac{1}{n} \sum_{i_1=1}^{n} g_{i_1}^{(t)} y_{i_1} \sum_{l_1=1}^{L} p_{l_1}^{(t,i_1)\top} V^{(t,i_1)\top} W_V^{(t)} \mu_1 - \frac{1}{n} \sum_{i_1=1}^{n} g_{i_1}^{(t)} y_{i_1} \sum_{l_1=1}^{L} p_{l_1}^{(t,i_1)\top} V^{(t,i_1)\top} W_V^{(t)} \mu_2 \right)$$

$$\leq \sum_{j_1=1}^{m_1} \sum_{j_2=1}^{m_1} a_{j_1} a_{j_2} \left\langle w_{j_1}^{(t)}, w_{j_2}^{(t)} \right\rangle \left( G^{(t)}(\mu_1) - G^{(0)}(\mu_1) - (G^{(t)}(\mu_2) - G^{(0)}(\mu_2)) \right.$$

$$\left. + G^{(0)}(\mu_1) + G^{(0)}(\mu_2) + O\left(\max_{i \in [n]} |F_i^{(0)}|\right) + o(1) \right)$$

$$+ m_1 \left( \frac{1}{n} \sum_{i_1=1}^{n} g_{i_1}^{(t)} y_{i_1} \sum_{l_1=1}^{L} p_{l_1}^{(t,i_1)\top} V^{(t,i_1)\top} W_V^{(t)} \mu_1 - \frac{1}{n} \sum_{i_1=1}^{n} g_{i_1}^{(t)} y_{i_1} \sum_{l_1=1}^{L} p_{l_1}^{(t,i_1)\top} V^{(t,i_1)\top} W_V^{(t)} \mu_2 \right).$$

By Theorem E.14, we have

$$m_1 \int_0^T \frac{1}{n} \sum_{i_1=1}^{n} g_{i_1}^{(t)} y_{i_1} \sum_{l_1=1}^{L} a_{j_1} p_{l_1}^{(t,i_1)\top} V^{(t,i_1)\top} W_V^{(t)} \mu_1 - \frac{1}{n} \sum_{i_1=1}^{n} g_{i_1}^{(t)} y_{i_1} \sum_{l_1=1}^{L} a_{j_1} p_{l_1}^{(t,i_1)\top} V^{(t,i_1)\top} W_V^{(t)} \mu_2 \, dt$$

$$\leq T \cdot \widetilde{O}(L \sigma_0^2 \sqrt{m} m_1),$$

$$\int_0^T \left( G^{(0)}(\mu_1) + G^{(0)}(\mu_2) + O\left( \max_{i \in [n]} |F_i^{(0)}| \right) + o(1) \right) \sum_{j_1=1}^{m_1} \sum_{j_2=1}^{m_1} a_{j_1} a_{j_2} \left\langle w_{j_1}^{(t)}, w_{j_2}^{(t)} \right\rangle \, dt$$

$$\leq O\left( \left( G^{(0)}(\mu_1) + G^{(0)}(\mu_2) + \max_{i \in [n]} |F_i^{(0)}| \right) T \sigma_1^2 m m_1 \right),$$

and

$$\exp\left( \int_0^T \sum_{j_1=1}^{m_1} \sum_{j_2=1}^{m_1} a_{j_1} a_{j_2} \left\langle w_{j_1}^{(t)}, w_{j_2}^{(t)} \right\rangle \, dt \right) = O(1).$$

By Grönwall's inequality, we have

$$G^{(T)}(\mu_1) - G^{(0)}(\mu_1) - (G^{(T)}(\mu_2) - G^{(0)}(\mu_2))$$

$$\leq T \cdot \widetilde{O}(L\sigma_0^2 \sqrt{m} m_1) + O\left( \left( G^{(0)}(\mu_1) + G^{(0)}(\mu_2) + \max_{i \in [n]} |F_i^{(0)}| \right) T \sigma_1^2 m m_1 \right)$$

$$+ \int_0^T \left( t \cdot \widetilde{O}(L\sigma_0^2 \sqrt{m}) + O\left( \left( G^{(0)}(\mu_1) + G^{(0)}(\mu_2) + \max_{i \in [n]} |F_i^{(0)}| \right) t \sigma_1^2 m \right) \right) \cdot O(1) \, dt$$

$$\leq O\left( G^{(0)}(\mu_1) + G^{(0)}(\mu_2) + \max_{i \in [n]} |F_i^{(0)}| \right).$$

This implies that

$$\left| \frac{1}{n} \sum_{i \in I_2} g_i^{(t)} - \frac{1}{n} \sum_{i \in I_3} g_i^{(t)} \right| \leq O\left( G^{(0)}(\mu_1) + G^{(0)}(\mu_2) + \max_{i \in [n]} |F_i^{(0)}| \right).$$

Thus, if the initialization scale is sufficiently small, $\frac{1}{n} \sum_{i \in I_2} g_i^{(t)}$ and $\frac{1}{n} \sum_{i \in I_3} g_i^{(t)}$ are only differed by a small constant. $\qquad\square$

### E.7 Small Score Movement in Phase 1

**Proposition E.22.** *Conditioned on the success of Lemma D.1 and Lemma D.3, with probability at least $1 - \delta$ over the randomness of $a$, for all $i \in [n]$ and $\mu, \nu \in X^{(i)}$, we have*

$$\left| \sum_{j=1}^{m_1} a_j \|k^{(0)}(\mu)\|_2^2 w_j^{(0)\top} v^{(0)}(\nu) \right| \leq O\left( \sigma_0^2 m \sigma_1 \sigma_0 \sqrt{m m_1 \log \frac{m_1 d}{\delta}} \sqrt{\log \frac{nd}{\delta}} \right),$$

$$\left| \sum_{j=1}^{m_1} a_j \|q^{(0)}(\mu)\|_2^2 w_j^{(0)\top} v^{(0)}(\nu) \right| \leq O\left( \sigma_0^2 m \sigma_1 \sigma_0 \sqrt{m m_1 \log \frac{m_1 d}{\delta}} \sqrt{\log \frac{nd}{\delta}} \right)$$

*and for all $l, l' \in [L]$ with $K_l^{(0,i)} \neq k^{(0)}(\nu)$ and $q_l^{(0,i)} \neq Q^{(0)}(\nu)$, we have*

$$\left| \sum_{j=1}^{m_1} a_j \nu^\top W_K^{(0)\top} K_l^{(0,i)} V_{l'}^{(0,i)\top} w_j^{(0)} \right| \leq \widetilde{O}(\sigma_0^2 \sqrt{m} \sigma_0 \sigma_1 \sqrt{m m_1}),$$

$$\left| \sum_{j=1}^{m_1} a_j \nu^\top W_Q^{(0)\top} q_l^{(0,i)} V_{l'}^{(0,i)\top} w_j^{(0)} \right| \leq \widetilde{O}(\sigma_0^2 \sqrt{m} \sigma_0 \sigma_1 \sqrt{m m_1}).$$

*Proof.* Fix $i \in [n]$ and $\mu, \nu \in X^{(i)}$. Consider the randomness of $a$. By Lemma D.1, Corollary D.2 and Lemma D.3, $a_j \|k^{(0)}(\mu)\|_2^2 w_j^{(0)\top} v^{(0)}(\nu)$ is a sub-Gaussian random variable with variance proxy $O((\sigma_0^2 m \sigma_1 \sigma_0 \sqrt{m} \sqrt{\log(dm_1/\delta)})^2)$. Then the following inequality holds.

$$\left| \sum_{j=1}^{m_1} a_j \|k^{(0)}(\mu)\|_2^2 w_j^{(0)\top} v^{(0)}(\nu) \right| \leq O\left( \sigma_0^2 m \sigma_1 \sigma_0 \sqrt{m m_1 \log \frac{m_1 d}{\delta}} \sqrt{\log \frac{2}{\delta}} \right).$$

Finally, take a union bound over $i \in [n]$, $\mu, \nu \in \{\mu_i\}_{i=1}^d$. The analysis for the second term is similar.

Next, note that $a_j \nu^\top W_K^{(0)\top} K_l^{(0,i)} V_{l'}^{(0,i)\top} w_j^{(0)}$ is a sub-Gaussian random variable with variance proxy $\widetilde{O}((\sigma_0^2 \sqrt{m} \sigma_0 \sigma_1 \sqrt{m})^2)$. Thus,

$$\left| \sum_{j=1}^{m_1} a_j \nu^\top W_K^{(0)\top} K_l^{(0,i)} V_{l'}^{(0,i)\top} w_j^{(0)} \right| \leq \widetilde{O}(\sigma_0^2 \sqrt{m} \sigma_0 \sigma_1 \sqrt{mm_1}).$$

$\square$

**Lemma E.23** (Score change). *For all $t \leq T_1$, for all $\nu, \mu \in \{\mu_i\}_{i=1}^d$, we have*

$$\left| \frac{\partial \nu^\top W_K^{(t)\top} W_Q^{(t)} \mu}{\partial t} \right| \leq \frac{1}{\sqrt{m}} \frac{\left| \{i : \mu \in X^{(i)}\} \right| + \left| \{i : \nu \in X^{(i)}\} \right|}{n} \widetilde{O}\left( \frac{1}{L} + \frac{1}{\sqrt{m}} \right)$$

*and thus,*

$$\left| \nu^\top W_K^{(t)\top} W_Q^{(t)} \mu - \nu^\top W_K^{(0)\top} W_Q^{(0)} \mu \right| \leq t \frac{1}{\sqrt{m}} \frac{\left| \{i : \mu \in X^{(i)}\} \right| + \left| \{i : \nu \in X^{(i)}\} \right|}{n} \widetilde{O}\left( \frac{1}{L} + \frac{1}{\sqrt{m}} \right).$$

*Proof.* First of all, by Lemma C.4, we expand the per step gradient descent update as follows:

$$\frac{\partial \nu^\top W_K^{(t)\top} W_Q^{(t)} \mu}{\partial t}$$

$$= \underbrace{\frac{1}{n\sqrt{m}} \sum_{i:\mu,\nu \in X^{(i)}} g_i^{(t)} y_i \sum_{j=1}^{m_1} a_j \|k^{(t)}(\nu)\|_2^2 \left( v^{(t)\top}(\nu) w_j^{(t)} - w_j^{(t)\top} V^{(t,i)} p_{l(i,\mu)}^{(t,i)} \right) p_{q \leftarrow \mu, k \leftarrow \nu}^{(t,i)}}_{(1)}$$

$$+ \underbrace{\frac{1}{n\sqrt{m}} \sum_{i:\mu \in X^{(i)}} g_i^{(t)} y_i \sum_{j=1}^{m_1} a_j \sum_{l=1}^{L} \nu^\top W_K^{(t)\top} K_l^{(t,i)} \left( V_l^{(t,i)\top} w_j^{(t)} - w_j^{(t)\top} V^{(t,i)} p_{l(i,\mu)}^{(t,i)} \right) p_{q \leftarrow \mu, k \leftarrow l}^{(t,i)} \mathbb{I}(K_l^{(t,i)} \neq k^{(t)}(\nu))}_{(2)}$$

$$+ \underbrace{\frac{1}{n\sqrt{m}} \sum_{i:\nu,\mu \in X^{(i)}} g_i^{(t)} y_i \sum_{j=1}^{m_1} a_j \|q^{(t)}(\mu)\|_2^2 p_{q \leftarrow \mu, k \leftarrow \nu}^{(t,i)} \left( w_j^{(t)\top} v^{(t,i)}(\nu) - w_j^{(t)\top} V^{(t,i)} p_{l(i,\mu)}^{(t,i)} \right)}_{(3)}$$

$$+ \underbrace{\frac{1}{n\sqrt{m}} \sum_{i:\nu \in X^{(i)}} g_i^{(t)} y_i \sum_{l=1}^{L} \sum_{j=1}^{m_1} a_j \mu^\top W_Q^{(t)\top} q_l^{(t,i)} p_{q \leftarrow l, k \leftarrow \nu}^{(t,i)} \left( w_j^{(t)\top} v^{(t,i)}(\nu) - w_j^{(t)\top} V^{(t,i)} p_l^{(t,i)} \right) \mathbb{I}(q_l^{(t,i)} \neq q^{(t)}(\mu))}_{(4)}$$

**Analysis of** $(1)$: By triangle inequality, we have

$$|(1)| \leq \underbrace{\frac{1}{n\sqrt{m}} \left| \sum_{i:\mu,\nu \in X^{(i)}} g_i^{(t)} y_i \sum_{j=1}^{m_1} a_j \|k^{(t)}(\nu)\|_2^2 v^{(t)\top}(\nu) w_j^{(t)} p_{q \leftarrow \mu, k \leftarrow \nu}^{(t,i)} \right|}_{(a)}$$

$$+ \underbrace{\frac{1}{n\sqrt{m}} \left| \sum_{i:\mu,\nu \in X^{(i)}} g_i^{(t)} y_i \sum_{j=1}^{m_1} a_j \|k^{(t)}(\nu)\|_2^2 w_j^{(t)\top} V^{(t,i)} p_{l(i,\mu)}^{(t,i)} p_{q \leftarrow \mu, k \leftarrow \nu}^{(t,i)} \right|}_{(b)}.$$

To analyze $(a)$, since $(a + \varepsilon_1)(b + \varepsilon_2) = ab + a\varepsilon_2 + b\varepsilon_1 + \varepsilon_1\varepsilon_2$, we have

$$\left| \sum_{j=1}^{m_1} a_j \|k^{(t)}(\nu)\|_2^2 w_j^{(t)\top} v^{(t)}(\nu) p_{q \leftarrow \mu, k \leftarrow \nu}^{(t,i)} - \sum_{j=1}^{m_1} a_j \|k^{(0)}(\nu)\|_2^2 w_j^{(0)\top} v^{(0)}(\nu) p_{q \leftarrow \mu, k \leftarrow \nu}^{(0,i)} \right|$$

$$\leq \left| \sum_{j=1}^{m_1} a_j \|k^{(t)}(\nu)\|_2^2 w_j^{(t)\top} v^{(t)}(\nu) - \sum_{j=1}^{m_1} a_j \|k^{(0)}(\nu)\|_2^2 w_j^{(0)\top} v^{(0)}(\nu) \right| p_{q\leftarrow\mu,k\leftarrow\nu}^{(t,i)}$$

$$+ \left| \sum_{j=1}^{m_1} a_j \|k^{(0)}(\nu)\|_2^2 w_j^{(0)\top} v^{(0)}(\nu) \right| \cdot \left| p_{q\leftarrow\mu,k\leftarrow\nu}^{(t,i)} - p_{q\leftarrow\mu,k\leftarrow\nu}^{(0,i)} \right|.$$

Further, by Lemma D.1, Lemma D.3, Definition E.1 and Definition E.2, we have

$$\left| \sum_{j=1}^{m_1} a_j \|k^{(t)}(\nu)\|_2^2 w_j^{(t)\top} v^{(t)}(\nu) - \sum_{j=1}^{m_1} a_j \|k^{(0)}(\nu)\|_2^2 w_j^{(0)\top} v^{(0)}(\nu) \right|$$

$$\leq \max_{\nu,j} \left| \|k^{(t)}(\nu)\|_2^2 w_j^{(t)\top} v^{(t)}(\nu) - \|k^{(0)}(\nu)\|_2^2 w_j^{(0)\top} v^{(0)}(\nu) \right|$$

$$\leq m_1 \left( R_K \widetilde{O}(\sigma_0 \sigma_1 \sqrt{m}) + R\widetilde{O}(\sigma_0^2 m) + R_K R \right). \tag{9}$$

Combining with Proposition E.22, this implies

$$\left| \sum_{j=1}^{m_1} a_j \|k^{(t)}(\nu)\|_2^2 w_j^{(t)\top} v^{(t)}(\nu) p_{q\leftarrow\mu,k\leftarrow\nu}^{(t,i)} - \sum_{j=1}^{m_1} a_j \|k^{(0)}(\nu)\|_2^2 w_j^{(0)\top} v^{(0)}(\nu) p_{q\leftarrow\mu,k\leftarrow\nu}^{(0,i)} \right|$$

$$\leq m_1 \cdot \max_{\nu,j} \left| \|k^{(t)}(\nu)\|_2^2 w_j^{(t)\top} v^{(t)}(\nu) - \|k^{(0)}(\nu)\|_2^2 w_j^{(0)\top} v^{(0)}(\nu) \right| p_{q\leftarrow\mu,k\leftarrow\nu}^{(t,i)}$$

$$+ \left| \sum_{j=1}^{m_1} a_j \|k^{(0)}(\nu)\|_2^2 w_j^{(0)\top} v^{(0)}(\nu) \right| \cdot \left| p_{q\leftarrow\mu,k\leftarrow\nu}^{(t,i)} - p_{q\leftarrow\mu,k\leftarrow\nu}^{(0,i)} \right|$$

$$\leq m_1 \left( R_K \widetilde{O}(\sigma_0 \sigma_1 \sqrt{m}) + R\widetilde{O}(\sigma_0^2 m) + R_K R \right) \left( \frac{2}{L} + R_P \right) + \widetilde{O}\left( R_P \sigma_0^2 m \sigma_1 \sigma_0 \sqrt{mm_1} \right).$$

Thus,

$$|(a)| = \left| \sum_{j=1}^{m_1} a_j \|k^{(t)}(\nu)\|_2^2 w_j^{(t)\top} v^{(t)}(\nu) p_{q\leftarrow\mu,k\leftarrow\nu}^{(t,i)} \right|$$

$$\leq \left| \sum_{j=1}^{m_1} a_j \|k^{(t)}(\nu)\|_2^2 w_j^{(t)\top} v^{(t)}(\nu) p_{q\leftarrow\mu,k\leftarrow\nu}^{(t,i)} - \sum_{j=1}^{m_1} a_j \|k^{(0)}(\nu)\|_2^2 w_j^{(0)\top} v^{(0)}(\nu) p_{q\leftarrow\mu,k\leftarrow\nu}^{(0,i)} \right|$$

$$+ \left| \sum_{j=1}^{m_1} a_j \|k^{(0)}(\nu)\|_2^2 w_j^{(0)\top} v^{(0)}(\nu) p_{q\leftarrow\mu,k\leftarrow\nu}^{(0,i)} \right|$$

$$\leq m_1 \left( R_K \widetilde{O}(\sigma_0 \sigma_1 \sqrt{m}) + R\widetilde{O}(\sigma_0^2 m) + R_K R \right) \left( \frac{2}{L} + R_P \right)$$

$$+ \widetilde{O}\left( R_P \sigma_0^2 m \sigma_1 \sigma_0 \sqrt{mm_1} \right) + \widetilde{O}\left( \sigma_0^2 m \sigma_1 \sigma_0 \sqrt{mm_1} \frac{1}{L} \right).$$

On the other hand, to analyze $(b)$, we have

$$\left| \sum_{j=1}^{m_1} a_j \|k^{(t)}(\nu)\|_2^2 w_j^{(t)\top} V^{(t,i)} p_{l(i,\mu)}^{(t,i)} p_{q\leftarrow\mu,k\leftarrow\nu}^{(t,i)} - \sum_{j=1}^{m_1} a_j \|k^{(0)}(\nu)\|_2^2 w_j^{(0)\top} V^{(0,i)} p_{l(i,\mu)}^{(0,i)} p_{q\leftarrow\mu,k\leftarrow\nu}^{(0,i)} \right|$$

$$\leq \left| \sum_{j=1}^{m_1} a_j \|k^{(t)}(\nu)\|_2^2 w_j^{(t)\top} V^{(t,i)} p_{l(i,\mu)}^{(t,i)} - \sum_{j=1}^{m_1} a_j \|k^{(0)}(\nu)\|_2^2 w_j^{(0)\top} V^{(0,i)} p_{l(i,\mu)}^{(0,i)} \right| \cdot p_{q\leftarrow\mu,k\leftarrow\nu}^{(t,i)}$$

$$+ \left| \sum_{j=1}^{m_1} a_j \|k^{(0)}(\nu)\|_2^2 w_j^{(0)\top} V^{(0,i)} p_{l(i,\mu)}^{(0,i)} \right| \cdot \left| p_{q\leftarrow\mu,k\leftarrow\nu}^{(t,i)} - p_{q\leftarrow\mu,k\leftarrow\nu}^{(0,i)} \right|. \tag{10}$$

Note that

$$
\left| \sum_{j=1}^{m_1} a_j \|k^{(t)}(\nu)\|_2^2 w_j^{(t)\top} V^{(t,i)} p_{l(i,\mu)}^{(t,i)} - \sum_{j=1}^{m_1} a_j \|k^{(0)}(\nu)\|_2^2 w_j^{(0)\top} V^{(0,i)} p_{l(i,\mu)}^{(0,i)} \right|
$$

$$
= \left| \sum_{l=1}^{L} \sum_{j=1}^{m_1} a_j \|k^{(t)}(\nu)\|_2^2 w_j^{(t)\top} V_l^{(t,i)} p_{l(i,\mu),l}^{(t,i)} - \sum_{l=1}^{L} \sum_{j=1}^{m_1} a_j \|k^{(0)}(\nu)\|_2^2 w_j^{(0)\top} V_l^{(0,i)} p_{l(i,\mu),l}^{(0,i)} \right|
$$

$$
\leq \sum_{l=1}^{L} \left| \sum_{j=1}^{m_1} a_j \|k^{(t)}(\nu)\|_2^2 w_j^{(t)\top} V_l^{(t,i)} p_{l(i,\mu),l}^{(t,i)} - \sum_{j=1}^{m_1} a_j \|k^{(0)}(\nu)\|_2^2 w_j^{(0)\top} V_l^{(0,i)} p_{l(i,\mu),l}^{(0,i)} \right|
$$

$$
\leq \sum_{l=1}^{L} \left| \sum_{j=1}^{m_1} a_j \|k^{(t)}(\nu)\|_2^2 w_j^{(t)\top} V_l^{(t,i)} - \sum_{j=1}^{m_1} a_j \|k^{(0)}(\nu)\|_2^2 w_j^{(0)\top} V_l^{(0,i)} \right| p_{l(i,\mu),l}^{(t,i)}
$$

$$
+ \sum_{l=1}^{L} \left| \sum_{j=1}^{m_1} a_j \|k^{(0)}(\nu)\|_2^2 w_j^{(0)\top} V_l^{(0,i)} \right| \cdot \left| p_{l(i,\mu),l}^{(t,i)} - p_{l(i,\mu),l}^{(0,i)} \right|
$$

$$
\leq \max_{l \in [L]} \left| \sum_{j=1}^{m_1} a_j \|k^{(t)}(\nu)\|_2^2 w_j^{(t)\top} V_l^{(t,i)} - \sum_{j=1}^{m_1} a_j \|k^{(0)}(\nu)\|_2^2 w_j^{(0)\top} V_l^{(0,i)} \right|
$$

$$
+ L R_P \max_{l \in [L]} \left| \sum_{j=1}^{m_1} a_j \|k^{(0)}(\nu)\|_2^2 w_j^{(0)\top} V_l^{(0,i)} \right|, \tag{11}
$$

which implies

$$
|(b)|
$$

$$
= \left| \sum_{j=1}^{m_1} a_j \|k^{(t)}(\nu)\|_2^2 w_j^{(t)\top} V^{(t,i)} p_{l(i,\mu)}^{(t,i)} p_{q \leftarrow \mu, k \leftarrow \nu}^{(t,i)} \right|
$$

$$
\leq \left| \sum_{j=1}^{m_1} a_j \|k^{(t)}(\nu)\|_2^2 w_j^{(t)\top} V^{(t,i)} p_{l(i,\mu)}^{(t,i)} p_{q \leftarrow \mu, k \leftarrow \nu}^{(t,i)} - \sum_{j=1}^{m_1} a_j \|k^{(0)}(\nu)\|_2^2 w_j^{(0)\top} V^{(0,i)} p_{l(i,\mu)}^{(0,i)} p_{q \leftarrow \mu, k \leftarrow \nu}^{(0,i)} \right|
$$

$$
+ \left| \sum_{j=1}^{m_1} a_j \|k^{(0)}(\nu)\|_2^2 w_j^{(0)\top} V^{(0,i)} p_{l(i,\mu)}^{(0,i)} p_{q \leftarrow \mu, k \leftarrow \nu}^{(0,i)} \right|
$$

$$
\stackrel{(i)}{\leq} \left| \sum_{j=1}^{m_1} a_j \|k^{(t)}(\nu)\|_2^2 w_j^{(t)\top} V^{(t,i)} p_{l(i,\mu)}^{(t,i)} - \sum_{j=1}^{m_1} a_j \|k^{(0)}(\nu)\|_2^2 w_j^{(0)\top} V^{(0,i)} p_{l(i,\mu)}^{(0,i)} \right| \cdot p_{q \leftarrow \mu, k \leftarrow \nu}^{(t,i)}
$$

$$
+ \left| \sum_{j=1}^{m_1} a_j \|k^{(0)}(\nu)\|_2^2 w_j^{(0)\top} V^{(0,i)} p_{l(i,\mu)}^{(0,i)} \right| \cdot \left| p_{q \leftarrow \mu, k \leftarrow \nu}^{(t,i)} - p_{q \leftarrow \mu, k \leftarrow \nu}^{(0,i)} \right|
$$

$$
+ \left| \sum_{j=1}^{m_1} a_j \|k^{(0)}(\nu)\|_2^2 w_j^{(0)\top} V^{(0,i)} p_{l(i,\mu)}^{(0,i)} p_{q \leftarrow \mu, k \leftarrow \nu}^{(0,i)} \right|
$$

$$
\stackrel{(ii)}{\leq} \left( \max_{l \in [L]} \left| \sum_{j=1}^{m_1} a_j \|k^{(t)}(\nu)\|_2^2 w_j^{(t)\top} V_l^{(t,i)} - \sum_{j=1}^{m_1} a_j \|k^{(0)}(\nu)\|_2^2 w_j^{(0)\top} V_l^{(0,i)} \right| \right.
$$

$$
\left. + L R_P \max_{l \in [L]} \left| \sum_{j=1}^{m_1} a_j \|k^{(0)}(\nu)\|_2^2 w_j^{(0)\top} V_l^{(0,i)} \right| \right) \cdot p_{q \leftarrow \mu, k \leftarrow \nu}^{(t,i)}
$$

$$+ \left| \sum_{j=1}^{m_1} a_j \|k^{(0)}(\nu)\|_2^2 w_j^{(0)\top} V^{(0,i)} p_{l(i,\mu)}^{(0,i)} \right| \cdot \left| p_{q\leftarrow\mu,k\leftarrow\nu}^{(t,i)} - p_{q\leftarrow\mu,k\leftarrow\nu}^{(0,i)} \right|$$

$$+ \left| \sum_{j=1}^{m_1} a_j \|k^{(0)}(\nu)\|_2^2 w_j^{(0)\top} V^{(0,i)} p_{l(i,\mu)}^{(0,i)} p_{q\leftarrow\mu,k\leftarrow\nu}^{(0,i)} \right|$$

$$\overset{(iii)}{\leq} \left( m_1 \left( R_K \widetilde{O}(\sigma_0\sigma_1\sqrt{m}) + R\widetilde{O}(\sigma_0^2 m) + R_K R \right) \right.$$

$$\left. + LR_P \widetilde{O}(\sigma_0^2 m\sigma_0\sigma_1\sqrt{mm_1}) \right) \cdot \left( \frac{1}{L} + R_P \right) + \widetilde{O}\left( \sigma_0^2 m\sigma_0\sigma_1\sqrt{mm_1} \left( \frac{1}{L} + R_P \right) \right)$$

where $(i)$ follows from Equation (10), $(ii)$ follows from Equation (11) and $(iii)$ follows from Equation (9) and Proposition E.22. Combining the upper bound for both $(a)$ and $(b)$, we obtain

$$|(1)| \leq \frac{1}{\sqrt{m}} \frac{\left|\{i: \mu,\nu \in X^{(i)}\}\right|}{n} \widetilde{O}\left( \left( m_1 \left( R_K \widetilde{O}(\sigma_0\sigma_1\sqrt{m}) + R\widetilde{O}(\sigma_0^2 m) + R_K R \right) \right. \right.$$

$$\left. \left. + LR_P \widetilde{O}(\sigma_0^2 m\sigma_0\sigma_1\sqrt{mm_1}) \right) \cdot \left( \frac{1}{L} + R_P \right) + \sigma_0^2 m\sigma_0\sigma_1\sqrt{mm_1} \left( \frac{1}{L} + R_P \right) \right)$$

$$= \frac{1}{\sqrt{m}} \frac{\left|\{i: \mu,\nu \in X^{(i)}\}\right|}{n} \widetilde{O}(1/L).$$

**Analysis of** $(2)$: $(2)$ can be analyzed similarly as $(1)$ with $\|k^{(0)}(\nu)\|_2^2$ replaced by $\nu^\top W_K^{(t)} K_l^{(t,i)}$. Thus, we only need to replace $\sigma_0^2 m$ with $\sigma_0^2\sqrt{m}$ and then take a sum over $l$. We have

$$|(2)| \leq \frac{1}{\sqrt{m}} \frac{\left|\{i: \mu \in X^{(i)}\}\right|}{n} \widetilde{O}\left( \left( m_1 \left( R_K \widetilde{O}(\sigma_0\sigma_1\sqrt{m}) + R\widetilde{O}(\sigma_0^2\sqrt{m}) + R_K R \right) \right. \right.$$

$$\left. \left. + LR_P \widetilde{O}(\sigma_0^2\sqrt{m}\sigma_0\sigma_1\sqrt{mm_1}) \right) \cdot (1 + LR_P) + \sigma_0^2\sqrt{m}\sigma_0\sigma_1\sqrt{mm_1}(1 + LR_P) \right)$$

$$= \frac{1}{\sqrt{m}} \frac{\left|\{i: \mu \in X^{(i)}\}\right|}{n} \widetilde{O}(1/\sqrt{m}).$$

**Analysis of** $(3)$: $(3)$ can be analyzed similarly to $(1)$, and we obtain

$$|(3)| \leq \frac{1}{\sqrt{m}} \frac{\left|\{i: \mu,\nu \in X^{(i)}\}\right|}{n} \widetilde{O}(1/L).$$

**Analysis of** $(4)$: $(4)$ can be analyzed similarly to $(2)$, and we obtain

$$|(4)| \leq \frac{1}{\sqrt{m}} \frac{\left|\{i: \nu \in X^{(i)}\}\right|}{n} \widetilde{O}(1/\sqrt{m}).$$

Finally, combining the bounds on $(1) - (4)$, we have

$$\left| \frac{\partial \nu^\top W_K^{(t)\top} W_Q^{(t)} \mu}{\partial t} \right| \leq \frac{1}{\sqrt{m}} \frac{\left|\{i: \mu,\nu \in X^{(i)}\}\right|}{n} \widetilde{O}(1/L) + \frac{1}{\sqrt{m}} \frac{\left|\{i: \mu \in X^{(i)}\}\right| + \left|\{i: \nu \in X^{(i)}\}\right|}{n} \widetilde{O}(1/\sqrt{m})$$

$$\leq \frac{1}{\sqrt{m}} \frac{\left|\{i: \mu \in X^{(i)}\}\right| + \left|\{i: \nu \in X^{(i)}\}\right|}{n} \widetilde{O}(1/L + 1/\sqrt{m}).$$

$\square$

**Corollary E.24** (Softmax change). *For $t \leq T_1$, we have the following:*

- *if both $X_{l_1}^{(i)}, X_{l_2}^{(i)} \in \mathcal{R}$, then*

$$\left| \frac{\partial p_{l_1,l_2}^{(t,i)}}{\partial t} \right| \leq \widetilde{O}\left( \frac{1}{L(L^2+n)\sqrt{m}} \right),$$

- *otherwise,*

$$\left| \frac{\partial p_{l_1,l_2}^{(t,i)}}{\partial t} \right| \leq \widetilde{O}\left( \frac{1}{L^2\sqrt{m}} \right).$$

*Thus,*

- *if both $X_{l_1}^{(i)}, X_{l_2}^{(i)} \in \mathcal{R}$, then*

$$\left| p_{l_1,l_2}^{(t,i)} - p_{l_1,l_2}^{(0,i)} \right| \leq \widetilde{O}\left( \left( \frac{1}{L} + \frac{1}{n} \right) \frac{1}{Lm^2} \left( \frac{1}{L} + \frac{1}{\sqrt{m}} \right) \right),$$

- *otherwise,*

$$\left| p_{l_1,l_2}^{(t,i)} - p_{l_1,l_2}^{(0,i)} \right| \leq \widetilde{O}\left( \frac{1}{Lm^2} \left( \frac{1}{L} + \frac{1}{\sqrt{m}} \right) \right).$$

*Proof.* Consider fixed $i, l_1, l_2$. By Lemma G.2, we have

$$\left| \frac{\partial p_{l_1,l_2}^{(t,i)}}{\partial t} \right| \leq p_{l_1,l_2}^{(t,i)} \left| \frac{\partial s_{l_1,l_2}^{(t,i)}}{\partial t} \right| + p_{l_1,l_2}^{(t,i)} \left| p_{l_1}^{(t,i)\top} \frac{\partial s_{l_1}^{(t,i)}}{\partial t} \right|.$$

By Lemma E.23, we have the following cases:

- if both $X_{l_1}^{(i)}, X_{l_2}^{(i)} \in \mathcal{R}$, then

$$\left| \frac{\partial s_{l_1,l_2}^{(t,i)}}{\partial t} \right| \leq \widetilde{O}\left( \frac{1}{nm} \left( \frac{1}{L} + \frac{1}{\sqrt{m}} \right) \right);$$

- otherwise,

$$\left| \frac{\partial s_{l_1,l_2}^{(t,i)}}{\partial t} \right| \leq \widetilde{O}\left( \frac{1}{m} \left( \frac{1}{L} + \frac{1}{\sqrt{m}} \right) \right).$$

Next, during Phase 1, we have $p_{l_1,l_2}^{(t,i)} \leq \frac{1}{L} + \frac{1}{Lm} + R_P$. Therefore,

- if $X_{l_1}^{(i)} \in \mathcal{R}$, then

$$\left| p_{l_1}^{(t,i)\top} \frac{\partial s_{l_1}^{(t,i)}}{\partial t} \right| \leq \widetilde{O}\left( \left( \frac{1}{L} + \frac{1}{n} \right) \frac{1}{m} \left( \frac{1}{L} + \frac{1}{\sqrt{m}} \right) \right);$$

- otherwise,

$$\left| p_{l_1}^{(t,i)\top} \frac{\partial s_{l_1}^{(t,i)}}{\partial t} \right| \leq \widetilde{O}\left( \frac{1}{m} \left( \frac{1}{L} + \frac{1}{\sqrt{m}} \right) \right).$$

Thus,

- if both $X_{l_1}^{(i)}, X_{l_2}^{(i)} \in \mathcal{R}$, then

$$\left| \frac{\partial p_{l_1,l_2}^{(t,i)}}{\partial t} \right| \leq \widetilde{O}\left( \left( \frac{1}{L} + \frac{1}{n} \right) \frac{1}{Lm} \left( \frac{1}{L} + \frac{1}{\sqrt{m}} \right) \right);$$

- otherwise,
$$\left|\frac{\partial p_{l_1,l_2}^{(t,i)}}{\partial t}\right| \leq \widetilde{O}\left(\frac{1}{Lm}\left(\frac{1}{L}+\frac{1}{\sqrt{m}}\right)\right).$$

This implies that

- if both $X_{l_1}^{(i)}, X_{l_2}^{(i)} \in \mathcal{R}$, then
$$\left|p_{l_1,l_2}^{(t,i)} - p_{l_1,l_2}^{(0,i)}\right| \leq \widetilde{O}\left(\left(\frac{1}{L}+\frac{1}{n}\right)\frac{1}{Lm^2}\left(\frac{1}{L}+\frac{1}{\sqrt{m}}\right)\right);$$

- otherwise,
$$\left|p_{l_1,l_2}^{(t,i)} - p_{l_1,l_2}^{(0,i)}\right| \leq \widetilde{O}\left(\frac{1}{Lm^2}\left(\frac{1}{L}+\frac{1}{\sqrt{m}}\right)\right).$$

$\square$

**Lemma E.25** ($K, Q$ self-correlation change). *For $t \leq T_1$, for $\mu, \nu \in \{\mu_i\}_{i=1}^d$, we have*
$$\left|\mu^\top W_K^{(t)\top} W_K^{(t)} \nu - \mu^\top W_K^{(0)\top} W_K^{(0)} \nu\right| \leq t\frac{1}{\sqrt{m}}\frac{\left|\{i:\ \mu \in X^{(i)}\}\right| + \left|\{i:\ \nu \in X^{(i)}\}\right|}{n}\widetilde{O}\left(\frac{1}{\sqrt{m}}\right),$$
$$\left|\mu^\top W_Q^{(t)\top} W_Q^{(t)} \nu - \mu^\top W_Q^{(0)\top} W_Q^{(0)} \nu\right| \leq t\frac{1}{\sqrt{m}}\frac{\left|\{i:\ \mu \in X^{(i)}\}\right| + \left|\{i:\ \nu \in X^{(i)}\}\right|}{n}\widetilde{O}\left(\frac{1}{\sqrt{m}}\right).$$

*Proof.* By Lemma C.4, we only need to replace $\widetilde{O}(\sigma_0^2 m)$ by $\widetilde{O}(\sigma_0^2\sqrt{m})$ in the proof of Lemma E.23. Thus, we omit the proof here. $\square$

### E.8 All Variables are within Range in Definition of Phase 1

Finally, we prove that at the end of Phase 1, all the variables in Definition E.2 stay in the range.

**Theorem E.26.** *For $t \leq C_{T_1}/(\sigma_1^2 m m_1)$ (where the constant $C_{T_1}$ is from the definition of $T_1$ in Definition E.2), all of the following hold:*

1. $\max_{j\in[m],\mu\in\{\mu_i\}_{i=1}^3}\left|w_j^{(t)} W_V^{(t)} \mu - w_j^{(0)} W_V^{(0)} \mu\right| \leq R$, *where $R = O(1/m_1)$;*

2. $\max_{j\in[m],\mu\notin\{\mu_i\}_{i=1}^3}\left|w_j^{(t)} W_V^{(t)} \mu - w_j^{(0)} W_V^{(0)} \mu\right| \leq O(R/n + R/\sqrt{m})$;

3. $\max_{\mu,\nu\in\{\mu_i\}_{i=1}^d}\left|\mu^\top W_Q^{(t)\top} W_K^{(t)} \nu - \mu^\top W_Q^{(0)\top} W_K^{(0)} \nu\right| \leq R_S$, *where $R_S \leq O(1/m\sqrt{m})$;*

4. $\max_{\mu,\nu\in\{\mu_i\}_{i=1}^d}\left|\mu^\top W_Q^{(t)\top} W_Q^{(t)} \nu - \mu^\top W_Q^{(0)\top} W_Q^{(0)} \nu\right| \leq R_Q$;

5. $\max_{\mu,\nu\in\{\mu_i\}_{i=1}^d}\left|\mu^\top W_K^{(t)\top} W_K^{(t)} \nu - \mu^\top W_K^{(0)\top} W_K^{(0)} \nu\right| \leq R_K$.

*Thus, $T_1 = C_{T_1}/(\sigma_1^2 m m_1)$.*

*Proof.* The first two results are proved by Theorem E.14, Theorem E.15, Lemma E.17. The third result is proved by Lemma E.23, The fourth and fifth results are proved by Lemma E.25. $\square$

**Theorem E.27** (End of Phase 1). *At $t = T_1$, we have $y_i F_i^{(T_1)} = \Theta(1)$ for all $i \in [n]$, and*
$$G^{(T_1)}(\mu_1) = \Theta(1), \quad G^{(T_1)}(\mu_2) = \Theta(1), \quad -G^{(T_1)}(\mu_3) = \Theta(1),$$
$$\forall \mu \in \mathcal{R}:\ G^{(T_1)}(\mu) = \widetilde{O}(\sigma_0\sigma_1\sqrt{m m_1}).$$
*Further,*
$$\sum_{j_1=1}^{m_1}\sum_{j_2=1}^{m_1}\left\langle a_{j_1} w_{j_1}^{(T_1)}, a_{j_2} w_{j_2}^{(T_1)}\right\rangle = \Theta(\sigma_1^2 m m_1).$$

*Proof.* By Theorem E.26, we have $y_i F_i^{(T_1)} = O(1)$ and

$$G^{(T_1)}(\mu_1) = O(1), \quad G^{(T_1)}(\mu_2) = O(1), \quad -G^{(T_1)}(\mu_3) = O(1).$$

Further, by Theorem E.14 and Theorem E.19, we have $y_i F_i^{(T_1)} \geq \Omega(1)$ for $i \in I_1$ and

$$G^{(T_1)}(\mu_1) \geq \Omega(1), \quad G^{(T_1)}(\mu_2) \geq \Omega(1).$$

Finally, by Theorem E.26, we have that $T_1 = C_{T_1}/m$. And note that if the constant $C_{T_1}$ in Definition E.2 is sufficiently large, then by Corollary E.20, we have $T' \leq T_1$. Thus, we have $y_i F_i^{(T_1)} \geq \Omega(1)$ for $i \in I_2 \cup I_3 \cup I_4$ and $-G^{(T_1)}(\mu_3) \geq \Omega(1)$.

By Lemma D.7 and Theorem E.15, we have

$$\forall \mu \in \mathcal{R}: \ G^{(T_1)}(\mu) = \widetilde{O}(\sigma_0 \sigma_1 \sqrt{mm_1}).$$

Finally, by Lemma D.3, Proposition E.5 and Proposition E.12, we have

$$\sum_{j_1=1}^{m_1} \sum_{j_2=1}^{m_1} \left\langle a_{j_1} w_{j_1}^{(T_1)}, a_{j_2} w_{j_2}^{(T_1)} \right\rangle = \Theta(\sigma_1^2 mm_1).$$

$\square$

**Theorem E.28** (Phase 1, formal restatement of Theorem 3.1). *With probability at least $1 - \delta$ over the randomness of weight initialization, there exists a time $T_1 = \widetilde{O}(1/m)$ such that*

- $G^{(T_1)}(\mu_1) \geq \Omega(1)$, $G^{(T_1)}(\mu_2) \geq \Omega(1)$, $G^{(T_1)}(\mu_3) \leq -\Omega(1)$.

- *All the training samples are correctly classified:* $y_i F_i^{(T_1)} = \Omega(1)$ *for all* $i \in [n]$.

- *For* $t \in [0, T_1]$,

$$\left| \left\langle w_{j_1}^{(t)}, w_{j_2}^{(t)} \right\rangle - \left\langle w_{j_1}^{(0)}, w_{j_2}^{(0)} \right\rangle \right| \leq \widetilde{O}\left(\frac{1}{m}\right)$$

$$\left| \left\langle v^{(t)}(\mu), v^{(t)}(\nu) \right\rangle - \left\langle v^{(0)}(\mu), v^{(0)}(\nu) \right\rangle \right| \leq \widetilde{O}\left(\frac{1}{m}\right)$$

$$\left| \nu^\top W_K^{(t)\top} W_Q^{(t)} \mu - \nu^\top W_K^{(0)\top} W_Q^{(0)} \mu \right| \leq \widetilde{O}\left(\frac{1}{m^{3/2}}\left(\frac{1}{L} + \frac{1}{\sqrt{m}}\right)\right)$$

$$\left| \mu^\top W_K^{(t)\top} W_K^{(t)} \nu - \mu^\top W_K^{(0)\top} W_K^{(0)} \nu \right| \leq \widetilde{O}\left(\frac{1}{m^2}\right)$$

$$\left| \mu^\top W_Q^{(t)\top} W_Q^{(t)} \nu - \mu^\top W_Q^{(0)\top} W_Q^{(0)} \nu \right| \leq \widetilde{O}\left(\frac{1}{m^2}\right)$$

- *The training loss satisfies* $\widehat{L}^{(T_1)} = \Theta(1)$.

*Proof.* The first two results are proved in Theorem E.27. The third result is proved by Lemma E.9, Proposition E.12, Lemma E.23, and Lemma E.25. The result on the training loss is a direct consequence of Definition E.2. $\square$

## F  Training Dynamics: Phase 2

The idea of the proof is to first define conditions for Phase 2, which guarantees that the small training loss can be achieved. Then we will show that those conditions can be satisfied starting from the end of Phase 1 and up to at least $\Omega(\text{poly}(m))$ time, which will serve as the end of Phase 2.

**Definition F.1.** *We define Phase 2 of the training to be* $t \in [T_1, T_2]$ *such that*

- *The change of $K, Q$ self-correlation is small:*

$$\max_{\mu,\nu} \left| \mu^\top W_K^{(t)\top} W_K^{(t)} \nu - \mu^\top W_K^{(0)\top} W_K^{(0)} \nu \right| = \widetilde{O}(\sigma_0^2 \sqrt{m})$$

$$\max_{\mu,\nu} \left| \mu^\top W_Q^{(t)\top} W_Q^{(t)} \nu - \mu^\top W_Q^{(0)\top} W_Q^{(0)} \nu \right| = \widetilde{O}(\sigma_0^2 \sqrt{m})$$

- *The change of softmax probability satisfies:*

$$\max_{l_1,l_2 \in [L],\, i \in [n]} \left| p_{l_1,l_2}^{(t,i)} - p_{l_1,l_2}^{(0,i)} \right| < O(1/L^2)$$

- *The sum of neuron correlation satisfies*

$$\sum_{j_1=1}^{m_1} \sum_{j_2=1}^{m_1} \left\langle a_{j_1} w_{j_1}^{(t)}, a_{j_2} w_{j_2}^{(t)} \right\rangle \le O(\sigma_1^2 m m_1).$$

- *The gradient of $W, W_V$ satisfies*

$$\frac{\max_{\mu \in \{\mu_i\}_{i=1}^d} \left| \sum_{j=1}^{m_1} a_j \frac{\partial w_j^{(t)}}{\partial t} W_V^{(t)} \mu \right|}{\sum_{j_1=1}^{m_1} \sum_{j_2=1}^{m_1} \left\langle a_{j_1} w_{j_1}^{(t)}, a_{j_2} w_{j_2}^{(t)} \right\rangle} \le o(1/L) \frac{1}{n} \sum_{i \in [n]} g_i^{(t)}$$

$$\frac{\max_{i \in [n]} \left| \sum_{l_1=1}^{L} \sum_{l_2=1}^{L} G^{(t)}(X_{l_2}^{(i)}) \frac{\partial p_{l_1,l_2}^{(t,i)}}{\partial t} \right|}{\sum_{j_1=1}^{m_1} \sum_{j_2=1}^{m_1} \left\langle a_{j_1} w_{j_1}^{(t)}, a_{j_2} w_{j_2}^{(t)} \right\rangle} \le o(1) \frac{1}{n} \sum_{i \in [n]} g_i^{(t)}$$

- *The gradients $\sum_{i \in I} g_i^{(t)}$ for $I \in \{I_1, I_2, I_3, I_4\}$ satisfies*

$$\sum_{i \in I_4} g_i \le \min\left( \sum_{i \in I_2} g_i^{(t)}, \sum_{i \in I_3} g_i^{(t)} \right) \le \max\left( \sum_{i \in I_2} g_i^{(t)}, \sum_{i \in I_3} g_i^{(t)} \right) \le \sum_{i \in I_1} g_i^{(t)} \le \sum_{i \in I_2 \cup I_3 \cup I_4} g_i^{(t)}$$

  *and*

$$\frac{1}{2} \le \frac{\sum_{i \in I_2} g_i^{(t)}}{\sum_{i \in I_3} g_i^{(t)}} \le 2.$$

- $y_i F_i^{(t)} > C$ *for all $i \in [n]$ for some fixed constant $C$.*

- $|G^{(t)}(\mu)| \le O(\log m)$ *for $\mu \in \{\mu_i\}_{i=1}^3$.*

- $\sum_{l_1=1}^{L} \sum_{l_2 \colon X_{l_2}^{(i)} \in \{\mu_k\}_{k=4}^d} G^{(t)}(X_{l_2}^{(i)}) p_{q \leftarrow l_1, k \leftarrow l_2}^{(t,i)} \le O(1)$.

- $T_2 \le O(\text{poly}(m))$.

**Corollary F.2.** *For $t \in [T_1, T_2]$, if $i, j \in I$ where $I \in \{I_1, I_2, I_3, I_4\}$, then*

$$\max_{i,j \in I} \frac{g_i^{(t)}}{g_j^{(t)}} \le O(1).$$

*Proof.* Take $I = I_1$ and the proof is similar for the remaining cases. For fixed $i, j \in I_1$, we have

$$\frac{g_i^{(t)}}{g_j^{(t)}} = \frac{1 + \exp(y_j F_j^{(t)})}{1 + \exp(y_i F_i^{(t)})} \le 2 \frac{\exp(y_j F_j^{(t)})}{\exp(y_i F_i^{(t)})},$$

where the inequality is due to $y_i F_i^{(t)} \ge C$ in Definition F.1. Now we consider

$$\frac{\exp(y_j F_j^{(t)})}{\exp(y_i F_i^{(t)})}$$

$$= \frac{\exp(y_j \sum_{l_1=1}^{L} \sum_{l_2=1}^{L} G^{(t)}(X_{l_2}^{(j)}) p_{q \leftarrow l_1, k \leftarrow l_2}^{(t,j)})}{\exp(y_i \sum_{l_1=1}^{L} \sum_{l_2=1}^{L} G^{(t)}(X_{l_2}^{(i)}) p_{q \leftarrow l_1, k \leftarrow l_2}^{(t,i)})}$$

$$= \exp\left(\underbrace{\left(\sum_{l_1=1}^{L} \sum_{\mu \in \{\mu_k\}_{k=1}^{3}} G^{(t)}(\mu) p_{q \leftarrow l_1, k \leftarrow \mu}^{(t,j)}\right) - \left(\sum_{l_1=1}^{L} \sum_{\mu \in \{\mu_k\}_{k=1}^{3}} G^{(t)}(\mu) p_{q \leftarrow l_1, k \leftarrow \mu}^{(t,i)}\right)}_{(1)}\right)$$

$$\cdot \exp\left(\underbrace{\left(\sum_{l_1=1}^{L} \sum_{l_2 \colon X_{l_2}^{(j)} \in \{\mu_k\}_{k=4}^{d}} G^{(t)}(X_{l_2}^{(j)}) p_{q \leftarrow l_1, k \leftarrow l_2}^{(t,j)}\right) - \left(\sum_{l_1=1}^{L} \sum_{l_2 \colon X_{l_2}^{(i)} \in \{\mu_k\}_{k=4}^{d}} G^{(t)}(X_{l_2}^{(i)}) p_{q \leftarrow l_1, k \leftarrow l_2}^{(t,i)}\right)}_{(2)}\right).$$

By Definition F.1, it is easy to see that $|(2)| \leq O(1)$. For (1), we have

$$\left| \left(\sum_{l_1=1}^{L} \sum_{\mu \in \{\mu_k\}_{k=1}^{3}} G^{(t)}(\mu) p_{q \leftarrow l_1, k \leftarrow \mu}^{(t,j)}\right) - \left(\sum_{l_1=1}^{L} \sum_{\mu \in \{\mu_k\}_{k=1}^{3}} G^{(t)}(\mu) p_{q \leftarrow l_1, k \leftarrow \mu}^{(t,i)}\right) \right|$$

$$\leq \left| \sum_{\mu \in \{\mu_k\}_{k=1}^{3}} G^{(t)}(\mu)(1 \pm o(1)) - \sum_{\mu \in \{\mu_k\}_{k=1}^{3}} G^{(t)}(\mu)(1 \pm o(1)) \right| \leq O(1).$$

Thus, we have

$$\frac{\exp(y_j F_j^{(t)})}{\exp(y_i F_i^{(t)})} \leq O(1)$$

for $t \in [T_1, T_2]$. $\qquad\square$

**Lemma F.3.** *Phase 2 in Definition F.1 is well-defined.*

*Proof.* We need to show that the definition is valid at $t = T_1$ with conditions satisfied with strict inequality. Then since everything changes continuously, there naturally exists a $T_2 > T_1$ such that Definition F.1 is well-defined.

First of all, by Lemma E.25, for $\mu, \nu \in \{\mu_i\}_{i=1}^{d}$, we have

$$\max_{\mu, \nu} \left| \mu^\top W_K^{(T_1)\top} W_K^{(T_1)} \nu - \mu^\top W_K^{(0)\top} W_K^{(0)} \nu \right| = \widetilde{O}(\sigma_0^2 \sqrt{m}),$$

$$\max_{\mu, \nu} \left| \mu^\top W_Q^{(T_1)\top} W_Q^{(T_1)} \nu - \mu^\top W_Q^{(0)\top} W_Q^{(0)} \nu \right| = \widetilde{O}(\sigma_0^2 \sqrt{m}).$$

Next, by Proposition E.12, Proposition E.5 and Lemma D.3, we have

$$\sum_{j_1=1}^{m_1} \sum_{j_2=1}^{m_1} \left\langle a_{j_1} w_{j_1}^{(T_1)}, a_{j_2} w_{j_2}^{(T_1)} \right\rangle = \Theta(\sigma_1^2 m m_1).$$

Recall that $\frac{1}{n} \sum_{i=1}^{n} g_i^{(T_1)} = \Theta(1)$. On the other hand, by Corollary E.10, Lemma D.3, we have

$$\max_{\mu \in \{\mu_i\}_{i=1}^{d}} \left| \sum_{j=1}^{m_1} a_j \frac{\partial w_j^{(T_1)}}{\partial t} W_V^{(T_1)} \mu \right| = O(\sigma_0^2 m m_1)$$

$$\implies \frac{\max_{\mu \in \{\mu_i\}_{i=1}^{d}} \left| \sum_{j=1}^{m_1} a_j \frac{\partial w_j^{(T_1)}}{\partial t} W_V^{(T_1)} \mu \right|}{\sum_{j_1=1}^{m_1} \sum_{j_2=1}^{m_1} \left\langle a_{j_1} w_{j_1}^{(T_1)}, a_{j_2} w_{j_2}^{(T_1)} \right\rangle} \leq O\left(\frac{\sigma_0^2}{\sigma_1^2}\right) = \widetilde{O}\left(\frac{m_1}{Lm}\right).$$

Also, by Corollary E.24, we have

$$\max_{i\in[n]}\left|\sum_{l_1=1}^{L}\sum_{l_2=1}^{L}G^{(T_1)}(X_{l_2}^{(i)})\frac{\partial p_{l_1,l_2}^{(T_1,i)}}{\partial t}\right|=\widetilde{O}\left(\frac{1}{\sqrt{m}}\right)$$

$$\implies \frac{\max_{i\in[n]}\left|\sum_{l_1=1}^{L}\sum_{l_2=1}^{L}G^{(T_1)}(X_{l_2}^{(i)})\frac{\partial p_{l_1,l_2}^{(T_1,i)}}{\partial t}\right|}{\sum_{j_1=1}^{m_1}\sum_{j_2=1}^{m_1}\left\langle a_{j_1}w_{j_1}^{(T_1)},a_{j_2}w_{j_2}^{(T_1)}\right\rangle}\leq\widetilde{O}\left(\frac{1}{m^{3/2}}\right)\leq o(1)\sum_{i\in[n]}g_i^{(T_1)}.$$

Further, by Corollary E.24, we have

$$\max_{l_1,l_2\in[L],\ i\in[n]}\left|p_{l_1,l_2}^{(T_1,i)}-p_{l_1,l_2}^{(0,i)}\right|<O(1/L^2).$$

Now, by Proposition E.21, if $F$ is small enough (which can be achieved by making the initialization scale small enough), then

$$\frac{1}{2}<\frac{\sum_{i\in I_2}g_i^{(T_1)}}{\sum_{i\in I_3}g_i^{(T_1)}}<2.$$

Lastly, by Theorem E.27, we have $y_iF_i^{(T_1)}\geq\Omega(1)$. And it is straightforward to see that $|G^{(T_1)}(\mu)|\leq O(\log m)$ for $\mu\in\{\mu_i\}_{i=1}^{3}$.

Finally, we prove $\sum_{l_1=1}^{L}\sum_{l_2:\ X_{l_2}^{(i)}\in\{\mu_k\}_{k=4}^{d}}G^{(T_1)}(X_{l_2}^{(i)})p_{q\leftarrow l_1,k\leftarrow l_2}^{(T_1,i)}\leq O(1)$. A simple corollary from Lemma D.7 and Lemma D.8 is that

$$\left|\sum_{l_1=1}^{L}\sum_{l_2:\ X_{l_2}^{(i)}\in\{\mu_k\}_{k=4}^{d}}G^{(0)}(X_{l_2}^{(i)})p_{q\leftarrow l_1,k\leftarrow l_2}^{(0,i)}\right|\leq O(1).$$

Next, we have

$$\left|\sum_{l_1=1}^{L}\sum_{l_2:\ X_{l_2}^{(i)}\in\{\mu_k\}_{k=4}^{d}}G^{(T_1)}(X_{l_2}^{(i)})p_{q\leftarrow l_1,k\leftarrow l_2}^{(T_1,i)}-\sum_{l_1=1}^{L}\sum_{l_2:\ X_{l_2}^{(i)}\in\{\mu_k\}_{k=4}^{d}}G^{(0)}(X_{l_2}^{(i)})p_{q\leftarrow l_1,k\leftarrow l_2}^{(0,i)}\right|$$

$$\leq\sum_{l_1=1}^{L}\sum_{l_2:\ X_{l_2}^{(i)}\in\{\mu_k\}_{k=4}^{d}}\left|(G^{(T_1)}(X_{l_2}^{(i)})-G^{(0)}(X_{l_2}^{(i)}))\right|p_{q\leftarrow l_1,k\leftarrow l_2}^{(0,i)}$$

$$+\sum_{l_1=1}^{L}\sum_{l_2:\ X_{l_2}^{(i)}\in\{\mu_k\}_{k=4}^{d}}G^{(0)}(X_{l_2}^{(i)})\left|p_{q\leftarrow l_1,k\leftarrow l_2}^{(T_1,i)}-p_{q\leftarrow l_1,k\leftarrow l_2}^{(0,i)}\right|$$

$$+\sum_{l_1=1}^{L}\sum_{l_2:\ X_{l_2}^{(i)}\in\{\mu_k\}_{k=4}^{d}}\left|G^{(T_1)}(X_{l_2}^{(i)})-G^{(0)}(X_{l_2}^{(i)})\right|\left|p_{q\leftarrow l_1,k\leftarrow l_2}^{(T_1,i)}-p_{q\leftarrow l_1,k\leftarrow l_2}^{(0,i)}\right|$$

$$\leq O(1)$$

which proves that

$$\sum_{l_1=1}^{L}\sum_{l_2:\ X_{l_2}^{(i)}\in\{\mu_k\}_{k=4}^{d}}G^{(T_1)}(X_{l_2}^{(i)})p_{q\leftarrow l_1,k\leftarrow l_2}^{(T_1,i)}\leq O(1).$$

$\square$

In Phase 2, we will analyze the dynamical system in a different way since all the variables now might change dramatically from their values at initialization.

**Lemma F.4.** *For $t \in [T_1, T_2]$, we have*

$$\frac{\partial}{\partial t} \sum_{j_1=1}^{m_1} \sum_{j_2=1}^{m_1} \left\langle a_{j_1} w_{j_1}^{(t)}, a_{j_2} w_{j_2}^{(t)} \right\rangle = \frac{2m_1}{n} \sum_{i=1}^{n} g_i^{(t)} y_i F_i^{(t)} > 0$$

*and thus,*

$$\sum_{j_1=1}^{m_1} \sum_{j_2=1}^{m_1} \left\langle a_{j_1} w_{j_1}^{(t)}, a_{j_2} w_{j_2}^{(t)} \right\rangle = \Omega(\sigma_1^2 m m_1).$$

*Proof.* By Lemma C.2, we obtain

$$\frac{\partial}{\partial t} \sum_{j_1=1}^{m_1} \sum_{j_2=1}^{m_1} \left\langle a_{j_1} w_{j_1}^{(t)}, a_{j_2} w_{j_2}^{(t)} \right\rangle$$

$$= \sum_{j_1=1}^{m_1} \sum_{j_2=1}^{m_1} \left( \frac{1}{n} \sum_{i=1}^{n} g_i^{(t)} y_i \sum_{l_1=1}^{L} a_{j_1} w_{j_1}^{(t)\top} V^{(t,i)} p_{l_1}^{(t,i)} + \frac{1}{n} \sum_{i=1}^{n} g_i^{(t)} y_i \sum_{l_2=1}^{L} a_{j_2} w_{j_2}^{(t)\top} V^{(t,i)} p_{l_2}^{(t,i)} \right)$$

$$= \frac{2m_1}{n} \sum_{i=1}^{n} g_i^{(t)} y_i F_i^{(t)}.$$

Then, note that the final term is positive since by Definition F.1, we have $y_i F_i^{(t)} > 0$ for all $i \in [n]$. Finally, by Theorem E.27, we have for all $t \in [T_1, T_2]$,

$$\sum_{j_1=1}^{m_1} \sum_{j_2=1}^{m_1} \left\langle a_{j_1} w_{j_1}^{(t)}, a_{j_2} w_{j_2}^{(t)} \right\rangle \geq \Omega(\sigma_1^2 m m_1).$$

$\square$

## F.1 Automatic Balancing of Gradients

**Lemma F.5** (Same as Lemma 4.4). *For $t \in [T_1, T_2]$, there exists a small constant $C \ll 1$ such that*

$$\frac{\left| \sum_{i \in I_2} g_i^{(t)} - \sum_{i \in I_3} g_i^{(t)} \right|}{\min(\sum_{i \in I_2} g_i^{(t)}, \sum_{i \in I_3} g_i^{(t)})} \leq C.$$

*Proof.* Without loss of generality, assume $\sum_{i \in I_2} g_i^{(t)} > \sum_{i \in I_3} g_i^{(t)}$. Then, we have

$$\frac{\left| \sum_{i \in I_2} g_i^{(t)} - \sum_{i \in I_3} g_i^{(t)} \right|}{\min(\sum_{i \in I_2} g_i^{(t)}, \sum_{i \in I_3} g_i^{(t)})} = \frac{\sum_{i \in I_2} g_i^{(t)} - \sum_{i \in I_3} g_i^{(t)}}{\sum_{i \in I_3} g_i^{(t)}} = \frac{\sum_{i \in I_2} g_i^{(t)}}{\sum_{i \in I_3} g_i^{(t)}} - 1.$$

Now by the quotient rule, we have $\frac{\partial}{\partial t} \left( \frac{\sum_{i \in I_2} g_i^{(t)}}{\sum_{i \in I_3} g_i^{(t)}} \right) \leq 0$ if and only if

$$\frac{\partial}{\partial t} \left( \sum_{i \in I_2} g_i^{(t)} \right) \left( \sum_{i \in I_3} g_i^{(t)} \right) - \left( \sum_{i \in I_2} g_i^{(t)} \right) \frac{\partial}{\partial t} \left( \sum_{i \in I_3} g_i^{(t)} \right) \leq 0$$

$$\Leftrightarrow \left( \sum_{i \in I_2} g'(y_i F_i^{(t)}) \frac{\partial y_i F_i^{(t)}}{\partial t} \right) \left( \sum_{i \in I_3} g_i^{(t)} \right) - \left( \sum_{i \in I_2} g_i^{(t)} \right) \left( \sum_{i \in I_3} g'(y_i F_i^{(t)}) \frac{\partial y_i F_i^{(t)}}{\partial t} \right) \leq 0$$

$$\Leftrightarrow \left( \sum_{i \in I_2} g_i^{(t)} \right) \left( \sum_{i \in I_3} g_i^{(t)} \right) \left( \frac{1}{\sum_{i \in I_2} g_i^{(t)}} \sum_{i \in I_2} g'(y_i F_i^{(t)}) \frac{\partial y_i F_i^{(t)}}{\partial t} - \frac{1}{\sum_{i \in I_3} g_i^{(t)}} \sum_{i \in I_3} g'(y_i F_i^{(t)}) \frac{\partial y_i F_i^{(t)}}{\partial t} \right) \leq 0$$

$$\Leftrightarrow \left( \frac{1}{\sum_{i \in I_2} g_i^{(t)}} \sum_{i \in I_2} g'(y_i F_i^{(t)}) \frac{\partial y_i F_i^{(t)}}{\partial t} - \frac{1}{\sum_{i \in I_3} g_i^{(t)}} \sum_{i \in I_3} g'(y_i F_i^{(t)}) \frac{\partial y_i F_i^{(t)}}{\partial t} \right) \leq 0$$

$$\Leftrightarrow \left( \frac{1}{\sum_{i \in I_2} g_i^{(t)}} \sum_{i \in I_2} \frac{-1}{(1 + \exp(y_i F_i^{(t)}))(1 + \exp(-y_i F_i^{(t)}))} \frac{\partial y_i F_i^{(t)}}{\partial t} \right)$$

$$- \left( \frac{1}{\sum_{i \in I_3} g_i^{(t)}} \sum_{i \in I_3} \frac{-1}{(1 + \exp(y_i F_i^{(t)}))(1 + \exp(-y_i F_i^{(t)}))} \frac{\partial y_i F_i^{(t)}}{\partial t} \right) \leq 0$$

$$\Leftrightarrow \left( \frac{1}{\sum_{i \in I_3} g_i^{(t)}} \sum_{i \in I_3} \frac{1}{(1 + \exp(y_i F_i^{(t)}))(1 + \exp(-y_i F_i^{(t)}))} \frac{\partial y_i F_i^{(t)}}{\partial t} \right)$$

$$\leq \left( \frac{1}{\sum_{i \in I_2} g_i^{(t)}} \sum_{i \in I_2} \frac{1}{(1 + \exp(y_i F_i^{(t)}))(1 + \exp(-y_i F_i^{(t)}))} \frac{\partial y_i F_i^{(t)}}{\partial t} \right). \tag{12}$$

Note that

$$\frac{\sum_{i \in I} g_i^{(t)}}{1 + \min_{i \in I} \exp(-y_i F_i^{(t)})} \geq \sum_{i \in I} \frac{1}{(1 + \exp(y_i F_i^{(t)}))(1 + \exp(-y_i F_i^{(t)}))} \geq \frac{\sum_{i \in I} g_i^{(t)}}{1 + \max_{i \in I} \exp(-y_i F_i^{(t)})}.$$

By Definition F.1, we have that for $i' \in I_2$,

$$\frac{\partial y_{i'} F_{i'}^{(t)}}{\partial t}$$

$$= y_{i'} \sum_{l_1=1}^{L} \sum_{j=1}^{m_1} \sum_{l_2=1}^{L} \left( \frac{\partial a_j w_j^{(t)} W_V^{(t)} X_{l_2}^{(i')}}{\partial t} p_{l_1,l_2}^{(t,i')} + a_j w_j^{(t)} W_V^{(t)} X_{l_2}^{(i')} \frac{\partial p_{l_1,l_2}^{(t,i')}}{\partial t} \right)$$

$$= y_{i'} \sum_{l_2=1}^{L} \frac{\partial G^{(t)}(X_{l_2}^{(i')})}{\partial t}(1 + o(1)) + \sum_{l_1=1}^{L} \sum_{l_2=1}^{L} y_{i'} G^{(t)}(X_{l_2}^{(i')}) \frac{\partial p_{l_1,l_2}^{(t,i')}}{\partial t}$$

$$= - \sum_{j_1=1}^{m_1} \sum_{j_2=1}^{m_1} \left\langle a_{j_1} w_{j_1}^{(t)}, a_{j_2} w_{j_2}^{(t)} \right\rangle \frac{1}{n} \left( \sum_{i \in I_1} g_i^{(t)} - \sum_{i \in I_2 \cup I_3 \cup I_4} g_i^{(t)} + \sum_{i \in I_1} g_i^{(t)} - \sum_{i \in I_2} g_i^{(t)} + o(1) \sum_{i \in [n]} g_i^{(t)} \right).$$

Similarly, for $i' \in I_3$, we have

$$\frac{\partial y_{i'} F_{i'}^{(t)}}{\partial t}$$

$$= - \sum_{j_1=1}^{m_1} \sum_{j_2=1}^{m_1} \left\langle a_{j_1} w_{j_1}^{(t)}, a_{j_2} w_{j_2}^{(t)} \right\rangle \frac{1}{n} \left( \sum_{i \in I_1} g_i^{(t)} - \sum_{i \in I_2 \cup I_3 \cup I_4} g_i^{(t)} + \sum_{i \in I_1} g_i^{(t)} - \sum_{i \in I_3} g_i^{(t)} + o(1) \sum_{i \in [n]} g_i^{(t)} \right).$$

Now, we analyze Equation (12). Note that for $t \in [T_1, T_2]$, if $\sum_{i \in I_2} g_i^{(t)}$ is sufficiently larger than $\sum_{i \in I_3} g_i^{(t)}$ (i.e., $\sum_{i \in I_2} g_i^{(t)} / \sum_{i \in I_3} g_i^{(t)}$ is bigger than some threshold), then $\frac{\partial G^{(t)}(\mu_1)}{\partial t} < \frac{\partial G^{(t)}(\mu_2)}{\partial t}$ and $\min_{i \in I_2} \frac{\partial y_i F_i^{(t)}}{\partial t} > \max_{i \in I_3} \frac{\partial y_i F_i^{(t)}}{\partial t}$. Thus, the ratio will decrease. Similarly, if $\sum_{i \in I_2} g_i^{(t)} / \sum_{i \in I_3} g_i^{(t)}$ is too small, the ratio will increase. Next, we compute a bound on $\sum_{i \in I_2} g_i^{(t)} / \sum_{i \in I_3} g_i^{(t)}$ so that $\frac{\partial}{\partial t} \left( \frac{\sum_{i \in I_2} g_i^{(t)}}{\sum_{i \in I_3} g_i^{(t)}} \right) = 0$. Substituting $\frac{\partial y_i F_i^{(t)}}{\partial t}$ in Equation (12), we have

$$\left( \frac{1}{\sum_{i \in I_3} g_i^{(t)}} \sum_{i \in I_3} \frac{1}{(1 + \exp(y_i F_i^{(t)}))(1 + \exp(-y_i F_i^{(t)}))} \right)$$

$$\cdot \left( - \sum_{j_1=1}^{m_1} \sum_{j_2=1}^{m_1} \left\langle a_{j_1} w_{j_1}^{(t)}, a_{j_2} w_{j_2}^{(t)} \right\rangle \left( \sum_{i \in I_1} g_i^{(t)} - \sum_{i \in I_2 \cup I_3 \cup I_4} g_i^{(t)} + \sum_{i \in I_1} g_i^{(t)} - \sum_{i \in I_2} g_i^{(t)} \pm o(1) \sum_{i \in [n]} g_i^{(t)} \right) \right)$$

$$= \left( \frac{1}{\sum_{i \in I_2} g_i^{(t)}} \sum_{i \in I_2} \frac{1}{(1 + \exp(y_i F_i^{(t)}))(1 + \exp(-y_i F_i^{(t)}))} \right)$$

$$\cdot \left( -\sum_{j_1=1}^{m_1} \sum_{j_2=1}^{m_1} \left\langle a_{j_1} w_{j_1}^{(t)}, a_{j_2} w_{j_2}^{(t)} \right\rangle \left( \sum_{i \in I_1} g_i^{(t)} - \sum_{i \in I_2 \cup I_3 \cup I_4} g_i^{(t)} + \sum_{i \in I_1} g_i^{(t)} - \sum_{i \in I_3} g_i^{(t)} \pm o(1) \sum_{i \in [n]} g_i^{(t)} \right) \right) \right)$$

$$\Rightarrow \frac{1 + \min_{i \in I_2} \exp(-y_i F_i^{(t)})}{1 + \max_{i \in I_3} \exp(-y_i F_i^{(t)})} \left( -\sum_{i \in I_1} g_i^{(t)} + \sum_{i \in I_2 \cup I_3 \cup I_4} g_i^{(t)} - \sum_{i \in I_1} g_i^{(t)} + \sum_{i \in I_2} g_i^{(t)} \pm o(1) \sum_{i \in [n]} g_i^{(t)} \right)$$

$$= \left( -\sum_{i \in I_1} g_i^{(t)} + \sum_{i \in I_2 \cup I_3 \cup I_4} g_i^{(t)} - \sum_{i \in I_1} g_i^{(t)} + \sum_{i \in I_3} g_i^{(t)} \pm o(1) \sum_{i \in [n]} g_i^{(t)} \right)$$

$$\Rightarrow \frac{\sum_{i \in I_2} g_i^{(t)}}{\sum_{i \in I_3} g_i^{(t)}} = 1 + \left( \frac{\frac{1}{\sum_{i \in I_3} g_i^{(t)}} \sum_{i \in I_3} \frac{1}{(1+\exp(y_i F_i^{(t)}))(1+\exp(-y_i F_i^{(t)}))}}{\frac{1}{\sum_{i \in I_2} g_i^{(t)}} \sum_{i \in I_2} \frac{1}{(1+\exp(y_i F_i^{(t)}))(1+\exp(-y_i F_i^{(t)}))}} \right.$$

$$\left. \cdot \frac{-\sum_{i \in I_1} g_i^{(t)} + \sum_{i \in I_2 \cup I_3 \cup I_4} g_i^{(t)} - \sum_{i \in I_1} g_i^{(t)} + \sum_{i \in I_2} g_i^{(t)} \pm o(1) \sum_{i \in [n]} g_i^{(t)}}{\sum_{i \in I_3} g_i^{(t)}} \right).$$

Since

$$\frac{1}{\sum_{i \in I} g_i^{(t)}} \sum_{i \in I} \frac{1}{(1 + \exp(y_i F_i^{(t)}))(1 + \exp(-y_i F_i^{(t)}))}$$

$$\in \left[ \frac{1}{1 + \max_{i \in I} \exp(-y_i F_i^{(t)})}, \frac{1}{1 + \min_{i \in I} \exp(-y_i F_i^{(t)})} \right],$$

we have

$$\frac{\frac{1}{\sum_{i \in I_3} g_i^{(t)}} \sum_{i \in I_3} \frac{1}{(1+\exp(y_i F_i^{(t)}))(1+\exp(-y_i F_i^{(t)}))}}{\frac{1}{\sum_{i \in I_2} g_i^{(t)}} \sum_{i \in I_2} \frac{1}{(1+\exp(y_i F_i^{(t)}))(1+\exp(-y_i F_i^{(t)}))}}$$

$$\in \left[ \frac{1 + \min_{i \in I_2} \exp(-y_i F_i^{(t)})}{1 + \max_{i \in I_3} \exp(-y_i F_i^{(t)})}, \frac{1 + \max_{i \in I_2} \exp(-y_i F_i^{(t)})}{1 + \min_{i \in I_3} \exp(-y_i F_i^{(t)})} \right].$$

By Definition F.1, we have

$$\left| \frac{-\sum_{i \in I_1} g_i^{(t)} + \sum_{i \in I_2 \cup I_3 \cup I_4} g_i^{(t)} - \sum_{i \in I_1} g_i^{(t)} + \sum_{i \in I_2} g_i^{(t)} \pm o(1) \sum_{i \in [n]} g_i^{(t)}}{\sum_{i \in I_3} g_i^{(t)}} \right| \le 6.$$

Thus,

$$\frac{\sum_{i \in I_2} g_i^{(t)}}{\sum_{i \in I_3} g_i^{(t)}} \in 1 \pm 6 \cdot \left( \left[ \frac{1 + \min_{i \in I_2} \exp(-y_i F_i^{(t)})}{1 + \max_{i \in I_3} \exp(-y_i F_i^{(t)})}, \frac{1 + \max_{i \in I_2} \exp(-y_i F_i^{(t)})}{1 + \min_{i \in I_3} \exp(-y_i F_i^{(t)})} \right] - 1 \right).$$

$\square$

**Lemma F.6** (Complete version of Lemma 4.5). *For $t \in [T_1, T_2]$, we have*

$$\frac{\sum_{i \in [n]} g_i^{(t)}}{\sum_{i \in I_2 \cup I_3 \cup I_4} g_i^{(t)} - \sum_{i \in I_1} g_i^{(t)}} = O(1), \qquad \frac{\sum_{i \in [n]} g_i^{(t)}}{\sum_{i \in I_1} g_i^{(t)} - \sum_{i \in I_2} g_i^{(t)}} = O(1)$$

$$\frac{\sum_{i \in [n]} g_i^{(t)}}{\sum_{i \in I_1} g_i^{(t)} - \sum_{i \in I_3} g_i^{(t)}} = O(1), \qquad \frac{\frac{\partial G^{(t)}(\mu_1)}{\partial t}}{\frac{\partial G^{(t)}(\mu_2)}{\partial t}} = \Theta(1).$$

*Further, there exists a constant $C$ such that*

$$(1 + C) \max\left( \frac{\partial G^{(t)}(\mu_1)}{\partial t}, \frac{\partial G^{(t)}(\mu_2)}{\partial t} \right) \le -\frac{\partial G^{(t)}(\mu_3)}{\partial t} \le (1 - C) \left( \frac{\partial G^{(t)}(\mu_1)}{\partial t} + \frac{\partial G^{(t)}(\mu_2)}{\partial t} \right).$$

*Proof.* Without loss of generality, assume $\frac{\partial G^{(t)}(\mu_1)}{\partial t} > \frac{\partial G^{(t)}(\mu_2)}{\partial t}$. First of all, by Definition F.1, we have

$$\frac{\partial G^{(t)}(\mu_1)}{\partial t} = \sum_{j_1=1}^{m_1} \sum_{j_2=1}^{m_1} \left\langle a_{j_1} w_{j_1}^{(t)}, a_{j_2} w_{j_2}^{(t)} \right\rangle \frac{1}{n} \left( \sum_{i \in I_1} g_i^{(t)} - \sum_{i \in I_2} g_i^{(t)} \pm o(1) \sum_{i \in [n]} g_i^{(t)} \right),$$

$$\frac{\partial G^{(t)}(\mu_2)}{\partial t} = \sum_{j_1=1}^{m_1} \sum_{j_2=1}^{m_1} \left\langle a_{j_1} w_{j_1}^{(t)}, a_{j_2} w_{j_2}^{(t)} \right\rangle \frac{1}{n} \left( \sum_{i \in I_1} g_i^{(t)} - \sum_{i \in I_3} g_i^{(t)} \pm o(1) \sum_{i \in [n]} g_i^{(t)} \right),$$

$$\frac{\partial G^{(t)}(\mu_3)}{\partial t} = \sum_{j_1=1}^{m_1} \sum_{j_2=1}^{m_1} \left\langle a_{j_1} w_{j_1}^{(t)}, a_{j_2} w_{j_2}^{(t)} \right\rangle \frac{1}{n} \left( \sum_{i \in I_1} g_i^{(t)} - \sum_{i \in I_2 \cup I_3 \cup I_4} g_i^{(t)} \pm o(1) \sum_{i \in [n]} g_i^{(t)} \right).$$

This implies that

$$\frac{\frac{\partial G^{(t)}(\mu_1)}{\partial t}}{\frac{\partial G^{(t)}(\mu_2)}{\partial t}} = \frac{\sum_{i \in I_1} g_i^{(t)} - \sum_{i \in I_2} g_i^{(t)} \pm o(1) \sum_{i \in [n]} g_i^{(t)}}{\sum_{i \in I_1} g_i^{(t)} - \sum_{i \in I_3} g_i^{(t)} \pm o(1) \sum_{i \in [n]} g_i^{(t)}},$$

$$\frac{-\frac{\partial G^{(t)}(\mu_3)}{\partial t}}{\frac{\partial G^{(t)}(\mu_1)}{\partial t}} = \frac{-\sum_{i \in I_1} g_i^{(t)} + \sum_{i \in I_2 \cup I_3 \cup I_4} g_i^{(t)} \pm o(1) \sum_{i \in [n]} g_i^{(t)}}{\sum_{i \in I_1} g_i^{(t)} - \sum_{i \in I_2} g_i^{(t)} \pm o(1) \sum_{i \in [n]} g_i^{(t)}}.$$

We next analyze $\frac{\partial}{\partial t} \frac{-\frac{\partial G^{(t)}(\mu_3)}{\partial t}}{\frac{\partial G^{(t)}(\mu_1)}{\partial t}}$. We first define the ratio $R(t) = \frac{\sum_{i \in I_2 \cup I_3 \cup I_4} g_i^{(t)} - \sum_{i \in I_1} g_i^{(t)}}{\sum_{i \in I_1} g_i^{(t)} - \sum_{i \in I_2} g_i^{(t)}}$. Since the dependence of $R$ on $t$ is clear and for the ease of notation, we omit this dependence below. Rearranging this definition, we obtain

$$(1 + R) \left( \sum_{i \in I_1} g_i^{(t)} - \sum_{i \in I_2} g_i^{(t)} \right) = \sum_{i \in I_3 \cup I_4} g_i^{(t)}. \tag{13}$$

We consider the range of $R \in [1, 3]$. By Definition F.1, we have

$$\frac{\sum_{i \in [n]} g_i^{(t)}}{\sum_{i \in I_2 \cup I_3 \cup I_4} g_i^{(t)} - \sum_{i \in I_1} g_i^{(t)}} = O(1), \qquad \frac{\sum_{i \in [n]} g_i^{(t)}}{\sum_{i \in I_1} g_i^{(t)} - \sum_{i \in I_2} g_i^{(t)}} = O(1), \tag{14}$$

where the second equation follows from Lemma F.5. This implies that

$$\frac{\frac{\partial G^{(t)}(\mu_1)}{\partial t}}{\frac{\partial G^{(t)}(\mu_2)}{\partial t}} = \Theta(1), \qquad \frac{\sum_{i \in [n]} g_i^{(t)}}{\sum_{i \in I_1} g_i^{(t)} - \sum_{i \in I_3} g_i^{(t)}} = O(1),$$

which proves the first result and

$$\frac{-\frac{\partial G^{(t)}(\mu_3)}{\partial t}}{\frac{\partial G^{(t)}(\mu_1)}{\partial t}} = \frac{-\sum_{i \in I_1} g_i^{(t)} + \sum_{i \in I_2 \cup I_3 \cup I_4} g_i^{(t)}}{\sum_{i \in I_1} g_i^{(t)} - \sum_{i \in I_2} g_i^{(t)}} + o(1).$$

Thus, to analyze $\frac{\partial}{\partial t} \frac{-\frac{\partial G^{(t)}(\mu_3)}{\partial t}}{\frac{\partial G^{(t)}(\mu_1)}{\partial t}}$, we can instead analyze

$$\frac{\partial}{\partial t} \frac{\sum_{i \in I_2 \cup I_3 \cup I_4} g_i^{(t)} - \sum_{i \in I_1} g_i^{(t)}}{\sum_{i \in I_1} g_i^{(t)} - \sum_{i \in I_2} g_i^{(t)}} \geq 0$$

$$\Leftrightarrow \frac{\partial}{\partial t} \left( \sum_{i \in I_2 \cup I_3 \cup I_4} g_i^{(t)} - \sum_{i \in I_1} g_i^{(t)} \right) \left( \sum_{i \in I_1} g_i^{(t)} - \sum_{i \in I_2} g_i^{(t)} \right)$$

$$- \left( \sum_{i \in I_2 \cup I_3 \cup I_4} g_i^{(t)} - \sum_{i \in I_1} g_i^{(t)} \right) \frac{\partial}{\partial t} \left( \sum_{i \in I_1} g_i^{(t)} - \sum_{i \in I_2} g_i^{(t)} \right) \geq 0$$

$$\Leftrightarrow \sum_{i\in I_2\cup I_3\cup I_4} \frac{\partial g_i^{(t)}}{\partial t} + R\sum_{i\in I_2} \frac{\partial g_i^{(t)}}{\partial t} \geq (1+R)\sum_{i\in I_1} \frac{\partial g_i^{(t)}}{\partial t}. \tag{15}$$

Recall that by Definition F.1, we have for $i_1 \in I_1$,

$$\frac{\partial y_{i_1} F_{i_1}^{(t)}}{\partial t} = \sum_{j_1=1}^{m_1}\sum_{j_2=1}^{m_1} \left\langle a_{j_1} w_{j_1}^{(t)}, a_{j_2} w_{j_2}^{(t)} \right\rangle \frac{1}{n}\left( 3\sum_{i\in I_1} g_i^{(t)} - \sum_{i\in I_2\cup I_3\cup I_4} g_i^{(t)} - \sum_{i\in I_2\cup I_3} g_i^{(t)} + o(1)\sum_{i\in[n]} g_i^{(t)} \right);$$

for $i_2 \in I_2$,

$$\frac{\partial y_{i_2} F_{i_2}^{(t)}}{\partial t} = -\sum_{j_1=1}^{m_1}\sum_{j_2=1}^{m_1} \left\langle a_{j_1} w_{j_1}^{(t)}, a_{j_2} w_{j_2}^{(t)} \right\rangle \frac{1}{n}\left( \sum_{i\in I_1} g_i^{(t)} - \sum_{i\in I_2\cup I_3\cup I_4} g_i^{(t)} + \sum_{i\in I_1} g_i^{(t)} - \sum_{i\in I_2} g_i^{(t)} + o(1)\sum_{i\in[n]} g_i^{(t)} \right);$$

for $i_3 \in I_3$,

$$\frac{\partial y_{i_3} F_{i_3}^{(t)}}{\partial t} = -\sum_{j_1=1}^{m_1}\sum_{j_2=1}^{m_1} \left\langle a_{j_1} w_{j_1}^{(t)}, a_{j_2} w_{j_2}^{(t)} \right\rangle \frac{1}{n}\left( \sum_{i\in I_1} g_i^{(t)} - \sum_{i\in I_2\cup I_3\cup I_4} g_i^{(t)} + \sum_{i\in I_1} g_i^{(t)} - \sum_{i\in I_3} g_i^{(t)} + o(1)\sum_{i\in[n]} g_i^{(t)} \right);$$

and for $i_4 \in I_4$,

$$\frac{\partial y_{i_4} F_{i_4}^{(t)}}{\partial t} = -\sum_{j_1=1}^{m_1}\sum_{j_2=1}^{m_1} \left\langle a_{j_1} w_{j_1}^{(t)}, a_{j_2} w_{j_2}^{(t)} \right\rangle \frac{1}{n}\left( \sum_{i\in I_1} g_i^{(t)} - \sum_{i\in I_2\cup I_3\cup I_4} g_i^{(t)} + o(1)\sum_{i\in[n]} g_i^{(t)} \right).$$

The above implies that for $i_1 \in I_1$,

$$\frac{\frac{\partial y_{i_1} F_{i_1}^{(t)}}{\partial t}}{\sum_{i\in I_1} g_i^{(t)} - \sum_{i\in I_2} g_i^{(t)}} = \sum_{j_1=1}^{m_1}\sum_{j_2=1}^{m_1} \left\langle a_{j_1} w_{j_1}^{(t)}, a_{j_2} w_{j_2}^{(t)} \right\rangle \frac{1}{n}\left( 1 + \frac{\sum_{i\in I_1} g_i^{(t)} - \sum_{i\in I_3} g_i^{(t)}}{\sum_{i\in I_1} g_i^{(t)} - \sum_{i\in I_2} g_i^{(t)}} - R + o(1) \right);$$

for $i_2 \in I_2$,

$$\frac{\frac{\partial y_{i_2} F_{i_2}^{(t)}}{\partial t}}{\sum_{i\in I_1} g_i^{(t)} - \sum_{i\in I_2} g_i^{(t)}} = -\sum_{j_1=1}^{m_1}\sum_{j_2=1}^{m_1} \left\langle a_{j_1} w_{j_1}^{(t)}, a_{j_2} w_{j_2}^{(t)} \right\rangle \frac{1}{n}\left( 1 - R + o(1) \right);$$

for $i_3 \in I_3$,

$$\frac{\frac{\partial y_{i_3} F_{i_3}^{(t)}}{\partial t}}{\sum_{i\in I_1} g_i^{(t)} - \sum_{i\in I_2} g_i^{(t)}} = -\sum_{j_1=1}^{m_1}\sum_{j_2=1}^{m_1} \left\langle a_{j_1} w_{j_1}^{(t)}, a_{j_2} w_{j_2}^{(t)} \right\rangle \frac{1}{n}\left( \frac{\sum_{i\in I_1} g_i^{(t)} - \sum_{i\in I_3} g_i^{(t)}}{\sum_{i\in I_1} g_i^{(t)} - \sum_{i\in I_2} g_i^{(t)}} - R + o(1) \right);$$

and for $i_4 \in I_4$,

$$\frac{\frac{\partial y_{i_1} F_{i_1}^{(t)}}{\partial t}}{\sum_{i\in I_1} g_i^{(t)} - \sum_{i\in I_2} g_i^{(t)}} = -\sum_{j_1=1}^{m_1}\sum_{j_2=1}^{m_1} \left\langle a_{j_1} w_{j_1}^{(t)}, a_{j_2} w_{j_2}^{(t)} \right\rangle \frac{1}{n}\left( -R + o(1) \right).$$

Substituting the above into Equation (15) and divide both sides by $\frac{1}{n}\sum_{j_1=1}^{m_1}\sum_{j_2=1}^{m_1} \left\langle a_{j_1} w_{j_1}^{(t)}, a_{j_2} w_{j_2}^{(t)} \right\rangle$, we have

$$(1+R)\sum_{i_2\in I_2} g'(F_{i_2}^{(t)})(R-1+o(1)) + \sum_{i_3\in I_3} g'(F_{i_3}^{(t)})\left( R - \frac{\sum_{i\in I_1} g_i^{(t)} - \sum_{i\in I_3} g_i^{(t)}}{\sum_{i\in I_1} g_i^{(t)} - \sum_{i\in I_2} g_i^{(t)}} + o(1) \right)$$

$$+ \sum_{i_4\in I_4} g'(F_{i_4}^{(t)})(R+o(1))$$

$$\geq (1+R)\sum_{i_1\in I_1} g'(F_{i_1}^{(t)})\left( 1 + \frac{\sum_{i\in I_1} g_i^{(t)} - \sum_{i\in I_3} g_i^{(t)}}{\sum_{i\in I_1} g_i^{(t)} - \sum_{i\in I_2} g_i^{(t)}} - R + o(1) \right). \tag{16}$$

By Lemma F.5, we obtain

$$\left| \frac{\sum_{i\in I_1} g_i^{(t)} - \sum_{i\in I_3} g_i^{(t)}}{\sum_{i\in I_1} g_i^{(t)} - \sum_{i\in I_2} g_i^{(t)}} - 1 \right| \le \varepsilon,$$

where we use $\varepsilon$ to denote the small deviation. Note that Equation (16) is a quadratic inequality in $R$ and can be rearranged as $aR^2 + bR + c \ge 0$, where

$$a = \sum_{i_1 \in I_1} g'(F_{i_1}^{(t)}) + \sum_{i_2 \in I_2} g'(F_{i_2}^{(t)}),$$

$$b = \sum_{i_3 \in I_3} g'(F_{i_3}^{(t)}) + \sum_{i_4 \in I_4} g'(F_{i_4}^{(t)}) - \sum_{i_1 \in I_1} g'(F_{i_1}^{(t)}),$$

$$c = (1 + o(1)) \left( -\sum_{i_2 \in I_2} g'(F_{i_2}^{(t)}) - (1 \pm \varepsilon) \sum_{i_3 \in I_3} g'(F_{i_3}^{(t)}) - (2 \pm \varepsilon) \sum_{i_1 \in I_1} g'(F_{i_1}^{(t)}) \right).$$

Now we analyze the equality condition in Equation (16) where we calculate the root of the equation $aR^2 + bR + c = 0$. We have

$$\frac{b}{a} = \frac{\sum_{i_3 \in I_3} g'(F_{i_3}^{(t)}) + \sum_{i_4 \in I_4} g'(F_{i_4}^{(t)}) - \sum_{i_1 \in I_1} g'(F_{i_1}^{(t)})}{\sum_{i_1 \in I_1} g'(F_{i_1}^{(t)}) + \sum_{i_2 \in I_2} g'(F_{i_2}^{(t)})},$$

$$\frac{c}{a} = \frac{(1 + o(1)) \left( -\sum_{i_2 \in I_2} g'(F_{i_2}^{(t)}) - (1 \pm \varepsilon) \sum_{i_3 \in I_3} g'(F_{i_3}^{(t)}) - (2 \pm \varepsilon) \sum_{i_1 \in I_1} g'(F_{i_1}^{(t)}) \right)}{\sum_{i_1 \in I_1} g'(F_{i_1}^{(t)}) + \sum_{i_2 \in I_2} g'(F_{i_2}^{(t)})}.$$

Note that

$$\sum_{i \in I} g_i^{(t)} \min_{i \in I} \frac{1}{1 + \exp(y_i F_i^{(t)})} \le -\sum_{i \in I} g'(F_i^{(t)}) \le \sum_{i \in I} g_i^{(t)} \max_{i \in I} \frac{1}{1 + \exp(y_i F_i^{(t)})}.$$

By Equation (14), we have $-\frac{b}{2a} \ge -1/4$ and

$$\left| \frac{b}{a} \right| \le (1 + O(\max_{i \in [n]} \exp(-y_i F_i^{(t)}))) \max \left( \left| \frac{\sum_{i_1 \in I_1} g(F_{i_1}^{(t)}) - \sum_{i_3 \in I_3} g(F_{i_3}^{(t)})}{\sum_{i_1 \in I_1} g(F_{i_1}^{(t)}) + \sum_{i_2 \in I_2} g(F_{i_2}^{(t)})} \right|, \left| \frac{\sum_{i_4 \in I_4} g(F_{i_4}^{(t)})}{\sum_{i_1 \in I_1} g(F_{i_1}^{(t)}) + \sum_{i_2 \in I_2} g(F_{i_2}^{(t)})} \right| \right)$$

$$\le C < 1,$$

and

$$\left| \frac{c}{a} \right| = (1 \pm O(\varepsilon) \pm O(\max_{i \in [n]} \exp(-y_i F_i^{(t)}))) \cdot 2.$$

Recall that we are considering the case of $R \in [1, 3]$. Thus, we only need to consider the root that is positive and we can calculate the root $R^\star = -\frac{b}{2a} + \sqrt{\frac{b^2}{4a^2} - \frac{c}{a}} \in (1, 2)$ if $\varepsilon$ and $\max_{i \in [n]} \exp(-y_i F_i^{(t)})$ are both suffiently small.

Next, since $g'(F_i^{(t)}) < 0$, the root $R^\star$ is contractive (i.e., if $R(t) > R^\star$ then $R(t)$ is decreasing and if $R(t) < R^\star$ then $R(t)$ is increasing).

Finally, the result at the end of Phase 1 implies that $R(T_1) \in [1, 3]$, which completes the proof. $\quad\square$

**Corollary F.7.** *For $t \in [T_1, T_2]$, we have*

$$\frac{G^{(t)}(\mu_1)}{G^{(t)}(\mu_2)}, \frac{G^{(t)}(\mu_1)}{-G^{(t)}(\mu_3)}, \frac{G^{(t)}(\mu_2)}{-G^{(t)}(\mu_3)} = \Theta(1).$$

*Proof.* We first prove $\frac{G^{(t)}(\mu_1)}{G^{(t)}(\mu_2)} = \Theta(1)$. Following from Lemma F.6, we have $\frac{\frac{\partial G^{(t)}(\mu_1)}{\partial t}}{\frac{\partial G^{(t)}(\mu_2)}{\partial t}} = \Theta(1)$. By Theorem E.27, we have $\frac{G^{(T_1)}(\mu_1)}{G^{(T_1)}(\mu_2)} = \Theta(1)$. Thus, for $t \in [T_1, T_2]$, we obtain

$$\frac{G^{(t)}(\mu_1)}{G^{(t)}(\mu_2)} = \frac{G^{(T_1)}(\mu_1) + \int_{T_1}^t \frac{\partial G^{(\tau)}(\mu_1)}{\partial \tau} d\tau}{G^{(T_1)}(\mu_2) + \int_{T_1}^t \frac{\partial G^{(\tau)}(\mu_2)}{\partial \tau} d\tau} = \Theta(1).$$

Note that Lemma F.6 and Definition F.1 imply that $\frac{\frac{\partial G^{(t)}(\mu_1)}{\partial t}}{-\frac{\partial G^{(t)}(\mu_3)}{\partial t}} = \Theta(1)$. Since $\frac{G^{(T_1)}(\mu_1)}{-G^{(T_1)}(\mu_3)} = \Theta(1)$, similarly, we have $\frac{G^{(t)}(\mu_1)}{-G^{(t)}(\mu_3)} = \Theta(1)$. Finally, $\frac{G^{(t)}(\mu_1)}{G^{(t)}(\mu_2)} = \Theta(1)$ and $\frac{G^{(t)}(\mu_1)}{-G^{(t)}(\mu_3)} = \Theta(1)$ imply that $\frac{G^{(t)}(\mu_2)}{-G^{(t)}(\mu_3)} = \Theta(1)$. $\qquad\square$

### F.2 How Fast the Loss Decreases

**Theorem F.8.** *For $t \in [T_1, T_2]$, we have*

$$\widehat{L}(t) = \frac{1}{\Theta(\sigma_1^2 m m_1)(t - T_1) + (1/\widehat{L}(T_1))}.$$

*Proof.* First of all, the gradient flow update for the empirical loss is given by $\frac{\partial \widehat{L}}{\partial t} = \sum_{i=1}^{n} \ell'(y_i F_i^{(t)}) \frac{\partial y_i F_i^{(t)}}{\partial t}$. By Definition F.1, we have for $i_1 \in I_1$,

$$\frac{\partial y_{i_1} F_{i_1}^{(t)}}{\partial t} = \sum_{j_1=1}^{m_1} \sum_{j_2=1}^{m_1} \left\langle a_{j_1} w_{j_1}^{(t)}, a_{j_2} w_{j_2}^{(t)} \right\rangle \frac{1}{n} \left( 3 \sum_{i \in I_1} g_i^{(t)} - \sum_{i \in I_2 \cup I_3 \cup I_4} g_i^{(t)} - \sum_{i \in I_2 \cup I_3} g_i^{(t)} + o(1) \sum_{i \in [n]} g_i^{(t)} \right);$$

for $i_2 \in I_2$,

$$\frac{\partial y_{i_2} F_{i_2}^{(t)}}{\partial t} = -\sum_{j_1=1}^{m_1} \sum_{j_2=1}^{m_1} \left\langle a_{j_1} w_{j_1}^{(t)}, a_{j_2} w_{j_2}^{(t)} \right\rangle \frac{1}{n} \left( \sum_{i \in I_1} g_i^{(t)} - \sum_{i \in I_2 \cup I_3 \cup I_4} g_i^{(t)} + \sum_{i \in I_1} g_i^{(t)} - \sum_{i \in I_2} g_i^{(t)} + o(1) \sum_{i \in [n]} g_i^{(t)} \right);$$

for $i_3 \in I_3$,

$$\frac{\partial y_{i_3} F_{i_3}^{(t)}}{\partial t} = -\sum_{j_1=1}^{m_1} \sum_{j_2=1}^{m_1} \left\langle a_{j_1} w_{j_1}^{(t)}, a_{j_2} w_{j_2}^{(t)} \right\rangle \frac{1}{n} \left( \sum_{i \in I_1} g_i^{(t)} - \sum_{i \in I_2 \cup I_3 \cup I_4} g_i^{(t)} + \sum_{i \in I_1} g_i^{(t)} - \sum_{i \in I_3} g_i^{(t)} + o(1) \sum_{i \in [n]} g_i^{(t)} \right);$$

and for $i_4 \in I_4$,

$$\frac{\partial y_{i_4} F_{i_4}^{(t)}}{\partial t} = -\sum_{j_1=1}^{m_1} \sum_{j_2=1}^{m_1} \left\langle a_{j_1} w_{j_1}^{(t)}, a_{j_2} w_{j_2}^{(t)} \right\rangle \frac{1}{n} \left( \sum_{i \in I_1} g_i^{(t)} - \sum_{i \in I_2 \cup I_3 \cup I_4} g_i^{(t)} + o(1) \sum_{i \in [n]} g_i^{(t)} \right).$$

By Lemma F.6, we have

$$\frac{\partial y_i F_i^{(t)}}{\partial t} = \sum_{j_1=1}^{m_1} \sum_{j_2=1}^{m_1} \left\langle a_{j_1} w_{j_1}^{(t)}, a_{j_2} w_{j_2}^{(t)} \right\rangle \Theta \left( \frac{1}{n} \sum_{i \in [n]} g_i^{(t)} \right)$$

for all $i \in [n]$. Therefore,

$$\frac{\partial \widehat{L}}{\partial t} = \frac{1}{n} \sum_{i=1}^{n} \ell'(y_i F_i^{(t)}) \frac{\partial y_i F_i^{(t)}}{\partial t}$$

$$= \frac{1}{n} \sum_{i=1}^{n} -g_i^{(t)} \sum_{j_1=1}^{m_1} \sum_{j_2=1}^{m_1} \left\langle a_{j_1} w_{j_1}^{(t)}, a_{j_2} w_{j_2}^{(t)} \right\rangle \Theta \left( \frac{1}{n} \sum_{i' \in [n]} g_{i'}^{(t)} \right)$$

$$= -\sum_{j_1=1}^{m_1} \sum_{j_2=1}^{m_1} \left\langle a_{j_1} w_{j_1}^{(t)}, a_{j_2} w_{j_2}^{(t)} \right\rangle \Theta \left( \left( \frac{1}{n} \sum_{i \in [n]} g_i^{(t)} \right)^2 \right)$$

$$= -\sum_{j_1=1}^{m_1} \sum_{j_2=1}^{m_1} \left\langle a_{j_1} w_{j_1}^{(t)}, a_{j_2} w_{j_2}^{(t)} \right\rangle \Theta \left( \widehat{L}^2 \right),$$

where the last equality follows from the property of binary cross-entropy loss that $\ell(x) = \Theta(-\ell'(x))$ for $x > 0$. By Definition F.1 and Lemma F.4, we have for all $t \in [T_1, T_2]$,

$$\sum_{j_1=1}^{m_1} \sum_{j_2=1}^{m_1} \left\langle a_{j_1} w_{j_1}^{(t)}, a_{j_2} w_{j_2}^{(t)} \right\rangle = \Theta(\sigma_1^2 m m_1).$$

Thus, we have

$$\frac{\partial \widehat{L}}{\partial t} = -\Theta(\sigma_1^2 m m_1) \widehat{L}^2.$$

Now, consider the differential equation $\frac{dL}{dt} = -C_1 L^2$. Note that this is a separable differential equation in $t$ and we can solve it by

$$\frac{1}{L^2} \frac{dL}{dt} + C_1 = 0 \quad \Rightarrow \quad \frac{d}{dt}\left(C_1 t - L^{-1} + C_2\right) = 0 \quad \Rightarrow \quad L(t) = \frac{1}{C_1 t + C_2}.$$

This implies for $t \in [T_1, T_2]$, we have

$$\widehat{L}(t) = \frac{1}{\Theta(\sigma_1^2 m m_1)(t - T_1) + (1/\widehat{L}(T_1))}$$

$\square$

**Corollary F.9.** *The following bound holds:*

$$\left| \sum_{j_1=1}^{m_1} \sum_{j_2=1}^{m_1} \left\langle a_{j_1} w_{j_1}^{(T_2)}, a_{j_2} w_{j_2}^{(T_2)} \right\rangle - \sum_{j_1=1}^{m_1} \sum_{j_2=1}^{m_1} \left\langle a_{j_1} w_{j_1}^{(T_1)}, a_{j_2} w_{j_2}^{(T_1)} \right\rangle \right| \leq \widetilde{O}\left(\frac{m_1}{m}\right).$$

*Proof.* This is a direct consequence of Lemma F.4 and Theorem F.8. $\square$

### F.3 Growth of Neuron Correlation

**Lemma F.10.** *For $t \in [T_1, T_2]$, we have*

$$\sum_{j_1=1}^{m_1} \sum_{j_2=1}^{m_1} \left\langle a_{j_1} w_{j_1}^{(t)}, a_{j_2} w_{j_2}^{(t)} \right\rangle - \sum_{j_1=1}^{m_1} \sum_{j_2=1}^{m_1} \left\langle a_{j_1} w_{j_1}^{(T_1)}, a_{j_2} w_{j_2}^{(T_1)} \right\rangle$$

$$= O\left(\frac{m_1 \log m \log t}{\sigma_1^2 m m_1}\right) = O\left(\frac{m_1 \log^2 m}{\sigma_1^2 m m_1}\right)$$

*and thus,*

$$\sum_{j_1=1}^{m_1} \sum_{j_2=1}^{m_1} \left\langle a_{j_1} w_{j_1}^{(t)}, a_{j_2} w_{j_2}^{(t)} \right\rangle = \Theta(\sigma_1^2 m m_1).$$

*Proof.* By Lemma F.4, we have

$$\frac{\partial}{\partial t} \sum_{j_1=1}^{m_1} \sum_{j_2=1}^{m_1} \left\langle a_{j_1} w_{j_1}^{(t)}, a_{j_2} w_{j_2}^{(t)} \right\rangle = \frac{2m_1}{n} \sum_{i=1}^{n} g_i^{(t)} y_i F_i^{(t)}.$$

By Theorem F.8, we have $\frac{1}{n} \sum_{i=1}^{n} g_i^{(t)} = O(\widehat{L}^{(t)}) = O\left(\frac{1}{(\sigma_1^2 m m_1)(t - T_1) + 1/\widehat{L}(T_1)}\right)$. Further, by Definition F.1, we have $|F_i^{(t)}| \leq O(\log m)$. Thus, for $t \in [T_1, T_2]$, we obtain

$$\sum_{j_1=1}^{m_1} \sum_{j_2=1}^{m_1} \left\langle a_{j_1} w_{j_1}^{(t)}, a_{j_2} w_{j_2}^{(t)} \right\rangle - \sum_{j_1=1}^{m_1} \sum_{j_2=1}^{m_1} \left\langle a_{j_1} w_{j_1}^{(T_1)}, a_{j_2} w_{j_2}^{(T_1)} \right\rangle$$

$$= 2m_1 \int_{T_1}^{t} \frac{1}{n} \sum_{i=1}^{n} g_i^{(\tau)} y_i F_i^{(\tau)} \, d\tau$$

$$\leq O(m_1 \log m) \int_{T_1}^t O\left(\frac{1}{(\sigma_1^2 mm_1)(\tau - T_1) + 1/\widehat{L}(T_1)}\right) d\tau$$

$$= O\left(\frac{m_1 \log m \log t}{\sigma_1^2 mm_1}\right) = O\left(\frac{m_1 \log^2 m}{\sigma_1^2 mm_1}\right)$$

where the last line follows because $T_2 \leq O(\text{poly}(m))$ in Definition F.1. Finally, by Theorem E.27 we have

$$\sum_{j_1=1}^{m_1} \sum_{j_2=1}^{m_1} \left\langle a_{j_1} w_{j_1}^{(T_1)}, a_{j_2} w_{j_2}^{(T_1)} \right\rangle = \Theta(\sigma_1^2 mm_1).$$

$\square$

### F.4   Growth of Correlation of Value-Transformed Data

We now analyze the correlation term with value-transformed data.

**Lemma F.11** (Growth of correlation of value-transformed data). *For $\mu, \nu \in \{\mu_i\}_{i=1}^d$, we have*

$$\frac{\partial}{\partial t} \mu^\top W_V^{(t)\top} W_V^{(t)} \nu = G^{(t)}(\nu) \frac{1}{n} \sum_{i:\mu \in X^{(i)}} g_i^{(t)} y_i \sum_{l=1}^L p_{q \leftarrow l, k \leftarrow \mu}^{(t,i)} + G^{(t)}(\mu) \frac{1}{n} \sum_{i:\nu \in X^{(i)}} g_i^{(t)} y_i \sum_{l=1}^L p_{q \leftarrow l, k \leftarrow \nu}^{(t,i)}.$$

*Thus, for $t \in [T_1, T_2]$, we have*

$$\max_{\mu, \nu} \left| \mu^\top W_V^{(t)\top} W_V^{(t)} \nu - \mu^\top W_V^{(T_1)\top} W_V^{(T_1)} \nu \right| \leq O\left(\frac{\log m \log t}{\sigma_1^2 mm_1}\right).$$

*Proof.* By the gradient flow update, we have

$$\frac{\partial \mu W_V^{(t)\top} W_V^{(t)} \nu}{\partial t}$$

$$= \frac{1}{n} \sum_{i:\mu \in X^{(i)}} g_i^{(t)} y_i \sum_{l=1}^L \sum_{j=1}^{m_1} a_j \nu^\top W_V^{(t)\top} w_j^{(t)} p_{q \leftarrow l, k \leftarrow \mu}^{(t,i)} + \frac{1}{n} \sum_{i:\nu \in X^{(i)}} g_i^{(t)} y_i \sum_{l=1}^L \sum_{j=1}^{m_1} a_j \mu^\top W_V^{(t)\top} w_j^{(t)} p_{q \leftarrow l, k \leftarrow \nu}^{(t,i)}$$

$$= G^{(t)}(\nu) \frac{1}{n} \sum_{i:\mu \in X^{(i)}} g_i^{(t)} y_i \sum_{l=1}^L p_{q \leftarrow l, k \leftarrow \mu}^{(t,i)} + G^{(t)}(\mu) \frac{1}{n} \sum_{i:\nu \in X^{(i)}} g_i^{(t)} y_i \sum_{l=1}^L p_{q \leftarrow l, k \leftarrow \nu}^{(t,i)}.$$

Thus, we obtain

$$\left| \mu W_V^{(\tau)\top} W_V^{(\tau)} \nu - \mu W_V^{(T_1)\top} W_V^{(T_1)} \nu \right|$$

$$\leq \int_{T_1}^\tau |G^{(t)}(\nu)| \left| \frac{1}{n} \sum_{i:\mu \in X^{(i)}} g_i^{(t)} y_i \right| \left| \sum_{l=1}^L p_{q \leftarrow l, k \leftarrow \mu}^{(t,i)} \right| + |G^{(t)}(\mu)| \left| \frac{1}{n} \sum_{i:\nu \in X^{(i)}} g_i^{(t)} y_i \right| \left| \sum_{l=1}^L p_{q \leftarrow l, k \leftarrow \nu}^{(t,i)} \right| dt.$$

By Definition F.1, for $t \in [T_1, T_2]$, we have $|G^{(t)}(\mu)| \leq O(\log m)$ and $\sum_{l=1}^L p_{q \leftarrow l, k \leftarrow \mu}^{(t,i)} \leq O(1)$. Further, by the property of the cross-entropy loss, we have

$$\frac{1}{n} \sum_{i:\mu \in X^{(i)}} g_i^{(t)} y_i = O(\widehat{L}^{(t)}).$$

Therefore, by Theorem F.8, we obtain

$$\left| \mu W_V^{(\tau)\top} W_V^{(\tau)} \nu - \mu W_V^{(T_1)\top} W_V^{(T_1)} \nu \right| \leq O(\log m) \int_{T_1}^\tau \widehat{L}^{(t)} dt \leq O(\log m) \cdot O\left(\frac{\log \tau}{\sigma_1^2 mm_1}\right).$$

$\square$

**Corollary F.12.** *For $t \in [T_1, T_2]$, we have*

$$\frac{\max_{\mu \in \{\mu_i\}_{i=1}^d} \left| \sum_{j=1}^{m_1} a_j \frac{\partial w_j^{(t)}}{\partial t} W_V^{(t)} \mu \right|}{\sum_{j_1=1}^{m_1} \sum_{j_2=1}^{m_1} \left\langle a_{j_1} w_{j_1}^{(t)}, a_{j_2} w_{j_2}^{(t)} \right\rangle} \leq o(1/L) \frac{1}{n} \sum_{i \in [n]} g_i^{(t)}.$$

*Proof.* By Lemma E.9 and and Lemma F.11, for all $\mu \neq \nu \in \{\mu_i\}_{i=1}^d$, we have $|\mu^\top W_V^{(t)\top} W_V^{(t)} \nu| \leq \widetilde{O}(\sigma_0^2 \sqrt{m} + 1/m)$ and $\|W_V^{(t)} \nu\|_2^2 = \widetilde{O}(\sigma_0^2 m + 1/m)$. Thus, by Lemma F.11, we have

$$\frac{\max_{\mu, \nu} \left| \mu^\top W_V^{(t)\top} W_V^{(t)} \nu \right|}{\sum_{j_1=1}^{m_1} \sum_{j_2=1}^{m_1} \left\langle a_{j_1} w_{j_1}^{(t)}, a_{j_2} w_{j_2}^{(t)} \right\rangle} \leq \widetilde{O} \left( \frac{\sigma_0^2 \sqrt{m} + 1/m}{\sigma_1^2 m m_1} \right),$$

$$\frac{\max_{\nu} \| W_V^{(t)} \nu \|_2^2}{\sum_{j_1=1}^{m_1} \sum_{j_2=1}^{m_1} \left\langle a_{j_1} w_{j_1}^{(t)}, a_{j_2} w_{j_2}^{(t)} \right\rangle} \leq \widetilde{O} \left( \frac{\sigma_0^2 m + 1/m}{\sigma_1^2 m m_1} \right).$$

Recall that

$$\sum_{j=1}^{m_1} a_j \frac{\partial w_j^{(t)}}{\partial t} W_V^{(t)} \mu = \frac{m_1}{n} \sum_{i_1=1}^{n} g_{i_1}^{(t)} y_{i_1} \sum_{l_1=1}^{L} p_{l_1}^{(t,i_1)\top} V^{(t,i_1)\top} W_V^{(t)} \mu.$$

This implies that

$$\frac{\max_{\mu \in \{\mu_i\}_{i=1}^d} \left| \sum_{j=1}^{m_1} a_j \frac{\partial w_j^{(t)}}{\partial t} W_V^{(t)} \mu \right|}{\sum_{j_1=1}^{m_1} \sum_{j_2=1}^{m_1} \left\langle a_{j_1} w_{j_1}^{(t)}, a_{j_2} w_{j_2}^{(t)} \right\rangle} \leq \widetilde{O} \left( \frac{\sigma_0^2 \sqrt{m} L + L/m + \sigma_0^2 m + 1/m}{\sigma_1^2 m} \right) \cdot \frac{1}{n} \sum_{i=1}^{n} g_i^{(t)}.$$

$\square$

**Corollary F.13** (Complete version of Corollary 4.6). *For $t \in [T_1, T_2]$, we have*

$$\frac{\partial}{\partial t} \mu_2^\top W_V^{(t)\top} W_V^{(t)} \mu_1 > 0, \qquad \frac{\partial}{\partial t} \mu_1^\top W_V^{(t)\top} W_V^{(t)} \mu_1 > 0, \qquad \frac{\partial}{\partial t} \mu_2^\top W_V^{(t)\top} W_V^{(t)} \mu_2 > 0$$

$$\frac{\partial}{\partial t} \mu_1^\top W_V^{(t)\top} W_V^{(t)} \mu_3 < 0, \qquad \frac{\partial}{\partial t} \mu_2^\top W_V^{(t)\top} W_V^{(t)} \mu_3 < 0.$$

*Proof.* This is a direct consequence of Lemma F.11, Lemma F.6 and Definition F.1. $\square$

### F.5  Change of Random-Token Sub-Network

**Lemma F.14.** *For $t \in [T_1, T_2]$ and $\mu \in \{\mu_i\}_{i=4}^d$, we have*

$$\left| \frac{\partial G^{(t)}(\mu)}{\partial t} \right| \leq O \left( \frac{1}{n} \widehat{L}^{(t)} \sigma_1^2 m m_1 \right) + \widetilde{O}(\widehat{L}^{(t)} m_1 (L(\sigma_0^2 \sqrt{m} + 1/m) + \sigma_0^2 m)).$$

*Thus,*

$$\left| G^{(t)}(\mu) - G^{(T_1)}(\mu) \right| \leq \widetilde{O} \left( \frac{1}{n} + \frac{m_1 (L(\sigma_0^2 \sqrt{m} + 1/m) + \sigma_0^2 m)}{\sigma_1^2 m m_1} \right).$$

*Proof.* By Lemma C.1 and Definition F.1, we have

$$\left| \frac{\partial G^{(t)}(\mu)}{\partial t} \right| = \left| \sum_{j=1}^{m_1} a_j w_j^{(t)} \frac{\partial W_V^{(t)} \mu}{\partial t} + a_j \frac{\partial w_j^{(t)}}{\partial t} W_V^{(t)} \mu \right|$$

$$\leq \left| \frac{1}{n} \sum_{i_2: \, \mu \in X^{(i_2)}} g_{i_2}^{(t)} y_{i_2} \sum_{l_2=1}^{L} \sum_{j_1=1}^{m_1} \sum_{j_2=1}^{m_1} a_{j_1} a_{j_2} \left\langle w_{j_1}^{(t)}, w_{j_2}^{(t)} \right\rangle p_{q \leftarrow l_2, k \leftarrow \mu}^{(t, i_2)} \right|$$

$$+ \left| \frac{1}{n} \sum_{i_1=1}^{n} g_{i_1}^{(t)} y_{i_1} m_1 \sum_{l_1=1}^{L} \sum_{l_2=1}^{L} \left\langle p_{l_1,l_2}^{(t,i_1)} v_{l_2}^{(t,i_1)}, v^{(t)}(\mu) \right\rangle \right|.$$

By Definition F.1 and Corollary F.2,

$$\left| \frac{1}{n} \sum_{i_2: \ \mu \in X^{(i_2)}} g_{i_2}^{(t)} y_{i_2} \sum_{l_2=1}^{L} \sum_{j_1=1}^{m_1} \sum_{j_2=1}^{m_1} a_{j_1} a_{j_2} \left\langle w_{j_1}^{(t)}, w_{j_2}^{(t)} \right\rangle p_{q \leftarrow l_2, k \leftarrow \mu}^{(t,i_2)} \right| \leq O\left( \frac{1}{n} \widehat{L}^{(t)} \sigma_1^2 m m_1 \right).$$

By Lemma E.9 and and Lemma F.11, we have $|\mu^\top W_V^{(t)\top} W_V^{(t)} \nu| \leq \widetilde{O}(\sigma_0^2 \sqrt{m} + 1/m)$ for $\mu \neq \nu$ and $\|W_V^{(t)} \mu\|_2^2 = O(\sigma_0^2 m)$. Thus,

$$\left| \frac{1}{n} \sum_{i_1=1}^{n} g_{i_1}^{(t)} y_{i_1} m_1 \sum_{l_1=1}^{L} \sum_{l_2=1}^{L} \left\langle p_{l_1,l_2}^{(t,i_1)} v_{l_2}^{(t,i_1)}, v^{(t)}(\mu) \right\rangle \right| \leq \widetilde{O}(\widehat{L}^{(t)} m_1 (L(\sigma_0^2 \sqrt{m} + 1/m) + \sigma_0^2 m)).$$

Thus, by Theorem F.8, for $t \in [T_1, T_2]$, we have

$$\left| G^{(t)}(\mu) - G^{(T_1)}(\mu) \right| = \left| \int_{T_1}^{t} \frac{\partial G^{(\tau)}(\mu)}{\partial \tau} \, d\tau \right| \leq \left( O\left( \frac{\sigma_1^2 m m_1}{n} \right) + \widetilde{O}(m_1 L(\sigma_0^2 \sqrt{m} + 1/m)) \right) \int_{T_1}^{t} \widehat{L}^{(\tau)} \, d\tau$$

$$\leq \widetilde{O}\left( \frac{1}{n} + \frac{m_1(L(\sigma_0^2 \sqrt{m} + 1/m) + \sigma_0^2 m)}{\sigma_1^2 m m_1} \right).$$

$\square$

**Corollary F.15.** *For $t \in [T_1, T_2]$, we have*

$$\left| \sum_{l_1=1}^{L} \sum_{l_2: \ X_{l_2}^{(i)} \in \{\mu_k\}_{k=4}^{d}} G^{(t)}(X_{l_2}^{(i)}) p_{q \leftarrow l_1, k \leftarrow l_2}^{(t,i)} \right| \leq O(1).$$

*Proof.* By Lemma F.3, we have

$$\left| \sum_{l_1=1}^{L} \sum_{l_2: \ X_{l_2}^{(i)} \in \{\mu_k\}_{k=4}^{d}} G^{(T_1)}(X_{l_2}^{(i)}) p_{q \leftarrow l_1, k \leftarrow l_2}^{(T_1,i)} \right| \leq O(1)$$

By Lemma F.14, for $t \in [T_1, T_2]$, $G^{(t)}(\mu) = O(1)$ for $\mu \in \{\mu_i\}_{i=4}^{d}$. On the other hand, by the triangle inequality, we have

$$\left| \sum_{l_1=1}^{L} \sum_{l_2: \ X_{l_2}^{(i)} \in \{\mu_k\}_{k=4}^{d}} G^{(T_1)}(X_{l_2}^{(i)}) p_{q \leftarrow l_1, k \leftarrow l_2}^{(T_1,i)} - \sum_{l_1=1}^{L} \sum_{l_2: \ X_{l_2}^{(i)} \in \{\mu_k\}_{k=4}^{d}} G^{(t)}(X_{l_2}^{(i)}) p_{q \leftarrow l_1, k \leftarrow l_2}^{(t,i)} \right|$$

$$\leq \sum_{l_1=1}^{L} \sum_{l_2: \ X_{l_2}^{(i)} \in \{\mu_k\}_{k=4}^{d}} \left| (G^{(T_1)}(X_{l_2}^{(i)}) - G^{(t)}(X_{l_2}^{(i)})) \right| p_{q \leftarrow l_1, k \leftarrow l_2}^{(t,i)}$$

$$+ \sum_{l_1=1}^{L} \sum_{l_2: \ X_{l_2}^{(i)} \in \{\mu_k\}_{k=4}^{d}} G^{(t)}(X_{l_2}^{(i)}) \left| p_{q \leftarrow l_1, k \leftarrow l_2}^{(T_1,i)} - p_{q \leftarrow l_1, k \leftarrow l_2}^{(t,i)} \right|$$

$$+ \sum_{l_1=1}^{L} \sum_{l_2: \ X_{l_2}^{(i)} \in \{\mu_k\}_{k=4}^{d}} \left| G^{(T_1)}(X_{l_2}^{(i)}) - G^{(t)}(X_{l_2}^{(i)}) \right| \left| p_{q \leftarrow l_1, k \leftarrow l_2}^{(T_1,i)} - p_{q \leftarrow l_1, k \leftarrow l_2}^{(t,i)} \right|$$

$$\leq O(1),$$

where the last inequality applies Definition F.1. This implies that, for $t \in [T_1, T_2]$, by Lemma F.14, we have

$$\left| \sum_{l_1=1}^{L} \sum_{l_2:\, X_{l_2}^{(i)} \in \{\mu_k\}_{k=4}^{d}} G^{(t)}(X_{l_2}^{(i)}) p_{q \leftarrow l_1, k \leftarrow l_2}^{(t,i)} \right| \leq O(1).$$

$\qquad\qquad\qquad\qquad\qquad\qquad\qquad\qquad\qquad\qquad\qquad\qquad\qquad\qquad\qquad\quad \square$

## F.6 Change of Score and Softmax Probability

**Lemma F.16** (Change of score, complete version of Lemma 4.7). *For $t \in [T_1, T_2]$, the attention scores are changing in the following way:*

- *For $\mu, \nu \in \{\mu_1, \mu_2\}$, $\mu \neq \nu$, the query-key-correlation score between the two target signals increases, while the query-key-correlation score between one target signal and the common token decreases, i.e.,*

$$\frac{\partial}{\partial t} \nu^\top W_K^{(t)\top} W_Q^{(t)} \mu = \frac{1}{\sqrt{m}} \widetilde{\Theta}(\widehat{L}^{(t)} \sigma_0^2 m) \frac{1}{L},$$

$$\frac{\partial}{\partial t} \mu_3^\top W_K^{(t)\top} W_Q^{(t)} \mu = -\frac{1}{\sqrt{m}} \widetilde{\Theta}(\widehat{L}^{(t)} \sigma_0^2 m) \frac{1}{L}.$$

- *The change of score satisfies:*

$$\max_{\mu, \nu \in \{\mu_i\}_{i=1}^3} \left| \nu^\top W_K^{(t)} W_Q^{(t)} \mu - \nu^\top W_K^{(T_1)} W_Q^{(T_1)} \mu \right| = \Theta\left( \frac{\sigma_0^2 m}{\sqrt{m} L \sigma_1^2 m m_1} + \frac{\sigma_0^2 \sqrt{m}}{\sqrt{m} \sigma_1^2 m m_1} \right).$$

- *For all $\mu \in \{\mu_1, \mu_2, \mu_3\}$, $\gamma \in \{\mu_i\}_{i=4}^{d}$, the query-key-correlation score changes as follows:*

$$\left| \frac{\partial}{\partial t} \mu^\top W_K^{(t)\top} W_Q^{(t)} \gamma \right| = \frac{1}{n\sqrt{m}} \widetilde{\Theta}(\widehat{L}^{(t)} \sigma_0^2 m) \frac{1}{L} + \widetilde{O}\left( \frac{1}{\sqrt{m}} \widehat{L}^{(t)} \sigma_0^2 \sqrt{m} \right),$$

*and*

$$\left| \mu^\top W_K^{(t)\top} W_Q^{(t)} \gamma - \mu^\top W_K^{(T_1)\top} W_Q^{(T_1)} \gamma \right| \leq \widetilde{O}\left( \frac{\sigma_0^2 m}{n\sqrt{m} L \sigma_1^2 m m_1} + \frac{\sigma_0^2 \sqrt{m}}{\sqrt{m} \sigma_1^2 m m_1} \right),$$

$$\left| \gamma^\top W_K^{(t)\top} W_Q^{(t)} \mu - \gamma^\top W_K^{(T_1)\top} W_Q^{(T_1)} \mu \right| \leq \widetilde{O}\left( \frac{\sigma_0^2 m}{n\sqrt{m} L \sigma_1^2 m m_1} + \frac{\sigma_0^2 \sqrt{m}}{\sqrt{m} \sigma_1^2 m m_1} \right).$$

*Proof.* By Lemma C.4, we have

$$\frac{\partial \nu^\top W_K^{(t)\top} W_Q^{(t)} \mu}{\partial t}$$

$$= \frac{1}{n\sqrt{m}} \sum_{i:\mu,\nu \in X^{(i)}} g_i^{(t)} y_i \sum_{j=1}^{m_1} a_j \|k^{(t)}(\nu)\|_2^2 \left( v^{(t)\top}(\nu) w_j^{(t)} - w_j^{(t)\top} V^{(t,i)} p_{l(i,\mu)}^{(t,i)} \right) p_{q \leftarrow \mu, k \leftarrow \nu}^{(t,i)}$$

$$+ \frac{1}{n\sqrt{m}} \sum_{i:\mu \in X^{(i)}} g_i^{(t)} y_i \sum_{j=1}^{m_1} a_j \sum_{l=1}^{L} \nu^\top W_K^{(t)\top} K_l^{(t,i)} \left( V_l^{(t,i)\top} w_j^{(t)} - w_j^{(t)\top} V^{(t,i)} p_{l(i,\mu)}^{(t,i)} \right) p_{q \leftarrow \mu, k \leftarrow l}^{(t,i)} \mathbb{I}(K_l^{(t,i)} \neq k^{(t)}(\nu))$$

$$+ \frac{1}{n\sqrt{m}} \sum_{i:\nu,\mu \in X^{(i)}} g_i^{(t)} y_i \sum_{j=1}^{m_1} a_j \|q^{(t)}(\mu)\|_2^2 p_{q \leftarrow \mu, k \leftarrow \nu}^{(t,i)} \left( w_j^{(t)\top} v^{(t,i)}(\nu) - w_j^{(t)\top} V^{(t,i)} p_{l(i,\mu)}^{(t,i)} \right)$$

$$+ \frac{1}{n\sqrt{m}} \sum_{i:\nu \in X^{(i)}} g_i^{(t)} y_i \sum_{l=1}^{L} \sum_{j=1}^{m_1} a_j \mu^\top W_Q^{(t)\top} q_l^{(t,i)} p_{q \leftarrow l, k \leftarrow \nu}^{(t,i)} \left( w_j^{(t)\top} v^{(t,i)}(\nu) - w_j^{(t)\top} V^{(t,i)} p_l^{(t,i)} \right) \mathbb{I}(q_l^{(t,i)} \neq q^{(t)}(\mu)).$$

Now, we take $\mu = \mu_1$, $\nu = \mu_2$. By Theorem E.27, we have $G^{(T_1)}(\mu_2) \geq \Omega(1)$. A consequence of Definition F.1 and Lemma F.6 is that $\frac{\partial G^{(t)}(\mu_2)}{\partial t} > 0$ for $t \in [T_1, T_2]$. Thus, we have $G^{(t)}(\mu_2) \geq \Omega(1)$.

Further by Theorem E.27, we have $G^{(T_1)}(\mu) = \widetilde{O}(\sigma_0\sigma_1\sqrt{mm_1})$ for $\mu \in \mathcal{R}$ and then by Lemma F.14, we have $G^{(t)}(\mu) = \widetilde{O}(\sigma_0\sigma_1\sqrt{mm_1}) + \widetilde{O}\left(\frac{1}{n} + \frac{m_1(L(\sigma_0^2\sqrt{m}+1/m)+\sigma_0^2 m)}{\sigma_1^2 mm_1}\right)$ for $t \in [T_1, T_2]$. Also, Definition F.1 implies that $G^{(t)}(\mu_2) - \sum_{j=1}^{m_1} w_j^{(t)} V^{(t,i)} p_{l(i,\mu)}^{(t,i)} \geq \Omega(1)$. Now, this yields

$$\frac{\partial \mu_2^\top W_K^{(t)\top} W_Q^{(t)} \mu_1}{\partial t} = \frac{1}{\sqrt{m}}\widetilde{\Theta}(\widehat{L}^{(t)}\sigma_0^2 m)\frac{1}{L} + \widetilde{O}\left(\frac{1}{\sqrt{m}}\widehat{L}^{(t)}\sigma_0^2\sqrt{m}\right).$$

On the other hand, by the analysis similar to the above, we obtain

$$\frac{\partial \mu_3^\top W_K^{(t)\top} W_Q^{(t)} \mu_1}{\partial t} = -\frac{1}{\sqrt{m}}\widetilde{\Theta}(\widehat{L}^{(t)}\sigma_0^2 m)\frac{1}{L} + \widetilde{O}\left(\frac{1}{\sqrt{m}}\widehat{L}^{(t)}\sigma_0^2\sqrt{m}\right).$$

Next, to prove the maximum change of the score, we have

$$\max_{\mu,\nu}\left|\frac{\partial \nu^\top W_K^{(t)} W_Q^{(t)} \mu}{\partial t}\right| \leq \frac{1}{\sqrt{m}}\widetilde{\Theta}(\widehat{L}^{(t)}\sigma_0^2 m)\frac{1}{L} + \widetilde{O}\left(\frac{1}{\sqrt{m}}\widehat{L}^{(t)}\sigma_0^2\sqrt{m}\right).$$

By Theorem F.8, we have

$$\left|\nu^\top W_K^{(t)\top} W_Q^{(t)} \mu - \nu^\top W_K^{(T_1)\top} W_Q^{(T_1)} \mu\right| \leq \int_{T_1}^t \frac{1}{\sqrt{m}}\widetilde{\Theta}(\widehat{L}^{(\tau)}\sigma_0^2 m)\frac{1}{L} + \widetilde{O}\left(\frac{1}{\sqrt{m}}\widehat{L}^{(\tau)}\sigma_0^2\sqrt{m}\right) d\tau$$

$$\leq \widetilde{O}\left(\frac{\sigma_0^2 m}{\sqrt{m}L\sigma_1^2 mm_1} + \frac{\sigma_0^2\sqrt{m}}{\sqrt{m}\sigma_1^2 mm_1}\right).$$

Finally, for $\gamma \in \{\mu_i\}_{i=4}^d$, $\mu \in \{\mu_i\}_{i=1}^d$, we have

$$\left|\frac{\partial}{\partial t}\mu^\top W_K^{(t)\top} W_Q^{(t)} \gamma\right| = \frac{1}{n\sqrt{m}}\widetilde{\Theta}(\widehat{L}^{(t)}\sigma_0^2 m)\frac{1}{L} + \widetilde{O}\left(\frac{1}{\sqrt{m}}\widehat{L}^{(t)}\sigma_0^2\sqrt{m}\right),$$

which implies that

$$\left|\mu^\top W_K^{(t)\top} W_Q^{(t)} \gamma - \mu^\top W_K^{(T_1)\top} W_Q^{(T_1)} \gamma\right| \leq \widetilde{O}\left(\frac{\sigma_0^2 m}{n\sqrt{m}L\sigma_1^2 mm_1} + \frac{\sigma_0^2\sqrt{m}}{\sqrt{m}\sigma_1^2 mm_1}\right).$$

$\square$

**Corollary F.17** (Change of softmax). *For $t \in [T_1, T_2]$, the softmax probability is changing in the following way:*

- *For $\mu, \nu \in \{\mu_1, \mu_2\}$, $\mu \neq \nu$, the softmax probability between the two target signals increases, whereas the softmax probability between one target signal and the common token decreases, i.e.,*

$$\frac{\partial}{\partial t}p_{q\leftarrow\mu,k\leftarrow\nu}^{(t,i)} = \frac{1}{\sqrt{m}}\widetilde{\Theta}(\widehat{L}^{(t)}\sigma_0^2 m)\frac{1}{L^2},$$

$$\frac{\partial}{\partial t}p_{q\leftarrow\mu,k\leftarrow\mu_3}^{(t,i)} = -\frac{1}{\sqrt{m}}\widetilde{\Theta}(\widehat{L}^{(t)}\sigma_0^2 m)\frac{1}{L^2}.$$

- *For all $\mu \in \{\mu_1, \mu_2\}$, $\gamma \in \{\mu_i\}_{i=4}^d$, the softmax probability between one target signal and a random token changes as follows:*

$$\left|\frac{\partial}{\partial t}p_{q\leftarrow\mu,k\leftarrow\gamma}^{(t,i)}\right| \leq \frac{1}{n\sqrt{m}}\widetilde{\Theta}(\widehat{L}^{(t)}\sigma_0^2 m)\frac{1}{L^2} + \widetilde{O}\left(\frac{1}{L\sqrt{m}}\widehat{L}^{(t)}\sigma_0^2\sqrt{m}\right).$$

*Furthermore, we have*

$$\max_{i\in[n],l_1,l_2\in[L]}\left|p_{l_1,l_2}^{(t,i)} - p_{l_1,l_2}^{(0,i)}\right| \leq O(1/L^2).$$

*Proof.* Take $\mu = \mu_1$, $\nu = \mu_2$. First of all, by Definition F.1 and Lemma F.16, we have

$$\left| p_{q\leftarrow\mu_1}^{(t,i)\top} \frac{\partial X^{(i)\top} W_K^{(t)\top} W_Q^{(t)} \mu_1}{\partial t} \right| \leq \frac{1}{\sqrt{m}} O(\widehat{L}^{(t)}\sigma_0^2 m)\frac{1}{L^2} + \frac{1}{n\sqrt{m}}\widetilde{\Theta}(\widehat{L}^{(t)}\sigma_0^2 m)\frac{1}{L} + \widetilde{O}\left(\frac{1}{\sqrt{m}}\widehat{L}^{(t)}\sigma_0^2\sqrt{m}\right).$$

Thus, by Lemma G.1 and Lemma F.16, for $i \in I_1$, we obtain

$$\frac{\partial}{\partial t} p_{q\leftarrow\mu_1,k\leftarrow\mu_2}^{(t,i)} = \frac{1}{m}\widetilde{\Theta}(\widehat{L}^{(t)}\sigma_0^2 m)\frac{1}{L^2},$$

$$\frac{\partial}{\partial t} p_{q\leftarrow\mu_1,k\leftarrow\mu_3}^{(t,i)} = -\frac{1}{m}\widetilde{\Theta}(\widehat{L}^{(t)}\sigma_0^2 m)\frac{1}{L^2}.$$

On the other hand, for $\gamma \in \{\mu_i\}_{i=4}^d$, we have

$$\left| \frac{\partial}{\partial t} p_{q\leftarrow\mu,k\leftarrow\gamma}^{(t,i)} \right| \leq \frac{1}{nm}\widetilde{\Theta}(\widehat{L}^{(t)}\sigma_0^2 m)\frac{1}{L^2} + \widetilde{O}\left(\frac{1}{Lm}\widehat{L}^{(t)}\sigma_0^2\sqrt{m}\right).$$

Finally, by Lemma F.16, we have

$$\max_{\mu,\nu}\left| \nu^\top W_K^{(t)} W_Q^{(t)}\mu - \nu^\top W_K^{(T_1)} W_Q^{(T_1)}\mu \right| \leq \widetilde{O}\left(\frac{\sigma_0^2 m}{\sqrt{m}L\sigma_1^2 m m_1} + \frac{\sigma_0^2\sqrt{m}}{\sqrt{m}\sigma_1^2 m m_1}\right).$$

Thus, by Lemma G.2, we obtain

$$\max_{i,l_1,l_2}\left| p_{l_1,l_2}^{(t,i)} - p_{l_1,l_2}^{(T_1,i)} \right|$$

$$\leq \widetilde{O}\left(\frac{\sigma_0^2 m}{mL^2\sigma_1^2 m m_1} + \frac{\sigma_0^2\sqrt{m}}{m\sigma_1^2 m m_1 L} + L\left(\frac{\sigma_0^2 m}{mL\sigma_1^2 m m_1} + \frac{\sigma_0^2\sqrt{m}}{m\sigma_1^2 m m_1}\right)^2\right) = O(1/L^2).$$

$\square$

**Corollary F.18.** *For $t \in [T_1, T_2]$, we have*

$$\frac{\max_{i\in[n]}\left| \sum_{l_1=1}^L \sum_{l_2=1}^L G^{(t)}(X_{l_2}^{(i)})\frac{\partial p_{l_1,l_2}^{(t,i)}}{\partial t} \right|}{\sum_{j_1=1}^{m_1}\sum_{j_2=1}^{m_1}\left\langle a_{j_1} w_{j_1}^{(t)}, a_{j_2} w_{j_2}^{(t)} \right\rangle} \leq o(1)\frac{1}{n}\sum_{i\in[n]} g_i^{(t)}.$$

*Proof.* This is a direct consequence of Definition F.1, Lemma F.10 and Corollary F.17. $\square$

### F.7 Change of Self-Correlation of Key/Query-Transformed data

**Lemma F.19** (Change of self-correlation). *For $\mu, \nu \in \{\mu_i\}_{i=1}^d$, we have*

$$\left| \nu^\top W_Q^{(t)\top} W_Q^{(t)}\mu - \nu^\top W_Q^{(T_1)\top} W_Q^{(T_1)}\mu \right| \leq \widetilde{O}\left(\frac{\sigma_0^2\sqrt{m}}{\sqrt{m}\sigma_1^2 m m_1}\right),$$

$$\left| \nu^\top W_K^{(t)\top} W_K^{(t)}\mu - \nu^\top W_K^{(T_1)\top} W_K^{(T_1)}\mu \right| \leq \widetilde{O}\left(\frac{\sigma_0^2\sqrt{m}}{\sqrt{m}\sigma_1^2 m m_1}\right).$$

*Proof.* We prove the result for $\nu^\top W_Q^{(t)\top} W_Q^{(t)}\mu$ and the proof for $\nu^\top W_K^{(t)\top} W_K^{(t)}\mu$ is similar. By Lemma C.4, we have

$$\frac{\partial \nu^\top W_Q^{(t)\top} W_Q^{(t)}\mu}{\partial t}$$

$$= \frac{1}{n\sqrt{m}}\sum_{i:\mu\in X^{(i)}} g_i^{(t)} y_i \sum_{j=1}^{m_1} a_j \nu^\top W_Q^{(t)\top} K^{(t,i)}\text{diag}\left(V^{(t,i)\top} w_j^{(t)} - w_j^{(t)\top} V^{(t,i)} p_{l(i,\mu)}^{(t,i)}\right) p_{l(i,\mu)}^{(t,i)}$$

$$+ \frac{1}{n\sqrt{m}}\sum_{i:\nu\in X^{(i)}} g_i^{(t)} y_i \sum_{j=1}^{m_1} a_j \mu^\top W_Q^{(t)\top} K^{(t,i)}\text{diag}\left(V^{(t,i)\top} w_j^{(t)} - w_j^{(t)\top} V^{(t,i)} p_{l(i,\nu)}^{(t,i)}\right) p_{l(i,\nu)}^{(t,i)}.$$

A simple result from Lemma F.16 is that $|\nu^\top W_K^{(t)\top} W_Q^{(t)} \mu| \leq \widetilde{O}(\sigma_0^2 \sqrt{m})$ for $t \in [T_1, T_2]$ and $\mu, \nu \in \{\mu_i\}_{i=1}^d$. Therefore, by Definition F.1,

$$\left| \frac{\partial \nu^\top W_Q^{(t)\top} W_Q^{(t)} \mu}{\partial t} \right| \leq \widetilde{O}\left( \frac{1}{\sqrt{m}} \widehat{L}^{(t)} \sigma_0^2 \sqrt{m} \log m \right),$$

which implies

$$\left| \nu^\top W_Q^{(t)\top} W_Q^{(t)} \mu - \nu^\top W_Q^{(T_1)\top} W_Q^{(T_1)} \mu \right| \leq \int_{T_1}^t \widetilde{O}\left( \frac{1}{\sqrt{m}} \widehat{L}^{(t)} \sigma_0^2 \sqrt{m} \log m \right) d\tau \leq \widetilde{O}\left( \frac{\sigma_0^2 \sqrt{m}}{\sqrt{m}\sigma_1^2 mm_1} \right),$$

where the inequality follows from Theorem F.8 and Definition F.1. $\qquad \square$

## F.8 Small Loss is Achieved

**Theorem F.20.** *Define* $T^\star = \min_t \{t : \widehat{L}^{(t)} = \Theta(1/\mathrm{poly}(m))\}$. *Then,* $T^\star \in [T_1, T_2]$.

*Proof.* The following results altogether show that Phase 2 can last for at least $\Theta(\mathrm{poly}(m))$ time:

- Lemma F.19 proves that the change of $K, Q$ self-correlation as follows:

$$\max_{\mu,\nu} \left| \mu^\top W_K^{(t)\top} W_K^{(t)} \nu - \mu^\top W_K^{(0)\top} W_K^{(0)} \nu \right| = \widetilde{O}(\sigma_0^2 \sqrt{m}),$$

$$\max_{\mu,\nu} \left| \mu^\top W_Q^{(t)\top} W_Q^{(t)} \nu - \mu^\top W_Q^{(0)\top} W_Q^{(0)} \nu \right| = \widetilde{O}(\sigma_0^2 \sqrt{m}).$$

- Corollary F.9 proves that

$$\sum_{j_1=1}^{m_1} \sum_{j_2=1}^{m_1} \left\langle a_{j_1} w_{j_1}^{(t)}, a_{j_2} w_{j_2}^{(t)} \right\rangle \leq O(\sigma_1^2 mm_1).$$

- Corollary F.12 proves that

$$\frac{\max_{\mu \in \{\mu_i\}_{i=1}^d} \left| \sum_{j=1}^{m_1} a_j \frac{\partial w_j^{(t)}}{\partial t} W_V^{(t)} \mu \right|}{\sum_{j_1=1}^{m_1} \sum_{j_2=1}^{m_1} \left\langle a_{j_1} w_{j_1}^{(t)}, a_{j_2} w_{j_2}^{(t)} \right\rangle} \leq o(1/L) \frac{1}{n} \sum_{i \in [n]} g_i^{(t)}.$$

- Corollary F.15 proves that

$$\left| \sum_{l_1=1}^L \sum_{l_2 : X_{l_2}^{(i)} \in \{\mu_k\}_{k=4}^d} G^{(t)}(X_{l_2}^{(i)}) p_{q \leftarrow l_1, k \leftarrow l_2}^{(t,i)} \right| \leq O(1).$$

- Corollary F.17 proves that

$$\max_{i \in [n], l_1, l_2 \in [L]} \left| p_{l_1, l_2}^{(t,i)} - p_{l_1, l_2}^{(0,i)} \right| \leq O(1/L^2).$$

- Corollary F.18 proves that

$$\frac{\max_{i \in [n]} \left| \sum_{l_1=1}^L \sum_{l_2=1}^L G^{(t)}(X_{l_2}^{(i)}) \frac{\partial p_{l_1, l_2}^{(t,i)}}{\partial t} \right|}{\sum_{j_1=1}^{m_1} \sum_{j_2=1}^{m_1} \left\langle a_{j_1} w_{j_1}^{(t)}, a_{j_2} w_{j_2}^{(t)} \right\rangle} \leq o(1) \frac{1}{n} \sum_{i \in [n]} g_i^{(t)}.$$

- Lemma F.5 implies that

$$\frac{1}{2} \leq \frac{\sum_{i \in I_2} g_i^{(t)}}{\sum_{i \in I_3} g_i^{(t)}} \leq 2.$$

- Lemma F.6 implies that the gradients $\sum_{i \in I} g_i^{(t)}$ for $I \in \{I_1, I_2, I_3, I_4\}$ satisfies

$$\sum_{i \in I_4} g_i \le \min\left(\sum_{i \in I_2} g_i^{(t)}, \sum_{i \in I_3} g_i^{(t)}\right) \le \max\left(\sum_{i \in I_2} g_i^{(t)}, \sum_{i \in I_3} g_i^{(t)}\right) \le \sum_{i \in I_1} g_i^{(t)} \le \sum_{i \in I_2 \cup I_3 \cup I_4} g_i^{(t)}.$$

- Lemma F.6 and Corollary F.17 proves that $y_i F_i^{(t)} \ge y_i F_i^{(T_1)} \ge C$.

The above shows that all the requirements needed to satisfy the definition of Phase 2 (Definition F.1) can hold for at least $\Omega(\text{poly}(m))$ time. Thus, $T_2 - T_1 \ge \Omega(\text{poly}(m))$. Further, by Theorem F.8, we have that $\widehat{L}^{(t)} \le O(1/\text{poly}(m))$ implies $t = \Theta(\text{poly}(m))$. $\qquad \square$

### F.9 Proof of Theorem 3.2

*Proof.* This is proved in Corollary F.13 and Lemma F.16. $\qquad \square$

### F.10 Proof of Theorem 3.3

*Proof.* This is proved as a direct consequence of Lemma F.6 and Theorem F.20.

For the generalization loss, since the training loss satisfies $\widehat{L}^{(T^\star)} \le 1/\text{poly}(m)$, for each class $I \in \{I_1, I_2, I_3, I_4\}$ there exists a sample $X_I^\star$ such that $\ell(X_I^\star) \le 1/\text{poly}(m)$. Note that by Definition F.1 the random tokens only contributes to $O(1)$ in $F^{(T^\star)}(X)$. Thus, given a fixed new sample $X \sim \mathcal{D}$, we have $|F^{(T^\star)}(X) - F^{(T^\star)}(X_I^\star)| \le O(1)$ which implies $\ell(y F^{(T^\star)}(X)) \le 1/\text{poly}(m)$. Since this holds for all $X \sim \mathcal{D}$, we have $L^{(T^\star)} \le 1/\text{poly}(m)$. $\qquad \square$

## G Auxiliary Results

The gradient of $p(x)_i = \text{Softmax}(x)_i$ for $x \in \mathbb{R}^n$:

$$\frac{\partial p(x)_i}{\partial x_j} = \frac{\partial}{\partial x_j} \frac{\exp(x_i)}{\sum_{k=1}^n \exp(x_k)} = \begin{cases} -\frac{\exp(x_i)}{\sum_k \exp(x_k)} \frac{\exp(x_j)}{\sum_k \exp(x_k)} = -p(x)_i p(x)_j & i \ne j \\ \frac{\exp(x_i)}{\sum_k \exp(x_k)} - \left(\frac{\exp(x_i)}{\sum_k \exp(x_k)}\right)^2 = p(x)_i(1 - p(x)_i) & i = j \end{cases}$$

$$= p(x)_i(\mathbb{I}(i = j) - p(x)_j)$$

$$\Rightarrow J(p(x)) = \text{diag}(p(x)) - p(x)p(x)^\top. \tag{17}$$

**Lemma G.1** (Gradient of softmax). *Let $s(t) \in \mathbb{R}^l$ be differentiable in $t$ and $p(s) = \text{Softmax}(s)$. Denote $p_i(s) = \text{Softmax}(s)_i$. Then*

$$\frac{\partial p(s(t))}{\partial t} = \frac{\partial p(s)}{\partial s} \frac{\partial s(t)}{\partial t} = (\text{diag}(p(s)) - p(s)p(s)^\top) \frac{\partial s(t)}{\partial t}.$$

*Proof.* By the chain rule and the gradient of softmax in Equation (17), we obtain

$$\frac{\partial p(s(t))}{\partial t} = \frac{\partial p(s)}{\partial s} \frac{\partial s(t)}{\partial t} = J(p(s)) \frac{\partial s(t)}{\partial t} = (\text{diag}(p(s)) - p(s)p(s)^\top) \frac{\partial s(t)}{\partial t}.$$

$\qquad \square$

**Lemma G.2** (Perturbation of softmax). *Let $s \in \mathbb{R}^l$ and $p(s) = \text{Softmax}(s)$. Denote $p_i(s) = \text{Softmax}(s)_i$. Consider a small perturbation $\varepsilon \in \mathbb{R}^l$ to $s$. Then*

$$p(s + \varepsilon) - p(s) = (\text{diag}(p(s)) - p(s)p(s)^\top)\varepsilon + \xi,$$

*where $\|\xi\|_\infty = O(\|\varepsilon\|_2^2)$.*

*Proof.* By Taylor's expansion theorem on softmax and the gradient of softmax in Equation (17), we have

$$p(s + \varepsilon) - p(s) = J(p(s))\varepsilon + \xi = (\text{diag}(p(s)) - p(s)p(s)^\top)\varepsilon + \xi,$$

where $\|\xi\|_\infty = O(\|\varepsilon\|_2^2)$. $\qquad \square$

# H Probability

**Lemma H.1** (Bernstein's inequality for bounded random variables). *Assume $Z_1, \ldots, Z_n$ are $n$ i.i.d. random variables with $\mathbb{E}[Z_i] = 0$ and $|Z_i| \leq M$ for all $i \in [n]$ almost surely. Let $Z = \sum_{i=1}^n Z_i$. Then, for all $t > 0$,*

$$\mathbb{P}[Z > t] \leq \exp\left(-\frac{t^2/2}{\sum_{j=1}^n \mathbb{E}[Z_j^2] + Mt/3}\right) \leq \exp\left(-\min\left\{\frac{t^2}{2\sum_{j=1}^n \mathbb{E}[Z_j^2]}, \frac{t}{2M}\right\}\right),$$

*which implies with probability at least $1 - \delta$,*

$$Z \leq \sqrt{2\sum_{j=1}^n \mathbb{E}[Z_j^2] \log\frac{1}{\delta}} + 2M \log\frac{1}{\delta}.$$

**Lemma H.2.** *For $w_1, w_2 \in \mathbb{R}^m$ with $w_1, w_2 \overset{i.i.d.}{\sim} \mathcal{N}(0, I_m/m)$, we have*

$$\mathbb{P}\left[\left|\|w_1\|_2^2 - 1\right| \geq \sqrt{\frac{4}{m}\log\frac{2}{\delta}} + \frac{4}{m}\log\frac{2}{\delta}\right] \leq \delta,$$

$$\mathbb{P}\left[|\langle w_1, w_2\rangle| \geq \sqrt{\frac{4}{m}\log\frac{2}{\delta}} + \frac{4}{m}\log\frac{2}{\delta}\right] \leq \delta.$$

*Proof.* We first have

$$\mathbb{E}\left[\|w_1\|_2^2\right] = \mathbb{E}\left[\sum_{i=1}^m w_{1,i}^2\right] = 1.$$

Note that $w_{1,i}^2$ is a sub-Gamma random variable with parameters $(\frac{4}{m^2}, \frac{4}{m})$. Thus, by Bernstein's inequality,

$$\mathbb{P}\left[\left|\|w_1\|_2^2 - \mathbb{E}\left[\|w_1\|_2^2\right]\right| \geq \sqrt{\frac{4}{m}\log\frac{2}{\delta}} + \frac{4}{m}\log\frac{2}{\delta}\right] \leq \delta.$$

Next,

$$\mathbb{E}[\langle w_1, w_2\rangle] = \mathbb{E}\left[\sum_{i=1}^m w_{1,i}w_{2,i}\right] = 0$$

By Bernstein's inequality, we obtain

$$\mathbb{P}\left[|\langle w_1, w_2\rangle| \geq \sqrt{\frac{4}{m}\log\frac{2}{\delta}} + \frac{4}{m}\log\frac{2}{\delta}\right] \leq \delta.$$

$\square$

