# OpenReview forum: "Training Dynamics of Transformers to Recognize Word Co-occurrence via Gradient Flow Analysis"
_NeurIPS.cc/2024/Conference — NeurIPS 2024 poster_

### Official Review · Reviewer_cFaT · 2024-07-11

**Soundness:** 3
**Presentation:** 3
**Contribution:** 3
**Rating:** 6
**Confidence:** 3

**Summary:**

This paper investigates the training dynamics of a single-layer transformer followed by a single MLP layer on a synthetic binary classification task, where the objective is to identify the co-occurrence of two specific tokens in the input sequence. They analyze the gradient flow dynamics for the case that all the attention parameters (key, query, value) and the linear layer all trainable and show that the model can achieve low loss despite the non-convexity of the objective. They identify two phases in the training, 1) the MLP aligns with the two target tokens at the start of the training and the model learns to classify all the samples correctly, 2) the attention and MLP parameters update to increase the classification margin and drive the loss to zero. They also run a small scale numerical experiment in their synthetic setup tp confirm their analysis.

**Strengths:**

The paper makes no restricting assumptions on the weights of the transformer model and performs the analysis on the joint optimization of all the parameters.

Although the paper and its proof are notation-heavy, the authors have broken down the complexity of the proof and notation in the main body to clarify the steps needed to prove the results.

**Weaknesses:**

There are some restrictive assumptions on the synthetic data model: The vocabulary set $d$ is considered to be larger than the number of training tokens, which is not the case in realistic setups. Thus, some tokens are not visited at training time. Also, they assume, apart from the two target tokens, the remaining tokens appear at maximum once in the training set.

The proof outline in the main body helps in understanding the high-level steps involved. However, it could still benefit from additional clarifications on some intermediate steps. For instance, in phases 1 and 2, it's mentioned how the alignment of the MLP with the target tokens $G^{(t)}(\mu_{1,2})$ behaves during training. However, it's not clear how this connects to the evolution of the attention scores in phase 2 as stated in Lemma 4.7.

**Questions:**

1. As far as I understand, in your synthetic task,  in the first phase of training, effectively only the MLP weights are learning the task. That is, the model can achieve 100% accuracy only by aligning the MLP weights with the relevant tokens $\mu_1,\mu_2$. So, the attention layer is not needed for identifying the co-occurrence of the tokens in this setup?

2. Can you also report validation and accuracy plots in your synthetic experiments? Does the validation loss decay at the same rate as the training loss as stated in Thm 3.3?

3. Regarding the proof sketch:

    a) The alignment of parameters with the target tokens is discussed in the main body. Can you also clarify how the gradients related to irrelevant tokens evolve? In particular, regarding the tokens that do not appear in the training set (since $nL\leq d$), does the model learn not to attend to them at test time?

    b) I find it confusing that the softmax output remains close to $1/L$ long in the training (line 320) and assigns uniform attention to all tokens in the sequence. Does this statement hold for all training samples? If yes, then how does the model learn to attend to the target tokens?

**Limitations:**

Yes

---

> ### Author Rebuttal · Authors · 2024-08-06
>
> We thank the reviewer very much for your time and efforts on providing helpful review comments.
>
> **Comment:**
> There are some restrictive assumptions on the synthetic data model: The vocabulary set $d$ is considered to be larger than the number of training tokens, which is not the case in realistic setups. Thus, some tokens are not visited at training time. Also, they assume, apart from the two target tokens, the remaining tokens appear at maximum once in the training set.
>
> **Response:** Thanks for the question. In fact, we can relax both assumptions by letting the remaining tokens be uniformly randomly sampled from the vocabulary. Our proof will still be valid. More specifically, our proof holds as long as the number of each individual random token is much less than the number of the signals. For example, if we uniformly sample the irrelevant tokens from the vocabulary, then in expectation each irrelevant token will only appear $\frac{nL}{d}$ times in the training set which is much less than $n$ (the number of times each signal appears in the training set) if $d \gg L$.
>
> **Comment:**
> In phases 1 and 2, it's mentioned how the alignment of the MLP with the target tokens $G^{(t)}(\mu_{1,2})$ behaves during training. However, it's not clear how this connects to the evolution of the attention scores in phase 2 as stated in Lemma 4.7.
>
> **Response:**
> This can be seen from the update of the score in the gradient flow dynamical system which is in Appendix C. Take the score between $\mu_1$ and $\mu_2$ for example, rearranging the terms, we have
> \begin{align*}
>     &\frac{\partial \mu_1^\top W_K^{(t) \top} W_Q^{(t)} \mu_2}{\partial t} \\\\
>     &= \frac{1}{n\sqrt{m}} \sum_{i=1}^n g_i^{(t)} y_i \sum_{l=1}^L \mu_1^\top W_K^{(t) \top} K^{(t,i)} \cdot \textnormal{diag}\left( G^{(t)}(X^{(i)}) - (G^{(t)}(X^{(i)}))^\top p_l^{(t,i)} \right) p_l^{(t,i)} x_l^{(i) \top} \mu_2 \\\\
>     &+ \frac{1}{n\sqrt{m}} \sum_{i=1}^n g_i^{(t)} y_i \sum_{l=1}^L \mu_2^\top W_Q^{(t) \top} q_l^{(t,i)} p_l^{(t,i) \top} \cdot \textnormal{diag}\left( G^{(t)}(X^{(i)}) - (G^{(t)}(X^{(i)}))^\top p_l^{(t,i)} \right) X^{(i) \top} \mu_1
> \end{align*}
> Since $G^{(t)}(\mu_1) = \Theta(1)$ after phase 1, we have $G^{(t)}(\mu_1) - (G^{(t)}(X^{(i)}))^\top p_l^{(t,i)} = \Theta(1)$. We can use this to show that $\frac{\partial \mu_1^\top W_K^{(t) \top} W_Q^{(t)} \mu_2}{\partial t}$ is positive in Lemma 4.7 and thus the score between $\mu_1$ and $\mu_2$ is increasing.
>
> We will add these clarifications in our proof outline. We will also try to enrich our proof outline by adding clarifications on some intermediate steps.
>
> **Comment:**
> As far as I understand, in your synthetic task, in the first phase of training, effectively only the MLP weights are learning the task. That is, the model can achieve 100\% accuracy only by aligning the MLP weights with the relevant tokens $\mu_1,\mu_2$. So, the attention layer is not needed for identifying the co-occurrence of the tokens in this setup?
>
> **Response:** Although MLP can achieve full accuracy, only MLP layer in phase 1 does not achieve a good performance margin, i.e., the loss is not vanishing yet, and such a model can easily make wrong classification when data have noise. The important role that attention layer plays in phase 2 is to enlarge the classification margin together with MLP so that the loss approaches zero and the trained model can generalize well.
>
> **Comment:**
> Can you also report validation and accuracy plots in your synthetic experiments? Does the validation loss decay at the same rate as the training loss as stated in Thm 3.3?
>
> **Response:** In the attached pdf file, Fig. 2(a) provides the accuracy plot, where both the training and test errors become zero after certain number of training steps. Fig. 2(b) provides the validation (i.e., test) and training losses, and they decay at similar rate.
>
> **Comment:**
> The alignment of parameters with the target tokens is discussed in the main body. Can you also clarify how the gradients related to irrelevant tokens evolve? In particular, regarding the tokens that do not appear in the training set (since $nL\leq d$), does the model learn not to attend to them at test time?
>
> **Response:** The alignment of the random tokens is much smaller than the signals, and remain to be small and bounded throughout the training process since they don't occur as much in the training set.
> In our proof, we handle the random tokens all together no matter if they occur in the training set or not.
> Their effect can be upper bounded during training (Theorem E.15 for phase 1 and Appendix F.5 for phase 2).
> So the answer to your question is yes -- the model does learn not to attend to those tokens not in the training set at test time.
>
> **Comment:**
> I find it confusing that the softmax output remains close to $1/L$ long in the training (line 320) and assigns uniform attention to all tokens in the sequence. Does this statement hold for all training samples? If yes, then how does the model learn to attend to the target tokens?
>
> **Response:**
> Sorry about the confusion. In line 320, we mean to explain how we prove convergence. In fact, there are 2 steps to show convergence: (1) show that the training loss can decrease when the softmax outputs remain to $1/L$, which is what we explain in the proof outline; (2) show that with the deviation in the softmax output from $1/L$, the loss value will still decrease, which is not explained in the proof outline (we will add this part to the proof outline to avoid confusion). The outputs of the softmax attention indeed change as stated in Lemma 4.7.
>
> We thank the reviewer again for your comments. We hope that our responses resolved your concerns. If so, we wonder if the reviewer could kindly increase your score. Certainly, we are more than happy to answer your further questions.

---

> > ### Author Response · Authors · 2024-08-10
> > **A gentle reminder**
> >
> > Dear Reviewer cFaT,
> >
> > We've taken your initial feedback into careful consideration in our response. Could you please check whether our responses have properly addressed your concerns? If so, could you please kindly consider increasing your initial score accordingly? Certainly, we are more than happy to answer your further questions.
> >
> > Thank you for your time and effort in reviewing our work!
> >
> > Best Regards,
> > Authors

---

> > > ### Comment · Reviewer_cFaT · 2024-08-12
> > >
> > > Thanks for your response.
> > >
> > > Given that the data setup is simple, the findings on the efficient learnability of this task are not particularly surprising, and might not provide much insight on practical tasks.
> > >
> > > However, the minimal assumptions on the model weights and the joint optimization analysis of all transformer parameters could make the techniques valuable for future transformer analyses. Thus, I will increase my score to weak accept. Since the main contribution of the paper lies in the proof technique, I suggest enhancing the presentation of the proof outline in the final manuscript.

---

> > > > ### Author Response · Authors · 2024-08-12
> > > > **Thank you**
> > > >
> > > > We thank the reviewer very much for your further feedback. We will follow the reviewer's suggestion and enhance the presentation of the proof outline in the revision. We also highly appreciate that you increased the rating of the paper.

---

### Official Review · Reviewer_ritc · 2024-07-12

**Soundness:** 3
**Presentation:** 3
**Contribution:** 3
**Rating:** 6
**Confidence:** 3

**Summary:**

This paper studies the training dynamics of a single hidden layer transformer network (self-attention + linear MLP) trained on a binary word cooccurrence task. Specifically, given a data matrix $X \in R^{d \times L}$ representing L "words" (each column of X is a word vector of dimension d), the model must output +1 if words 1 and 2 both occur in X, and -1 otherwise. The paper shows that a transformer layer is able to learn this task, and that the training occurs in two stages: First, the linear MLP layer learns to classify data points correctly by positively aligning with the embeddings for words 1 and 2 (but without making large changes to attention matrices). Second, it drives the loss down further by using the attention matrices to positively correlate q,k,v for words 1 and 2, and anti-correlate the q,k,v for a common word (denoted word "3" in the paper) relative to words 1,2.  After these phases, both the training and generalization losses go to zero (as long as embedding dimension is large enough).

Overall, I found the results interesting and insightful, though not very surprising, and the practical implications of these results were not very clear to me. Thus, I currently recommend weak accept. Importantly, my primary research area is not learning theory, so my knowledge of the related work is relatively limited, and thus my review confidence is relatively low.

**Strengths:**

- It is interesting to see that the training dynamics for this word cooccurrence task can be analyzed rigorously, with relatively few assumptions.
- The theoretical results are validated with a nice synthetic experiments, that demonstrates that the two phases predicted by the theory do occur in practice.

**Weaknesses:**

- This word cooccurrence task is very simple, and thus it is not surprising that a single transformer layer can easily learn it.
- Only full gradients are considered, whereas transformers are typically trained with mini-batch Adam(W).

**Questions:**

- What other tasks (beyond word cooccurrence) do you think could be analyzed with this methodology?
- What are the implications of this result to more complex/realistic tasks, like next token prediction?
- If mini-batch Adam is used during training, do the two phases still occur?
- Can you add more details about the experimental setup to the main paper?
- Can you add more discussion about the automatic balancing of gradients, and its significance, in a more central part of the text (e.g., section 3, not 4)?

**Limitations:**

Yes, limitations have been discussed.

---

> ### Author Rebuttal · Authors · 2024-08-06
>
> We thank the reviewer very much for your time and efforts on providing helpful review comments.
>
> **Comment:**
> This word co-occurrence task is very simple, and thus it is not surprising that a single transformer layer can easily learn it.
>
> **Response:**
> We agree that this is a simple task from a representational perspective.
> However, our primary goal here is to use this setting as an **analytically tractable** case and develop theoretical techniques to understand the *training dynamics* of transformers, which is a highly non-trivial task considering the complicated nature of self-attention and transformers.
>
> **Comment:**
> Only full gradients are considered, whereas transformers are typically trained with mini-batch Adam(W).
>
> **Response:**
> Regarding full gradients, since we are using gradient flow, it is more convenient to analyze the full gradients. We can indeed use gradient descent and relax the full gradients to consider stochastic gradients. However, this will create some unnecessary hassle and won't change the essential nature of two-phase training dynamics of transformers.
>
> Regarding Adam or its variant, it is indeed an interesting direction. However, its analysis would be more complicated due to the adaptive momentum terms in the iterative update. The solution that transformers converge to can also have different implicit bias from SGD. Thus, we will leave this as future work.
>
> **Comment:**
> What other tasks (beyond word co-occurrence) do you think could be analyzed with this methodology?
>
> **Response:**
> Thank you for asking.
> Our proof techniques open up for a great possibility of settings to analyze such as NLP tasks with one hot embedding. For example, one popular way of modeling natural language is by modeling the language sequences by Markov chains. Our techniques can be used to study the training dynamics when we want the transformer model to predict the next input given the previous tokens in the sequence.
>
> **Comment:**
> What are the implications of this result to more complex/realistic tasks, like next token prediction?
>
> **Response:** Our work could have some implications applicable to more complex tasks including next-token prediction. (i) The training can experience two or more training phases, where each phase captures one salient change of certain attention scores or MLP weights. (ii) The gradient of well-designed loss function can facilitate classification and enlarge the margin by driving attention scores between query and relevant key tokens to change properly during the training process.
>
> **Comment:**
> If mini-batch Adam is used during training, do the two phases still occur?
>
> **Response:**
> We expect the training phases of Adam will exhibit some differences from gradient descent considered in our work. From a generic nonconvex optimization perspective, (Shuo \& Li) showed that Adam can be viewed as normalized gradient descent in $\ell_\infty$ geometry whereas gradient descent operates on $\ell_2$ geometry.
> This is due to the coordinate-wise normalization operation uniquely occurring in Adam but not in GD. We expect that the loss with transformers trained by Adam will somewhat also follow such an observation.
>
> Xie, Shuo, and Zhiyuan Li. "Implicit Bias of AdamW: $\ell_\infty $-Norm Constrained Optimization." Forty-first International Conference on Machine Learning.
>
> **Comment:**
> Can you add more details about the experimental setup to the main paper?
>
> **Response:** Thanks for the suggestion. We will make more space in the main paper and add the experimental setup.
>
> **Comment:**
> Can you add more discussion about the automatic balancing of gradients, and its significance, in a more central part of the text (e.g., section 3, not 4)?
>
> **Response:**
> Thank you for the suggestion. We will make the change as the reviewer suggested.
>
> We thank the reviewer again for your comments. We hope that our responses resolved your concerns. If so, we wonder if the reviewer could kindly consider to increase your score. Certainly, we are more than happy to answer your further questions.

---

> > ### Author Response · Authors · 2024-08-10
> > **A gentle reminder**
> >
> > Dear Reviewer ritc,
> >
> > We've taken your initial feedback into careful consideration in our response. Could you please check whether our responses have properly addressed your concerns? If so, could you please kindly consider increasing your initial score accordingly? Certainly, we are more than happy to answer your further questions.
> >
> > Thank you for your time and effort in reviewing our work!
> >
> > Best Regards,
> > Authors

---

> > ### Comment · Reviewer_ritc · 2024-08-13
> >
> > Thank you very much for your response. I keep my score unchanged, due to what I perceive to be the limited impact/scope of the work.
> >
> > Regarding my question: "If mini-batch Adam is used during training, do the two phases still occur?" ---> Could you add an experiment to check this?

---

> > > ### Author Response · Authors · 2024-08-14
> > >
> > > Thank you very much for your feedback.
> > >
> > > As we mentioned in our rebuttal, we expect the training phases of Adam will exhibit some differences from gradient descent considered in our work.
> > > This is due to the coordinate-wise normalization operation uniquely occurring in Adam but not in GD.
> > > Our experiments confirmed our thoughts.
> > > We provide the experiment results on AdamW with default parameter setup below, where we show the changes in $G(\mu_1)$, which is the MLP alignment with the signal $\mu_1$, and the attention score when the query is $\mu_1$ and the key is $\mu_2$.
> > > All the values are rounded to a tenth of decimal.
> > > As you can see from the experiment results below, both the MLP and score change dramatically in the first 50 steps and then the changes slow down.
> > > The behavior thus is very different from gradient descent, where only MLP changes rapidly in the initial training and then both MLP and score jointly change substantially.
> > >
> > > | Step | 0 | 50 | 100 | 150 | 200 | 250 |
> > > | --- | --- | --- | --- | --- | --- | --- |
> > > | $G(\mu_1)$ | 0.2 | 50.8 | 56.8 | 59.7 | 62.0 | 64.0 |
> > > | Score $(\times 10^{-1})$ | -0.8 | 3.0 | 3.5 | 3.8 | 4.1 | 4.3 |
> > >
> > > We further emphasize that the main contribution of this paper lies in the development of the **new techniques** for analyzing the training dynamics of transformers, especially the joint optimization analysis of all transformer parameters, as Reviewer cFaT pointed out. We expect that these mathematical techniques can be generalized to more complicated transformer architectures in the future.
> > >
> > > We thank the reviewer for the time and efforts.
> > > We hope that our response answers your question.

---

### Official Review · Reviewer_G3j5 · 2024-07-13

**Soundness:** 3
**Presentation:** 2
**Contribution:** 3
**Rating:** 5
**Confidence:** 4

**Summary:**

This article delves into the gradient flow dynamics for detecting word co-occurrence, demonstrating that the gradient flow approach can achieve minimal loss. The training process commences with random initialization and can be delineated into two distinct phases.

**Strengths:**

- This article noticed an interesting phase transition during training in this special setting and demonstrates it with solid calculation and experiments.
- A new property of gradient flow is noticed and contributes to prove near minimum training loss together with the analysis of softmax.

**Weaknesses:**

The setting of empirical experiments is also simple and ideal and readers may have no idea if this is a general phenomenon during training for detecting word co-occurrence.

**Questions:**

- In line 151 and 152, it is confusing why concentration theorems lead to the specific probability in (i) of Assumption 2.3.
- Lack of explanation for $\langle w_{j_1}^{(t)},w_{j_2}^{(t)} \rangle$ in line 194.
- It is not obvious why "the samples with only one target signal may be classified incorrectly as co-occurence" in line 282.
- The notation in line 169 is somewhat misleading.

**Limitations:**

The semantic mechanism this article focuses on is not general enough in NLP and the the results cannot give instruction useful enough to guide the training process of Transformer.

---

> ### Author Rebuttal · Authors · 2024-08-06
>
> We thank the reviewer very much for your time and efforts on providing helpful review comments.
>
> **Comment:**
> The setting of empirical experiments is also simple and ideal and readers may have no idea if this is a general phenomenon during training for detecting word co-occurrence.
>
> **Response:** We clarify that the experiment presented in the paper is used to guide and validate our theoretical analysis, and hence it is designed to match our theoretical setting. Our goal here is to understand the training dynamics of transformers under **analytically tractable** setting so that it can help us understand the mechanism behind self-attention and transformers.
>
> Regarding the reviewer's question on the generality of the phenomenon, we attached a pdf file that contains a preliminary experiment that we worked out. Figure 1 shows that for more general settings with multi-layer transformers, our main theoretical findings still hold: (i) the loss converges to the global minimum (i.e., with zero value) despite being highly nonconvex; (ii) the training process achieves correct detection with zero classification error as seen in phase 1 of our characterization, and the loss value continues to decrease to zero due to enlargement of the classification margin as seen in phase 2 of our characterization. We will continue to work on more experiments.
>
> We also point out that in multi-layer transformers, it is hard to interpret the meaning of the softmax attention in the middle layers since unlike the one layer case, the inputs to the middle layers are changing from iterations to iterations. Thus, it is hard to find a fixed subspace to project on and study its changes.
>
> **Comment:**
> In line 151 and 152, it is confusing why concentration theorems lead to the specific probability in (i) of Assumption 2.3.
>
> **Response:** Thanks for the question. Such an assumption helps to simplify our notations throughout the paper so that our main results can have simpler version and it is easy for readers to digest the main insight of the result.
>
> Those specific ratios can be satisfied with high probability if the size of the training set $n$ is large enough. This can be achieved by applying Hoeffding's inequality as follows. Take $I_1$ as an example. We have $\mathbb{P}\Big[|\frac{1}{n} \sum_{i=1}^n \mathbb{I}(X^{(i)} \in I_1) - \frac{1}{2}| \geq \sqrt{\frac{\log(2/\delta)}{n}}\Big] \leq \delta$.
>
> **Comment:**
> Lack of explanation for $\langle w_{j_1}^{(t)},w_{j_2}^{(t)} \rangle$ in line 194.
>
> **Response:** Thanks!
> This is the inner product of the neuron weights between $j_1$-th and $j_2$-th neuron.
> This term appears in calculating the update of $\nu^\top W_V^{(t)\top} W_V^{(t)} \mu$ and is needed to make the dynamical system complete. We will add this explanation to the revision.
>
> **Comment:**
> It is not obvious why "the samples with only one target signal may be classified incorrectly as co-occurrence" in line 282.
>
> **Response:**
> This is because during the beginning of training, the linear MLP layer will positively align with $\mu_1, \mu_2$, i.e., $G^{(t)}(\mu_1), G^{(t)}(\mu_2) > 0$, while for the common token $\mu_3$, we have $G^{(t)}(\mu_3) \approx 0$.
> At the same time, the linear MLP will also output something near zero for random tokens.
> Thus, for those samples $X$ with only $\mu_1$ or $\mu_2$ (i.e., $X \in I_2 \cup I_3$), we have $f^{(t)}(X) > 0$ which is incorrect as the samples in $I_2$ and $I_3$ has label $-1$.
>
> **Comment:**
> The notation in line 169 is somewhat misleading.
>
> **Response:**
> $p_{q\leftarrow \nu, k \leftarrow \mu}^{(i)}$ is the output of the softmax attention given input $X^{(i)}$ when the query is $\nu$ and key is $\mu$.
> In addition, $l(i,\mu)$ denotes the index in $X^{(i)}$ such that $X^{(i)}_{l(i,\mu)} = \mu$. We will add this explanation to the revision.
>
> We thank the reviewer again for your comments. We hope that our responses resolved your concerns. If so, we wonder if the reviewer could kindly consider to increase your score. Certainly, we are more than happy to answer your further questions.

---

> > ### Author Response · Authors · 2024-08-10
> > **A gentle reminder**
> >
> > Dear Reviewer G3j5,
> >
> > We've taken your initial feedback into careful consideration in our response. Could you please check whether our responses have properly addressed your concerns? If so, could you please kindly consider increasing your initial score accordingly? Certainly, we are more than happy to answer your further questions.
> >
> > Thank you for your time and effort in reviewing our work!
> >
> > Best Regards,
> > Authors

---

> ### Comment · Reviewer_G3j5 · 2024-08-12
>
> Thank you for your clarifications. Specifically, I have noticed the new experiment. However, these results remain preliminary and do not verify that the training dynamics of weights in this work can be generalized to broader cases. For example, the dynamics "first learning MLP and then learning Attention" cannot be adequately captured by the loss or accuracy presented.
>
> I will maintain my score.

---

> > ### Author Response · Authors · 2024-08-13
> >
> > We thank the reviewer very much for the highly inspiring question.
> >
> > To clarify the reviewer's concern, we note that the attention quantities that we study in one-layer case are the projections of attention to the input signals. In multi-layer transformers, the inputs to the middle layers are changing from iterations to iterations, and hence it is hard to find a fixed subspace to project on. Consequently, it is hard to find a natural and meaningful projection of the softmax attention in the middle layers to illustrate the phase change. Similarly, the inputs to the middle MLP layers are also changing, which makes it hard to find a fixed space for projection.
> >
> > This said, we still expect that the phase-wise learning (first MLP then attention) can occur for broader class of architectures. One feasible setting to demonstrate this is one-layer **multi-headed** transformer, which generalizes our original architecture and still maintains the meaningful projection of softmax attention to the fixed input space. We conducted additional experiments for one-layer 3-headed transformer. In our table below, we provide the values $G(\mu_1)$ of the MLP alignment with $\mu_1$ and the attention scores when the query is $\mu_1$ and key is $\mu_2$. The experiment results are all rounded to a tenth of decimal for better illustration. Take the first head as an example. The result clearly indicates a two-phase training process: (i) in phase 1 (during the first 250 steps), the MLP layer $G(\mu_1)$ grows quickly whereas the change in the attention score is very small compared to its later changes; and (ii) in phase 2 (after 250 steps), both MLP and attention score change substantially.
> >
> > | | Step | 0 | 250 | 500 | 750 | 1000 | 1250 | 1500 | 1750 |
> > |---|---|---|---|---|---|---|---|---|---|
> > | Head No.1 | score ($\times 10^{-2}$) |7.7 | 7.8 | 8.2 | 8.6 | 9.0 | 9.3 | 9.5 | 9.8 |
> > | | $G(\mu_1)$ | 0.8 | 4.8 | 8.0 | 10.7 | 13.0 | 14.8 |16.4 | 17.6 |
> > | Head No.2 | score ($\times 10^{-2}$) |2.1 | 2.2 | 2.6 | 3.0 | 3.3 | 3.6 | 3.9 | 4.1 |
> > | | $G(\mu_1)$ | -0.1 | 2.5 | 4.6 | 6.6 | 8.3 | 9.7 |10.9 | 11.9 |
> > | Head No.3 | score ($\times 10^{-2}$) |-12.4 | -12.2 | -11.7 | -11.1 | -10.6 | -10.2 | -9.9 | -9.6 |
> > | | $G(\mu_1)$ | 0.7 | 4.6 | 7.6 | 10.2 | 12.4 | 14.2 | 15.6 | 16.8 |
> >
> > We further emphasize that the main contribution of this paper lies in the development of the **new techniques** for analyzing the training dynamics of transformers, especially the joint optimization analysis of all transformer parameters, as Reviewer cFaT pointed out. We expect that these mathematical techniques can be generalized to more complicated transformer architectures in the future.
> >
> > We thank the reviewer again for your time and efforts. If our response resolved your concerns, could you please kindly consider to increase your score. We are also more than happy to answer your further questions.

---

> > > ### Author Response · Authors · 2024-08-13
> > > **A gentle reminder**
> > >
> > > Dear Reviewer G3j5,
> > >
> > > It occurs to us that we uploaded our response earlier and OpenReview didn't send out notification.
> > > This message serves as a gentle reminder for the reviewer to look at our most recent response.
> > >
> > > We once again appreciate the reviewer's time and effort.
> > > If our response resolved your concerns, could you please kindly consider to increase your current score accordingly?
> > > We are happy to hear your response.
> > >
> > > Best,
> > >
> > > Authors

---

### Author Rebuttal · Authors · 2024-08-06

Dear Reviewers,

Please find all the mentioned additional experiment results in rebuttals (to Reviewer G3j5 and Reviewer cFaT) in the attached pdf.

Thank you.

Best,

Authors

---

### Comment · Area_Chair_X85Y · 2024-08-12
**Please respond to rebuttal**

Dear reviewers,

The author-reviewer discussion period will end soon. Please make sure you read the authors' rebuttal and respond to it. If you have additional questions after reading the rebuttal please discuss with the authors.  For those who have done so, thank you!


AC

---

### Decision · Program_Chairs · 2024-09-25

**Decision:**

Accept (poster)

**Comment:**

In this paper the authors propose using gradient flow to investigate the training dynamics of a one-layer transform in a case study of word co-occurrence recognition.  The analysis does not impose commonly employed simplifications in literature such as weight reparameterization, linear attention, special initialization and lazy regime.  By analyzing the gradient flow the authors identify a two-phase training phenomena in the investigated transformer in the word co-occurrence recognition task.   The major novelty of the work is that the authors introduce a new proof technique that may benefit the machine learning community for analysis of training dynamics.  The major weakness of the paper which all reviewers agree upon is that the proposed gradient flow analysis is only conducted on the word co-occurrence task which is a very simple task.  Therefore, it is not clear whether the observed two-phase training dynamics can be generalized to a broader scope of applications.